# Alignment-Sensitive Minimax Rates for Spectral Algorithms with Learned Kernels

**Dongming Huang** [1]  **Zhifan Li** [2]  **Yicheng Li** [3]  **Qian Lin** [3]

## Abstract

We study spectral algorithms in settings where kernels may arise from a learning procedure, and ask how the resulting spectral order affects risk. We introduce the effective span dimension (ESD), an alignment-sensitive complexity measure that depends jointly on the target signal, the spectral order induced by the kernel, and the noise level $\sigma^2$. The ESD is defined without requiring eigen-decay conditions or source conditions, and it captures how much of the target signal lies in the leading spectral span. We prove that, for sequence models whose ESD is at most $K$, the minimax excess risk scales as $\sigma^2 K$, and we extend the framework to linear models and RKHS regression. Furthermore, we analyze over-parameterized gradient flow in a fixed-eigenbasis spectral learning model and prove that it can reduce the ESD under certain conditions. Together with numerical experiments, these results connect adaptive feature learning with reductions in ESD and offer a novel perspective on generalization beyond traditional fixed-kernel theories.

## 1. Introduction

Many modern learning procedures are adaptive: during training they update a representation, which effectively changes the induced kernel. Such adaptivity can improve generalization when it increases alignment between the learned kernel and the target function (Ba et al., 2022; Kunin et al., 2024; Liu et al., 2024; Bordelon et al., 2025; Xu & Ziyin, 2025; Zhang et al., 2024a). In contrast, classical theory for fixed kernel regression does not capture this effect. For instance, the Neural Tangent Kernel (NTK) theory approximates training dynamics as kernel regression (Jacot et al., 2018; Allen-Zhu et al., 2019), which enables the study of generalization using the classical regression theory in Reproducing Kernel Hilbert Spaces (RKHS) (Bauer et al., 2007; Yao et al., 2007). However, the NTK approximation is intrinsically non-adaptive and therefore cannot explain why finite-width networks, whose features evolve during training, often outperform traditional methods (Ghorbani et al., 2020; Gatmiry et al., 2021; Karp et al., 2021; Shi et al., 2023; Wenger et al., 2023; Seleznova & Kutyniok, 2022).

Recent simplified models isolate one tractable form of adaptivity: learning the kernel eigenvalues while keeping the eigenfunctions fixed. In these settings, reshaping the spectrum can increase signal-kernel alignment and lead to better performance (Li & Lin, 2024; 2025). More broadly, many analyses point to the same mechanism: generalization improves when the target places more energy on the leading kernel eigenfunctions (Arora et al., 2019; Woodworth et al., 2020; Kornblith et al., 2019; Radhakrishnan et al., 2024). However, classical RKHS theory is typically developed under fixed spectral assumptions, such as polynomial eigenvalue decay, together with signal regularity conditions, such as source conditions (Engl et al., 1996). If the kernel is learned, these assumptions are difficult to justify. To explain the advantages of adaptive kernel methods, we need a refined theoretical framework that remains valid beyond fixed-kernel assumptions.

In this paper, we propose the **Effective Span Dimension (ESD)** to bridge fixed-kernel theory and adaptive learning. The ESD is a population complexity measure to quantify signal-kernel alignment. Importantly, it is well defined for any kernel, including adaptive kernels whose eigenvalues and eigenfunctions evolve during training. Unlike classical measures that ignore the signal, ESD depends on the *signal, spectrum, and noise level* and directly relates to the minimax risk. It recovers classical rates when source and eigen-decay assumptions hold, while remaining meaningful for learned kernels where those assumptions may be unavailable. Our framework provides new theoretical insights that are absent in classical analyses. In particular, we achieve the following:

(i) We establish sharp minimax optimal convergence rates

---

[1]Department of Statistics and Data Science, National University of Singapore, Singapore [2]School of Statistics and Mathematics, Zhongnan University of Economics and Law, Wuhan, China [3]Department of Statistics and Data Science, Tsinghua University, Beijing, China. Correspondence to: Dongming Huang <stahd@nus.edu.sg>, Qian Lin <qianlin@tsinghua.edu.cn>.

*Proceedings of the 43rd International Conference on Machine Learning*, Seoul, South Korea. PMLR 306, 2026. Copyright 2026 by the author(s).

for sequence models using ESD.

(ii) We extend our definitions and theory to linear regression and kernel regression, which demonstrates the broad applicability of our framework.

(iii) We prove an ESD reduction result for a tractable eigenvalue-learning model with a fixed eigenbasis. For learned kernels whose eigenfunctions also evolve, the ESD remains well defined through the time-dependent eigensystem and is illustrated via a deep linear network experiment.

The ESD quantitatively explains the relationship between signal-kernel alignment and generalization. In our framework, successful adaptive learning can be interpreted as a reduction in ESD. The interpretation is as follows. Before adaptive training, the kernel can be badly aligned, so the ESD is large and the target lies in a minimax-hard family. After training, the kernel is reshaped so that the ESD becomes smaller, which moves the same target into an easier family where efficient spectral methods can generalize well.

Detailed comparisons with related notions of target–kernel alignment, balanced spectral cutoff, and effective dimension are deferred to Appendix A.

*Notations.* Write $a \lesssim b$ if there exists a constant $C > 0$ such that $a \leq Cb$, and write $a \asymp b$ if $a \lesssim b$ and $b \lesssim a$, where the dependence of the constants on other parameters is determined by the context. For $d \in \mathbb{N}_+$, let $[d] = \{1, 2, \ldots, d\}$; for $d = \infty$, let $[d] = \mathbb{N}_+$. $\mathbf{1}_{\{\cdot\}}$ denotes an indicator function.

## 2. Background on Kernel Methods

We first review kernel regression to provide context. Let $(\boldsymbol{x}_i, y_i)_{i=1}^n$ be i.i.d. samples from $y = f^*(\boldsymbol{x}) + \epsilon$, where $\boldsymbol{x} \sim \mu$ on a compact space $\mathcal{X}$, $\epsilon$ is an independent noise term with $\mathbb{E}[\epsilon] = 0$ and $\mathrm{Var}(\epsilon) = \sigma_0^2$. For an estimator $\hat{f}$ of the target function $f^*$, the excess risk is $\mathcal{R}(\hat{f}; f^*) = \mathbb{E}_{\boldsymbol{x} \sim \mu}[(\hat{f}(\boldsymbol{x}) - f^*(\boldsymbol{x}))^2]$.

A symmetric, positive-definite, and continuous kernel $\mathbf{k}(\cdot, \cdot) : \mathcal{X} \times \mathcal{X} \to \mathbb{R}$ induces an RKHS $\mathcal{H} \subset L^2(\mathcal{X}, \mu)$ with inner product $\langle \cdot, \cdot \rangle_{\mathcal{H}}$ and norm $\| \cdot \|_{\mathcal{H}}$ (Wahba, 1990; Schölkopf & Smola, 2001). Assume that $\mathbf{k}$ admits the spectral expansion

$$\mathbf{k}(\boldsymbol{x}, \boldsymbol{x}') = \sum_{j=1}^{\infty} \lambda_j \phi_j(\boldsymbol{x}) \phi_j(\boldsymbol{x}'), \quad \boldsymbol{x}, \boldsymbol{x}' \in \mathcal{X}, \quad (1)$$

where $\{\lambda_j\}_{j \geq 1}$ are the positive eigenvalues and $\{\phi_j\}_{j \geq 1} \subset \mathcal{H}$ is an orthonormal system in $L^2(\mathcal{X}, \mu)$. For background, see Steinwart & Christmann (2008); Steinwart & Scovel (2012).

Assuming that $f^*$ lies in the closed span of $\{\phi_j\}_{j \geq 1}$, kernel regression estimates $f^*$ using $f = \sum_j \beta_j \phi_j$ and regularizes via a filter of the kernel spectrum $\{\lambda_j\}$ (Rosasco et al., 2005;

Caponnetto & De Vito, 2007; Gerfo et al., 2008). If $f^*$ satisfies the *Hölder source condition* $\sum_j \langle f^*, \phi_j \rangle^2 / \lambda_j^s \leq R_s$ for some positive constants $s$ and $R_s$ (Engl et al., 1996; Mathé & Pereverzev, 2003) and the spectrum decays polynomially $\lambda_j \asymp j^{-\gamma}$, then the minimax rate is $n^{-s\gamma/(s\gamma+1)}$ (Yao et al., 2007; Li et al., 2024; Wang et al., 2024). The choice of kernels can significantly affect the performance (Li & Lin, 2024; Zhang et al., 2024a), so it is beneficial when the kernel eigenvalues align well with the expansion of the target function.

Since the kernel is usually chosen without knowing $f^*$, fixed-kernel methods may encounter misalignment. To address this limitation, adaptive methods have recently emerged. For instance, Li & Lin (2025) propose adapting kernel eigenvalues while fixing eigenfunctions. Specifically, they consider the kernel $\mathbf{k}_{\boldsymbol{a}}(x, x') = \sum_{j \geq 1} a_j^2 \phi_j(x) \phi_j(x')$ indexed by $\boldsymbol{a} = (a_j)_{j \geq 1}$ and the candidate $f = \sum_{j \geq 1} \beta_j a_j \phi_j$, where $a_j$'s and $\beta_j$'s are learned jointly via gradient flow. Such adaptation often improves performance, yet classical analyses built on fixed spectral assumptions do not explain these gains, because (a) adapted eigenvalues typically deviate from standard eigenvalue decay assumptions, and (b) it is unclear whether the classical source condition holds with respect to the adapted kernel, and if so, what the value of $s$ is.

Given the above limitation of classical analyses, we therefore seek a refined theoretical framework that explicitly captures signal-kernel alignment and explains the gains achieved by kernel adaptation.

**Bridge to the sequence model.** We connect RKHS regression with the sequence model to be analyzed in the next section. This bridge is motivational and the rigorous RKHS treatment is given in Appendix C. For any $j \in \mathbb{N}_+$, define

$$\theta_j^* = \langle f^*, \phi_j \rangle, \quad z_j = n^{-1} \sum_i y_i \phi_j(\boldsymbol{x}_i),$$
$$\text{and } \xi_j = n^{-1} \sum_i \epsilon_i \phi_j(\boldsymbol{x}_i). \quad (2)$$

The law of large numbers implies that for large $n$, $n^{-1} \sum_i \phi_j(\boldsymbol{x}_i) \phi_k(\boldsymbol{x}_i) \approx \mathbb{E}[\phi_j(\boldsymbol{x}) \phi_k(\boldsymbol{x})] = \mathbf{1}_{\{j=k\}}$, which implies that

$$z_j \approx \theta_j^* + \xi_j, \text{ and } \mathbb{E}[\xi_j] = 0,$$
$$\mathrm{Cov}(\xi_j, \xi_k) \approx n^{-1} \sigma_0^2 \mathbf{1}_{\{j=k\}}, \quad \forall j, k \in \mathbb{N}_+. \quad (3)$$

This reduction connects RKHS regression to a sequence model where the observations are $z_j = \theta_j^* + \xi_j$ and the noise terms $\{\xi_j\}$ are uncorrelated with variance $\sigma^2 := n^{-1} \sigma_0^2$. The error in the approximation due to finite $n$ will inflate the estimation variance compared to the sequence model. This approximation error can be controlled if $f^*$ is bounded; see Appendix C for a rigorous treatment.

## 3. Effective Span Dimension and Span Profile

To bridge existing theory and adaptive kernel methods as discussed in Section 2, we propose a novel framework to characterize the alignment between spectrum and signal. To focus on the main idea, we use the reduction in Equation (3) and first present our framework using sequence models.

**Sequence models.** A sequence model assumes observations are sampled as follows:

$$z_j = \theta_j^* + \xi_j, \quad 1 \leq j \leq d, \tag{4}$$

where $d \in \{\infty\} \cup \mathbb{N}_+$, $\boldsymbol{\theta}^* = (\theta_j^*)_{j=1}^d$ is a sequence of unknown parameters, $\xi_j$'s are uncorrelated random variables with mean zero and variance $\sigma^2$ (the noise level). For an estimator $\widehat{\boldsymbol{\theta}} = (\widehat{\theta}_j)_{j=1}^d$, we consider the loss $\mathcal{L}(\widehat{\boldsymbol{\theta}}; \boldsymbol{\theta}^*) = \sum_{j=1}^d (\widehat{\theta}_j - \theta_j^*)^2$ and risk $\mathcal{R}(\widehat{\boldsymbol{\theta}}; \boldsymbol{\theta}^*) = \mathbb{E}\mathcal{L}(\widehat{\boldsymbol{\theta}}; \boldsymbol{\theta}^*)$. The sequence model captures core estimation phenomena while permitting explicit analysis (Brown et al., 2002; Johnstone, 2017). In Appendix B, we use whitening to deal with correlated noise and analyze fixed-design linear regression. In Appendix C, we leverage the approximation in Equation (3) to analyze RKHS regression and random-design linear regression.

**Spectral estimators.** Given eigenvalues $\boldsymbol{\lambda} = (\lambda_j)_{j=1}^d$, spectral estimators take the form $\widehat{\theta}_j = (1 - \psi_\nu(\lambda_j)) z_j$, where $\psi_\nu(\lambda)$ is a filter such that larger $\nu$ induces more shrinkage. Some examples are:

Ridge (R): $\quad \psi_\nu^{\mathrm{R}}(\lambda) = \dfrac{1}{1 + \lambda/\nu}, \widehat{\theta}_j^{\mathrm{R},\nu} = \dfrac{\lambda_j}{\lambda_j + \nu} z_j.$ (5)

Gradient Flow (GF): $\quad \psi_\nu^{\mathrm{GF}}(\lambda) = e^{-\lambda/\nu},$

$$\widehat{\theta}_j^{\mathrm{GF},\nu} = \left(1 - e^{-\lambda_j/\nu}\right) z_j. \tag{6}$$

Principal Component (PC): $\quad \psi_\nu^{\mathrm{PC}}(\lambda) = \mathbf{1}_{\{\lambda < \nu\}},$

$$\widehat{\theta}_j^{\mathrm{PC},\nu} = \mathbf{1}_{\{\lambda_j \geq \nu\}} z_j. \tag{7}$$

For spectral estimators, the risk decomposes into squared bias $\sum_j (\psi_\nu(\lambda_j))^2 \theta_j^2$ and variance $\sum_j (1 - \psi_\nu(\lambda_j))^2 \sigma^2$, where $\nu$ controls the bias-variance trade-off. Classical analyses often assume $\boldsymbol{\theta}^*$ lies in an ellipsoid $\Theta_{\mathbf{a}} = \left\{\theta : \sum_j^\infty a_j^2 \theta_j^2 \leq C^2\right\}$ and derive convergence rates for sequences with $a_i \asymp i^\alpha$ (Johnstone, 2017). Our theoretical framework aims to bypass these assumptions.

### 3.1. Effective Span Dimension

Our goal is to quantify how estimation difficulty depends jointly on the signal $\boldsymbol{\theta}^*$, the spectrum $\boldsymbol{\lambda}$, and the noise variance $\sigma^2$. The definition below is motivated by the classical balanced spectral-cutoff principle, which balances the squared tail bias against the accumulated variance. Classical

inverse-problem and early-stopping work uses this cutoff to choose stopping rules for one prescribed operator (Blanchard & Mücke, 2018). Here we use the same crossing not as a stopping rule, but as a population index of the alignment between a target signal and the ordering induced by a spectrum.

Throughout, we assume $\{\lambda_j\}_{j \in [d]}$ are distinct; otherwise, any deterministic tie-breaking rule can be applied.

**Definition 3.1.** Let $\pi$ be an ordering of $[d]$ induced by decreasing $\lambda_j$; equivalently, $\lambda_{\pi_1} \geq \lambda_{\pi_2} \geq \dots$. We define the *Effective Span Dimension (ESD)* $d^\dagger$ of $\boldsymbol{\theta}^*$ w.r.t. the spectrum $\boldsymbol{\lambda}$ and variance $\sigma^2$ as

$$d^\dagger(\sigma^2; \boldsymbol{\theta}^*, \boldsymbol{\lambda}) = \min\{k \in [d] : \frac{1}{k} \sum_{i=k+1}^d (\theta_{\pi_i}^*)^2 \leq \sigma^2\}.$$

When $\lambda_j$ is decreasing, the summation in the above definition is simply the sum of the squared tail of $\boldsymbol{\theta}^*$. Using the indices $\pi_i$, the definition applies to a general ordering.

Equivalently, the ESD $d^\dagger$ is the smallest number $k$ of leading eigencoordinates (ordered by $\boldsymbol{\lambda}$) needed so that the remaining signal energy falls below $k\sigma^2$. It records how favorable the spectral ordering is for the target signal at noise level $\sigma^2$. This definition is at the population level and does not prescribe an estimation algorithm.

The following result records that $d^\dagger$ describes the oracle–tuned PC estimator, which follows from the standard bias–variance calculation (Blanchard & Mücke, 2018).

**Proposition 3.2.** *Let* $\widehat{\boldsymbol{\theta}}^{\mathrm{PC},\nu}$ *be the PC estimator in Equation (7) for the sequence model in Equation (4). Denote by* $\mathcal{R}_*^{\mathrm{PC}}$ *the minimal possible risk over all choices of* $\nu$. *Let* $d^\dagger = d^\dagger(\sigma^2; \boldsymbol{\theta}^*, \boldsymbol{\lambda})$ *be the ESD of* $\boldsymbol{\theta}^*$ *w.r.t. the spectrum* $\boldsymbol{\lambda}$ *and the variance* $\sigma^2$. *It holds that*

$$(d^\dagger - 1)\sigma^2 \leq \mathcal{R}_*^{\mathrm{PC}} \leq 2 d^\dagger \sigma^2.$$

In classical theories on sequence models, the oracle–tuned PC estimator is known to be minimax rate optimal under regularity assumptions that are analogous to the polynomial eigen-decay condition and the source condition in kernel regression (see Propositions 3.11 and 4.23 of Johnstone (2017)). In view of Proposition 3.2, those minimax rates should be of order of $O(d^\dagger \sigma^2)$. Indeed, we can use the ESD to characterize the intrinsic difficulty of estimation, which is the goal of the following theorem.

**Theorem 3.3.** *For any* $K \in [d]$, *spectrum* $\boldsymbol{\lambda} = \{\lambda_j\}_{j \in [d]}$, *and variance* $\sigma^2$, *define the ESD-bounded class*

$$\mathcal{F}_{K,\boldsymbol{\lambda}}^{(\sigma^2)} = \left\{\boldsymbol{\theta} \in \mathbb{R}^d : d^\dagger(\sigma^2; \boldsymbol{\theta}, \boldsymbol{\lambda}) \leq K\right\}. \tag{8}$$

*Suppose the sample* $\boldsymbol{Z}$ *is drawn from the sequence model in Equation (4) and consider any estimator* $\widehat{\boldsymbol{\theta}}$ *based on* $\boldsymbol{Z}$. *We*

*have*

$$\inf_{\widehat{\boldsymbol{\theta}}} \sup_{\boldsymbol{\theta}^* \in \mathcal{F}_{K,\boldsymbol{\lambda}}^{(\sigma^2)}} \mathcal{R}(\widehat{\boldsymbol{\theta}}, \boldsymbol{\theta}^*) \asymp K\sigma^2.$$

The class $\mathcal{F}_{K,\boldsymbol{\lambda}}^{(\sigma^2)}$ consists of parameter sequences whose ESDs are at most $K$. We interpret $K$ as the quota for ESD: the larger $K$, the larger $\mathcal{F}_{K,\boldsymbol{\lambda}}^{(\sigma^2)}$. Theorem 3.3 states that, over $\mathcal{F}_{K,\boldsymbol{\lambda}}^{(\sigma^2)}$, the minimax risk grows linearly with $K$. Theorem 3.3 confirms that the ESD quantifies the best possible performance of *any estimator* over the class $\mathcal{F}_{K,\boldsymbol{\lambda}}^{(\sigma^2)}$.

*Example* 3.4. For any given spectrum and $\boldsymbol{\theta}^*$, we can find the smallest $K^*$ such that $\boldsymbol{\theta}^* \in \mathcal{F}_{K^*,\boldsymbol{\lambda}}^{(\sigma^2)}$, over which the minimax risk is $\sigma^2 K^*$. In the following examples, $\{\lambda_i\}_{i=1}^d$ is any decreasing positive sequence.

**(1)** If $\theta_i^* = i^{-\alpha/2}$ for some $\alpha > 1$, then

$$\sigma^2 K^* \asymp \min\left\{\sigma^{2-\frac{2}{\alpha}}, d\sigma^2\right\}.$$

**(2)** If $d < \infty$, $\theta_i^* = i^{-1/2}$, and $\sigma^2 \leq \log d$, then

$$\sigma^2 K^* \asymp \begin{cases} d\sigma^2, & d\sigma^2 \leq e, \\ \log\left(d\sigma^2\right) - \log\log\left(d\sigma^2\right), & d\sigma^2 > e. \end{cases}$$

**(3)** If $d < \infty$, $0 < \alpha < 1$, and $\theta_i^* = i^{-\alpha/2}$, then

$$\sigma^2 K^* \asymp d\min\{d^{-\alpha}, \sigma^2\}.$$

Details and proofs are deferred to Appendix E.3.

*Example* 3.5. Suppose $\lambda_i = i^{-\beta}$. For some fixed $s, \beta, R > 0$, define the source ellipsoid

$$\Theta_s(R) = \left\{\boldsymbol{\theta} \in \mathbb{R}^d : \sum_{i=1}^d \lambda_i^{-s}\theta_i^2 \leq R\right\},$$

and the corresponding ESD quota

$$K_{\text{src}} = \min\left\{d, \left\lceil \left(\frac{R}{\sigma^2}\right)^{1/(1+s\beta)} \right\rceil\right\}.$$

We have

$$\Theta_s(R) \subseteq \mathcal{F}_{K_{\text{src}}(\sigma^2),\boldsymbol{\lambda}}(\sigma^2).$$

Moreover, this embedding is sharp in the worst case:

$$\sup_{\boldsymbol{\theta} \in \Theta_s(R)} d^\dagger(\sigma^2; \boldsymbol{\theta}, \boldsymbol{\lambda}) \asymp K_{\text{src}}.$$

Thus the source ellipsoid is contained in an ESD class of the classical size, and this ESD quota cannot be improved in general. See Appendix E.4.

If we take $\sigma^2 = \sigma_0^2/n$ to align with the approximation developed in Equation (3), then

$$\sigma^2 K_{\text{src}} \asymp \sigma_0^2 \min\left(n^{-\frac{s\beta}{1+s\beta}}, \frac{d}{n}\right),$$

which matches the well-known optimal rate under the source condition and the polynomial eigen-decay condition in the case when $d = \infty$. When $d < \infty$, there is a phase transition around $d_0 := n^{\frac{1}{1+s\beta}}$: if $d \lesssim d_0$, the upper bound is $d\sigma_0^2/n$; if $d \gtrsim d_0$, the upper bound is the same as if $d = \infty$. $\square$

The above examples suggest that the notion of ESD allows us not only to recover classical results but also to explore new settings where the classical framework is inapplicable.

We emphasize that the ESD is a population quantity for theoretical analysis, rather than an input supplied to a training algorithm. Although its definition uses the classical bias–variance crossing that appears in studies of early stopping for fixed operators, the ESD uses the crossing as a calibration point for alignment-sensitive statistical difficulty. More specifically, for a given pair of spectrum $\boldsymbol{\lambda}$ and noise variance $\sigma^2$, the value $d^\dagger(\sigma^2; \boldsymbol{\theta}^*, \boldsymbol{\lambda})$ is the smallest quota $K$ such that the ESD-bounded class $\mathcal{F}_{K,\boldsymbol{\lambda}}^{(\sigma^2)}$ contains the signal $\boldsymbol{\theta}^*$. By Theorem 3.3, the minimax risk over this class scales as $\sigma^2 d^\dagger(\sigma^2; \boldsymbol{\theta}^*, \boldsymbol{\lambda})$. A detailed comparison between the ESD and balanced spectral cutoffs is given in Appendix A.1.

This interpretation is useful for learned kernels because the same target can be evaluated under different learned spectra. Comparing the resulting ESD values compares the statistical difficulty induced by different signal–spectrum alignments. Appendix D.1 gives a sparse-signal example in which ESD distinguishes two spectra with the same set of eigenvalues but different allocations of leading eigenvalues. Signal-agnostic quantities such as *effective dimensions* (Zhang, 2005) and *effective ranks* (Bartlett et al., 2020) depend only on the spectrum and cannot distinguish these allocations, so they do not quantify this alignment effect. Appendix A.2 contrasts ESD with these measures.

We remark that the ESD is not the only measure for quantifying signal–spectrum alignment or, more generally, target–kernel alignment. Classical target–kernel alignment measures have been used widely in empirical studies, but they do not directly provide the same noise-indexed minimax characterization; see Appendix A.3. Although the bias–variance crossing principle can also be used in other estimators, we focus on the ESD because it admits a clean analytic characterization and enables the study of dynamic signal–kernel alignment in adaptive learning. Appendix A.4 also explains why such estimator-guided alternatives may reflect limitations of the estimator rather than the signal–spectrum difficulty measured by the ESD.

## 3.2. Span Profile

The definition of ESD explicitly depends on the noise level $\sigma^2$, which distinguishes it from other complexity measures in the literature. The dependence on $\sigma^2$ reflects the bias-variance trade-off nature of ESD: as $\sigma^2$ decreases, more coordinates can be estimated unbiasedly while controlling the overall variance, thereby removing more bias. To focus on the alignment between a given signal $\boldsymbol{\theta}^*$ and a spectrum $\boldsymbol{\lambda}$, we examine the ESD by varying the noise level.

**Definition 3.6.** We define the *span profile* of $\boldsymbol{\theta}^*$ w.r.t. the spectrum $\boldsymbol{\lambda}$ as $\mathbf{D}_{\boldsymbol{\theta}^*,\boldsymbol{\lambda}} : \tau \mapsto d^\dagger(\tau; \boldsymbol{\theta}^*, \boldsymbol{\lambda})$.

The span profile $\mathbf{D}_{\boldsymbol{\theta}^*,\boldsymbol{\lambda}}$ is a well-defined object that depends only on $\boldsymbol{\theta}^*$ and the ordering of $\boldsymbol{\lambda}$, and it summarizes how $\sigma^2$ affects the ESD. Proposition 3.2 suggests that for two spectra $\boldsymbol{\lambda}^{(1)}$ and $\boldsymbol{\lambda}^{(2)}$, we can compare their alignments with the signal by the ratio of $r(\tau) = \mathbf{D}_{\boldsymbol{\theta}^*,\boldsymbol{\lambda}^{(1)}}(\tau)/\mathbf{D}_{\boldsymbol{\theta}^*,\boldsymbol{\lambda}^{(2)}}(\tau)$ for small $\tau$, because, if this ratio is very small (and in particular if the limit is 0 for $\tau \to 0$), then an oracle–tuned PC estimator using $\boldsymbol{\lambda}^{(1)}$ can achieve a smaller risk than one that uses $\boldsymbol{\lambda}^{(2)}$. Such comparisons are not as convenient in classical theory. See Appendix D for more illustrations.

A closely related object is the *trade-off function* of $\boldsymbol{\theta}^*$ relative to $\boldsymbol{\lambda}$, which is defined for any $k \in [d]$ as

$$\mathbf{H}_{\boldsymbol{\theta}^*,\boldsymbol{\lambda}}(k) \;=\; \frac{1}{k} \sum_{i=k+1}^{d} \left(\theta_{\pi_i}^*\right)^2 \;=\; \frac{1}{k} \sum_{i:\, \lambda_i < \lambda_{\pi_k}} (\theta_i^*)^2. \quad (9)$$

The quantity $\sigma^{-2}\mathbf{H}_{\boldsymbol{\theta}^*,\boldsymbol{\lambda}}(k)$ equals the bias-variance ratio of the PC estimator using the $k$ leading coordinates. Properties of span profiles and trade-off functions are summarized as follows.

**Proposition 3.7.** *(1) Both* $\mathbf{D}_{\boldsymbol{\theta}^*,\boldsymbol{\lambda}} : \tau \mapsto [d]$ *and* $\mathbf{H}_{\boldsymbol{\theta}^*,\boldsymbol{\lambda}} : [d] \mapsto [0,\infty)$ *are nonincreasing. (2) For any* $\tau$, *it holds that* $\mathbf{D}_{\boldsymbol{\theta}^*,\boldsymbol{\lambda}}(\tau) = \min\{k \in [d] : \mathbf{H}_{\boldsymbol{\theta}^*,\boldsymbol{\lambda}}(k) \leq \tau\}$. *(3) For two spectra* $\boldsymbol{\lambda}^{(1)}$ *and* $\boldsymbol{\lambda}^{(2)}$, *if* $\mathbf{H}_{\boldsymbol{\theta}^*,\boldsymbol{\lambda}^{(1)}}(k) \leq \mathbf{H}_{\boldsymbol{\theta}^*,\boldsymbol{\lambda}^{(2)}}(k)$ *for all* $k \in [d]$, *then* $\mathbf{D}_{\boldsymbol{\theta}^*,\boldsymbol{\lambda}^{(1)}}(\tau) \leq \mathbf{D}_{\boldsymbol{\theta}^*,\boldsymbol{\lambda}^{(2)}}(\tau), \quad \forall \tau > 0$.

Property (3) in Proposition 3.7 suggests that the faster $\mathbf{H}_{\boldsymbol{\theta}^*,\boldsymbol{\lambda}}(\cdot)$ decreases, the better the spectrum $\boldsymbol{\lambda}$ aligns with the signal $\boldsymbol{\theta}^*$. In the extreme case where the ordering of $\lambda_i$ matches the ordering of $|\theta_i^*|^2$, the decay of $\mathbf{H}_{\boldsymbol{\theta}^*,\boldsymbol{\lambda}}(\cdot)$ is the fastest, which leads to the most favorable span profile.

**Extensions.** To save space, we defer the extensions to linear models and kernel regression to Appendices B and C, respectively. For the kernel regression model in Equations (1) and (2), we define the ESD of $f^*$ w.r.t. the kernel $\mathbf{k}$ and the effective noise variance $\sigma^2 := (\sigma_0^2 + \|f^*\|_\infty^2)/n$ as

$$d^\dagger(\sigma^2; f^*, \mathbf{k}) = \min\{k \in \mathbb{N}_+ \cup \{\infty\} : \mathbf{H}_{\boldsymbol{\theta}^*,\boldsymbol{\lambda}}(k) \leq \sigma^2\}.$$

## 4. Minimax Optimal Convergence Rates

When using the span profile to characterize the signal-spectrum alignment, it is of interest to establish the optimal convergence rates. Since the setting where $d = d_n$ grows along with $n$ has been studied in Theorem 3.3, we focus on the case where $d = \infty$ and the spectrum $\boldsymbol{\lambda}$ is given with ordering denoted by $\{\pi_j\}$ such that $\lambda_{\pi_1} > \lambda_{\pi_2} > \dots$. For the convenience of the asymptotic analysis, we examine the span profile at $\tau = \sigma_0^2/n$, where $\sigma_0^2$ is fixed and $n$ enumerates $\mathbb{N}_+$.

We begin by defining a class of populations whose span profile is bounded by a sequence of nondecreasing quotas $\boldsymbol{K} = \{K_n\}_{n=1}^\infty$. Given any fixed integer $n_0 \in \mathbb{N}_+$, we consider the following class of parameters:

$$\mathcal{F}_{\boldsymbol{K},\boldsymbol{\lambda}} := \left\{\boldsymbol{\theta} \in \mathbb{R}^\infty : \mathbf{D}_{\boldsymbol{\theta},\boldsymbol{\lambda}}\!\left(\frac{\sigma_0^2}{n}\right) \leq K_n, \forall\, n \geq n_0\right\}. \quad (10)$$

For each $\boldsymbol{\theta} \in \mathcal{F}_{\boldsymbol{K},\boldsymbol{\lambda}}$, the sequence model in Equation (4) with $\boldsymbol{\theta}^* = \boldsymbol{\theta}$ and $\sigma^2 = \sigma_0^2/n$ will have an ESD no greater than $K_n$. For a sample $\boldsymbol{Z}^{(n)}$ from this sequence model and any estimator $\widehat{\boldsymbol{\theta}}$ based on $\boldsymbol{Z}^{(n)}$, we aim to determine the convergence rate of the following minimax risk:

$$\inf_{\widehat{\boldsymbol{\theta}}} \sup_{\boldsymbol{\theta} \in \mathcal{F}_{\boldsymbol{K},\boldsymbol{\lambda}}} \mathcal{R}(\widehat{\boldsymbol{\theta}}, \boldsymbol{\theta}). \quad (11)$$

We emphasize that $\boldsymbol{K}$ is a *model-class descriptor*. It is not a parameter of the distribution, but rather describes a condition on the distribution. For example, the sparsity assumption in high-dimensional regression states that $\|\boldsymbol{\beta}\|_0 \leq s$, so $s$ describes a class of distributions; yet $s$ is not a parameter of the distribution. Our minimax result requires a regularity condition on the quota sequence $\boldsymbol{K}$. Let $\bar{K} := \sup\{K_n\} \in \mathbb{N} \cup \{\infty\}$. For any $k \in [\bar{K}]$, let $M_k := \max\{n : K_n = k\}$.

**Condition 4.1.** (1) $0 \leq K_{n+1} - K_n \leq 1$ for all $n$. (2) For all $1 \leq k \leq \bar{K} - 1$, it holds that $(k+1)/M_{k+1} \leq k/M_k$.

Condition 4.1 ensures that $K_n$ does not grow faster than $n$, and the ratio sequence $\{k/M_k\}$ is nonincreasing. Condition 4.1 is easily satisfied by common growth laws.

*Example* 4.2. (1) Suppose $K_n \asymp n^a$ where $0 < a < 1$. For any $k$, we have $M_k \asymp k^{1/a}$. Since $k/k^{1/a}$ is decreasing, Condition 4.1 holds.
(2) Suppose $K_n \asymp (\log n)^b$ where $b > 0$. For any $k$, we have $M_k \asymp e^{k^{1/b}}$. Since $k/e^{k^{1/b}}$ is decreasing, Condition 4.1 holds.

The next theorem provides a lower bound on the minimax risk in Equation (11).

**Theorem 4.3.** *Suppose Condition 4.1 holds for a quota sequence* $\boldsymbol{K} = \{K_n\}_{n=1}^\infty$. *Let* $c_0 = 1/4$. *If* $\boldsymbol{Z}^{(n)}$ *is drawn*

*from the sequence model with $\boldsymbol{\theta}^* = \boldsymbol{\theta}$ and $\sigma^2 = \sigma_0^2/n$, it holds that*

$$\inf_{\widehat{\boldsymbol{\theta}}} \sup_{\boldsymbol{\theta} \in \mathcal{F}_{\boldsymbol{K},\boldsymbol{\lambda}}} \mathcal{R}(\widehat{\boldsymbol{\theta}}, \boldsymbol{\theta}) \geq c_0 \sigma_0^2 \frac{K_n}{n}.$$

Theorem 4.3 shows that given a quota sequence $\boldsymbol{K}$, no estimator can, uniformly over the class $\mathcal{F}_{\boldsymbol{K},\boldsymbol{\lambda}}$, achieve a faster convergence rate of risk than $\sigma_0^2 K_n/n$. The matching upper bound is achieved by the PC estimator retaining the top $K_n$ coordinates. We thus conclude that the minimax optimal rate over $\mathcal{F}_{\boldsymbol{K},\boldsymbol{\lambda}}$ is $\sigma_0^2 K_n/n$.

Since our theory does not invoke any source condition or eigenvalue-decay condition, it goes beyond the classical analysis in the literature. It suggests that the ESD is an essential quantity, and the span profile provides a useful characterization of the attainable error rate for spectral methods.

To see why our framework is more general, the next example presents a case where the minimax convergence rate is slower than the rate in the fixed-dimensional setting while being faster than the standard rate in classical infinite-dimensional settings.

*Example* 4.4. Let $b \geq 1$ be a constant and $K_n = \lceil (\log n)^{1/b} \rceil$ for $n \in \mathbb{N}_+$. Suppose $\{\lambda_j\}_{j=1}^\infty$ is decreasing and $\theta_{j+1}^* = \sqrt{\sigma_0^2 \left[ je^{-j^b} - (j+1)e^{-(j+1)^b} \right]}$ for $j \geq 1$ and $\theta_1^* = 0$. Then, $\boldsymbol{\theta}^* \in \mathcal{F}_{\boldsymbol{K},\boldsymbol{\lambda}}$ and the optimal rate is $\sigma_0^2 (\log n)^{1/b} n^{-1}$. In contrast, the traditional convergence rate based on the source condition and polynomial eigendecay is $\sigma_0^2 n^{-\gamma/(1+\gamma)}$ for some $\gamma > 0$, which is not sharp.

**Connecting adaptivity and generalization.** Our span profile framework clarifies why adaptive machine learning methods often outperform classical fixed-kernel approaches. In classical methods with a fixed kernel spectrum $\boldsymbol{\lambda}^{(0)}$, the target signal $\boldsymbol{\theta}^*$ might exhibit poor alignment, resulting in a large span profile $\mathbf{D}_{\boldsymbol{\theta}^*,\boldsymbol{\lambda}^{(0)}}$. Consequently, the signal resides in a class $\mathcal{F}_{\mathbf{K}^{(0)},\boldsymbol{\lambda}^{(0)}}$ with a large quota sequence $\mathbf{K}^{(0)}$, which implies high minimax risk. By contrast, adaptive methods modify the kernel during training. Successful adaptation improves the alignment by reducing the span profile $\mathbf{D}_{\boldsymbol{\theta}^*,\boldsymbol{\lambda}^{(a)}}$ of the same signal w.r.t. the adapted kernel spectrum $\boldsymbol{\lambda}^{(a)}$. This adaptation places the signal in a class $\mathcal{F}_{\mathbf{K}^{(a)},\boldsymbol{\lambda}^{(a)}}$ with a smaller quota sequence $\mathbf{K}^{(a)}$, which implies lower minimax risk.

# 5. Adaptive Eigenvalues via Over-parameterized Gradient Flow

This section will investigate the benefits of learning eigenvalues via over-parameterized gradient flow (OP-GF) in sequence models (Li & Lin, 2024) through the lens of ESDs. We provide an adaptive result in a tractable setting with eigenvalue learning under a fixed eigenbasis. For learned kernels with evolving eigenfunctions, Appendix D.2 defines the pathwise ESD using the time-dependent eigensystem and gives supporting experimental evidence; a general theorem for evolving eigenfunctions is left for future work.

Inspired by the over-parameterized nature of deep neural networks, we follow the scalar deeper over-parameterization in Li & Lin (2024) and set $\theta_j = a_j b_j^D \beta_j$, where the integer $D$ is the depth parameter and $(a_j, b_j, \beta_j)$ are learned parameters. The gradient flow w.r.t. the sequence-model loss $L = \frac{1}{2} \sum_j (\theta_j - z_j)^2$ is (for all $j \in [d]$)

$$\begin{aligned}
\dot{a}_j &= -\nabla_{a_j} L = b_j^D \beta_j (z_j - \theta_j), \\
\dot{b}_j &= -\nabla_{b_j} L = D a_j b_j^{D-1} \beta_j (z_j - \theta_j), \\
\dot{\beta}_j &= -\nabla_{\beta_j} L = a_j b_j^D (z_j - \theta_j), \\
a_j(0) &= \lambda_j^{1/2}, \quad b_j(0) = b_0 > 0, \quad \beta_j(0) = 0,
\end{aligned} \tag{12}$$

where $\{\lambda_j\}_{j \in [d]}$ are the initial eigenvalues and $b_0$ is the common initialization of all $b_j$. At time $t$, the learned eigenvalues are given by $\tilde{\lambda}_j(t) = (a_j(t) b_j^D(t))^2$ and the OP-GF estimates are $\hat{\theta}_j^{OP}(t) = \tilde{\lambda}_j^{\frac{1}{2}}(t) \beta_j(t)$ for $j \in [d]$. Li & Lin (2024) consider infinite-dimensional sequence models with a polynomial decay condition on the initial eigenvalues and establish upper bounds on the risk of the OP-GF estimator with proper early stopping.

Here we study the dynamics of eigenvalues in OP-GF and how it changes the ESD. At time $t$, the learned eigenvalues are $\tilde{\boldsymbol{\lambda}}(t) := (\tilde{\lambda}_j(t))_{j \in [d]}$, and the ESD is $d^\dagger(t) = d^\dagger(\sigma^2; \boldsymbol{\theta}^*, \tilde{\boldsymbol{\lambda}}(t))$. We aim to show that under some conditions on the underlying signal and initial parameters, OP-GF can adjust the ordering of eigenvalues $\tilde{\boldsymbol{\lambda}}(t)$ to reduce the ESD $d^\dagger(t)$, which leads to a better signal-spectrum alignment.

We begin with some notation for the sequence model in Equation (4). We focus on the *large-sample case* where $\sigma^2 = \frac{\sigma_0^2}{n}$ and $\sigma_0 = 1$ without loss of generality. Denote $\tilde{d} = \sum_{i=1}^d \lambda_i$ (i.e., sum of initial eigenvalues). Assume $\lambda_j > 0$ for all $j$ and $\tilde{d} < \infty$. Throughout the large-sample statements below, the initial spectrum $(\lambda_j)_{j \in [d]}$ is a fixed-kernel quantity and does not vary with $n$. Let $\pi_t^{-1}(i)$ denote the rank of $\tilde{\lambda}_i(t)$ at time $t$. All ranks are resolved using a fixed deterministic tie-breaking rule.

**Assumption 5.1.** (1) Each noise $\xi_j$ in Equation (4) is sub-Gaussian with variance proxy bounded by $C_{\text{proxy}} \sigma^2$. (2) Let $\varepsilon = 2C_{\text{proxy}}^{1/2} n^{-1/2} \sqrt{\ln(e + n\tilde{d}) \cdot \ln n}$ and $\varepsilon' = 2C_{\text{proxy}}^{1/2} n^{-1/2} \sqrt{\ln n}$. Define $S := \{j \in [d] : |\theta_j^*| > \varepsilon\}$. We have $|S| \leq n$. (3) $\inf_{j \in S} \lambda_j > n^{-\delta}$ for some $\delta \in (0, 1)$.

**Theorem 5.2.** *Suppose that Assumption 5.1 holds and $D \geq 1$ is fixed. There exist some constants $c$, $C$, $C_M$, $C_{max}$, $C_\eta$, $c_\eta$, $c_B$, $c'$, and $\delta_{\text{w}} \in (0,1)$, and an integer $n_0$, depending*

*only on $D$, $\delta$, $C_{proxy}$, and $\tilde{d}$, but not on $n$, $\theta^*$, or $t_1$, such that the following holds for every $n \geq n_0$. Assume the initialization in Equation* (12) *satisfies $b_0 = c_B D^{\frac{D+1}{D+2}} \varepsilon^{\frac{1}{D+2}}$. Define $t_2 = C \cdot D^{\frac{D}{D+2}} (\varepsilon)^{-\frac{2D+2}{D+2}} \ln n$. Then, on an event $\mathcal{E}$ with probability at least $1 - 4/n$, we have*

$$d^\dagger(t_2) \leq d^\dagger(t_1)$$

*for every $t_1 \in [0, t_2)$ with $d^\dagger(t_1) < \infty$ whenever the following conditions hold:*

1. *For any $j \in S$, we have $M \leq |\theta_j^*|$, where $M := C_M \varepsilon$;*

2. *For any $j \in S^c$, we have $|\theta_j^*| \leq \tilde{\sigma}$, where $\tilde{\sigma} = c' \varepsilon$.*

3. *For any $i, j \in S$, let $\eta_{i,j} := |\theta_i^*| - |\theta_j^*|$. At least one of the following holds: (a) $\eta_{i,j} \leq 0$, (b) $\eta_{i,j} \geq C_\eta \varepsilon$ and $|\theta_i^*| \leq C_{max} M$, or (c) $\frac{|\theta_i^*|}{|\theta_j^*|} > (1 + \frac{c_\eta}{D})$.*

4. *At time $t_1$, define two subsets of $S^c$: $A_1 := \{i \in S^c : \pi_{t_1}^{-1}(i) \leq d^\dagger(t_1), \tilde{\lambda}_i(t_1) < c \cdot b_0^{2D} D^{-\frac{D}{D+2}} \cdot M^{\frac{2}{D+2}}\}$ and $B_1 := \{i \in S^c : \pi_{t_1}^{-1}(i) > d^\dagger(t_1)\}$, and define a subset of $S$: $B_2 := \{i \in S : \pi_{t_1}^{-1}(i) > d^\dagger(t_1)\}$. It holds that with $C_{B_1} := \min\{(|A_1| - |B_2|)_+, |B_1|\}$, the following four conditions hold: (i) for every $j \in B_2$, $\tilde{\lambda}_j(t_1) < c \cdot b_0^{2D} D^{-\frac{D}{D+2}} \cdot M^{\frac{2}{D+2}}$; (ii) for every $j \in B_1$, either $\tilde{\lambda}_j(t_1) < c \cdot b_0^{2D} D^{-\frac{D}{D+2}} \cdot \varepsilon^{\frac{2}{D+2}}$ or $\min_{\ell : \pi_{t_1}^{-1}(\ell) \leq d^\dagger(t_1)} \tilde{\lambda}_\ell(t_1) \geq (1 + \delta_w) \tilde{\lambda}_j(t_1)$; (iii) for every $j \in S^c$ satisfying $\pi_{t_1}^{-1}(j) \leq d^\dagger(t_1)$ and $j \notin A_1$, $\tilde{\lambda}_j(t_1) \geq (1 + \delta_w) c b_0^{2D} D^{-\frac{D}{D+2}} M^{\frac{2}{D+2}}$; (iv) $|B_2| + C_{B_1} \leq |B_2| \left(\frac{C_M}{c'}\right)^2$.*

Theorem 5.2 shows that OP-GF can weakly reduce the ESD under some conditions.

Conditions (1)–(2) impose a dichotomy: coordinates in $S$ carry signal at least $M$, whereas coordinates in $S^c$ carry signal at most $\tilde{\sigma}$. Condition (3) excludes problematic near-ties among strong coordinates, so that the OP-GF dynamics can preserve or improve their relative learned-spectrum ordering. Condition (4) concerns the time $t_1$: the learned spectrum may be misaligned, but not so severely that too many strong coordinates lie behind weak coordinates near the ESD cutoff. Appendix H gives an illustrative verification of these conditions in a concrete misaligned-spectrum example. Under these conditions, OP-GF amplifies coordinates carrying strong signal faster than weak ones, improves the endpoint ordering of the learned eigenvalues, and reduces the signal tail sorted by the learned spectrum. This yields the endpoint comparison $d^\dagger(t_2) \leq d^\dagger(t_1)$.

Theorem 5.2 also indicates a bottleneck for successful adaptation. Condition (4) rules out states where too many strong

coordinates are buried behind weak coordinates near the current ESD or weak top coordinates lack a sufficient learned-spectrum margin. If such a configuration occurs, the OP-GF may not reorder the learned eigenvalues enough by $t_2$, and the endpoint ESD reduction guarantee need not hold.

# 6. Numerical Experiments

**Data Generation.** We utilize the **misalignment** setting in Li & Lin (2024) to specify a $d$-dimensional sequence model. We fix the eigen-decay rate $\gamma > 0$, the signal decay rate $p > 0$, and the number of nonzero signals $J$. Given any misalignment parameter $q \geq 1$, we set eigenvalues as $\lambda_j = j^{-\gamma}, j \in [d]$, and set the true nonzero parameters as $\theta_{\ell(j)}^* = C \cdot j^{-\frac{p+1}{2}}$, where $\ell(j) = [j^q]$ and $j \leq J$. Here all other elements of $\theta_j^*$ are zero and $d \geq J^q$ so $\|\boldsymbol{\theta}^*\|_0 = J$. The observations are sampled as $z_i \sim N(\theta_i^*, \sigma^2)$. This setting provides a flexible way to control the alignment between the signal structure and the spectrum. When $q = 1$, the ordering of $\boldsymbol{\theta}^*$ aligns perfectly with the ordering of $\boldsymbol{\lambda}$. As $q$ increases, more nonzero elements of $\boldsymbol{\theta}^*$ are located on the tail where the eigenvalues are smaller, and more large eigenvalues are associated with zero signals, creating a worse signal-spectrum alignment.

**Evolution of Span Profile.** The first experiment visualizes the span profile of the signal w.r.t. the learned spectrum at various stages in the OP-GF process with $D = 0$ (i.e., Equation (12) without $b_j$). Given a sample, we approximate the gradient flow in Equation (12) by discrete-time gradient descent and obtain the solution $\{(a_j(t), \beta_j(t))_{t \geq 0} : j \in [d]\}$. The trained eigenvalue sequence $\tilde{\boldsymbol{\lambda}}(t)$ at time $t$ is given by $\tilde{\lambda}_j(t) = a_j^2(t)$ for $j \in [d]$. We focus on time points before the optimal stopping time.

Figure 1 illustrates the evolution of the span profile w.r.t. the learned spectrum for different training times $t$ and various values of $q$. When $q = 1$ (Top-Left panel), the span profiles at different training times $t$ are nearly identical. This is because the initial spectrum already aligns perfectly with the signal and there is no room for improvement. For $q > 1$ (Top-Right, Bottom-Left, Bottom-Right panels), we observe that as the training time $t$ increases from $0$ to $80$, the span profile shifts downwards. This suggests that the training process refines the alignment between the spectrum and the signal. In addition, the reduction in the span profile is more significant for $q = 3$ compared to $q = 1.5$, because $q = 3$ corresponds to a greater initial misalignment between the signal and the spectrum, rendering the improvement from OP-GF more substantial.

**Evolution of ESD and Estimation Error of PC Estimators.** We next empirically investigate the evolution of the ESD $d^\dagger$ and the estimation error as well as the impact of

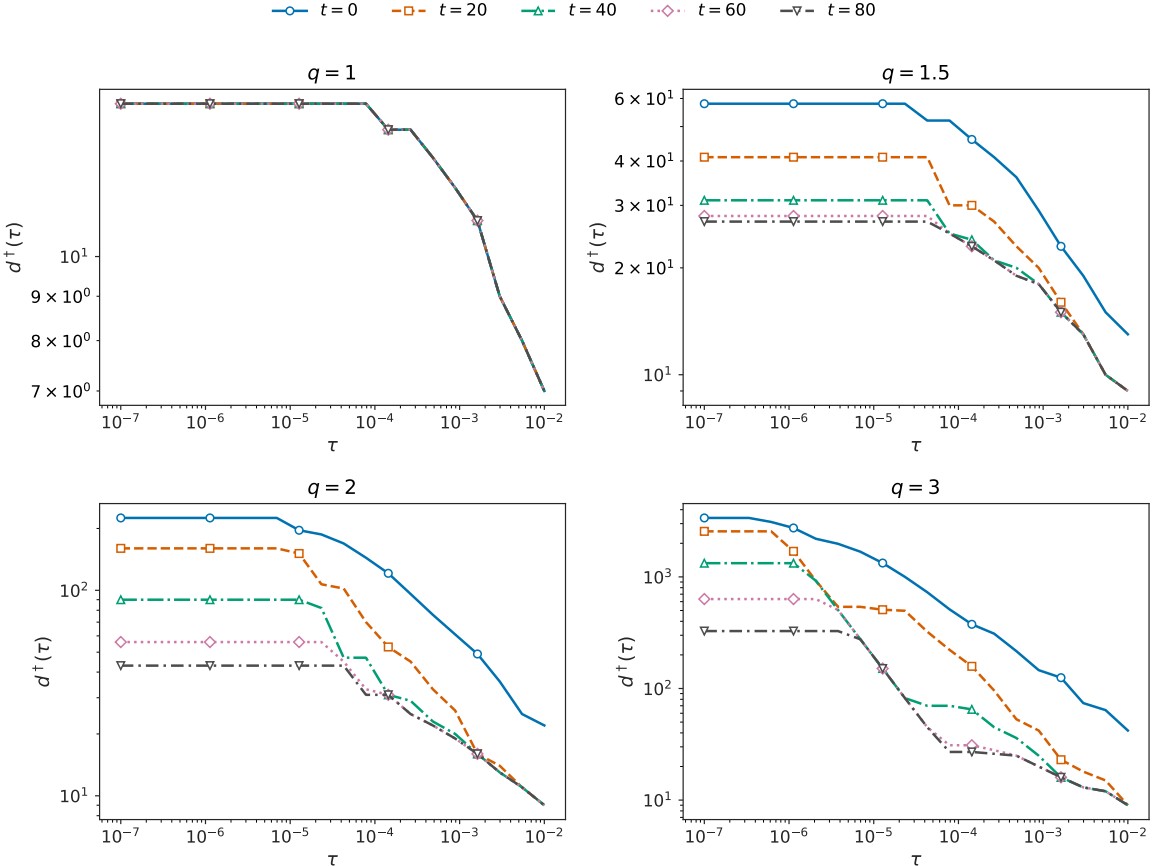

*Figure 1.* Evolution of span profiles during the training of an over-parameterized gradient flow. The misalignment level $q$ varies from 1 to 3. Fixed parameters are $n = 10000$, $\sigma_0 = 1$, $d = 5000$, $J = 15$, $p = 2.5$, and $\gamma = 1$.

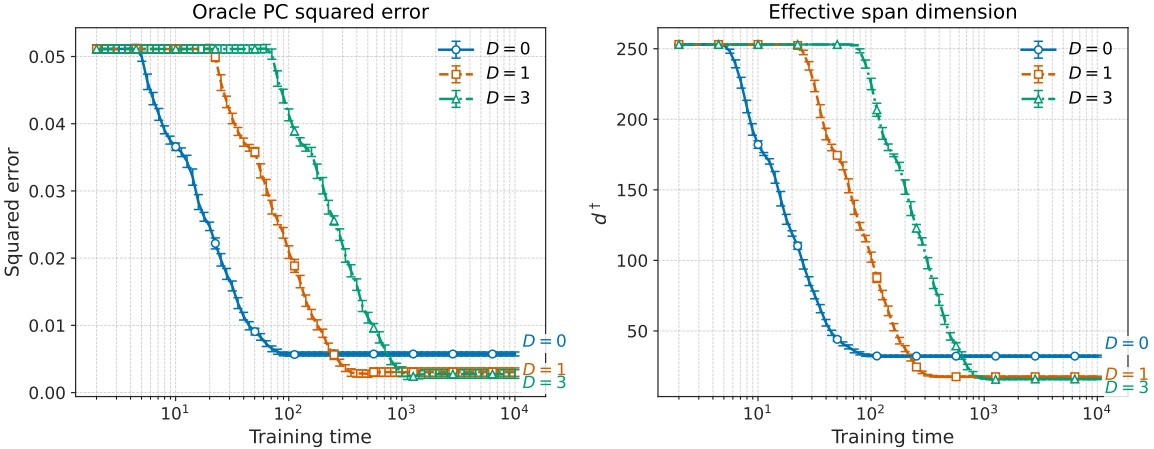

*Figure 2.* Averaged squared error of the oracle–tuned PC estimator and ESD as a function of the training time. Each average is computed based on 20 replications and each error bar represents one standard error.

the number of layers $D$. At any time $t$, we compute the ESD $d^\dagger(t)$ based on the learned eigenvalue sequence $\tilde{\boldsymbol{\lambda}}(t)$ and also the oracle–tuned PC estimate $\widehat{\boldsymbol{\theta}}(t)$ based on $\tilde{\boldsymbol{\lambda}}(t)$, with the number of components determined by $d^\dagger(t)$. Proposition 3.2 and Theorem 3.3 suggest that the PC estimator tuned by the ESD can achieve the minimax risk rate, so we expect $\widehat{\boldsymbol{\theta}}(t)$ to perform well.

The empirical evaluation involved 20 Monte Carlo repetitions. Figure 2 displays the averaged estimation error of $\widehat{\boldsymbol{\theta}}(t)$ and the averaged $d^\dagger(t)$ as functions of training time $t$. We observe that both the ESD and the squared error of the tuned PC estimator exhibit a general decay trend over training time $t$. Furthermore, for the shallow model with $D = 0$ (with no $b_{j,i}$ parameters), the initial decrease in ESD and MSE occurs earlier compared to the deeper models with $D = 1$ or $D = 3$. However, with sufficient training iterations, the deeper models with $D = 1$ or $D = 3$ can achieve lower ESD values than the shallow model with $D = 0$. These findings suggest that increased model depth ($D > 0$) may facilitate a better adaptation of the spectrum, and thus lead to lower estimation error. This observation offers a perspective on the benefits of depth in spectral learning, but a comprehensive study for general models is left for future research.

## 7. Discussion

This paper introduces the effective span dimension (ESD) and span profile to analyze the interplay between the signal structure and the kernel spectrum. Our framework moves beyond classical static assumptions relative to a fixed kernel (e.g., source conditions and polynomial eigenvalue decay) and offers a dynamic, noise-dependent perspective on signal complexity. Unlike traditional source conditions, the ESD is more flexible and remains applicable when the spectrum itself is learned from data.

Like sparsity in high-dimensional statistics, ESD is a population-level descriptor of a statistical problem rather than a quantity that the training algorithm must know. Its role in this paper is explanatory and comparative: it quantifies how favorable a kernel is for a given signal at a given noise level, and it determines the minimax difficulty of the corresponding ESD-bounded class.

The framework suggests a route for studying learned representations: when adaptive learning reshapes the kernel so that the same target has a smaller ESD, the target moves into an easier minimax class, and thus the corresponding spectral estimation becomes statistically easier. In this paper we establish this mechanism rigorously only for fixed-eigenbasis eigenvalue learning and provide the pathwise ESD formulation and experimental evidence for evolving-eigenfunction settings. A theorem for adaptive learning with evolving

eigenfunctions remains open.

Although the ESD is a population quantity and is not directly used as a training input, practical surrogates are possible. In the sequence model, $d^\dagger$ is the PC truncation level at which the squared tail bias first becomes comparable to the variance. Thus one may estimate an ESD surrogate by selecting the truncation level through data-driven risk estimation or penalized projection or model-selection criteria, in the spirit of Birgé & Massart (2001). Relatedly, designing training procedures that target ESD reduction is another future direction.

In summary, the ESD framework provides a novel view of generalization that connects classical kernel methods with modern adaptive learning. We expect to relate this framework to learned representations in neural networks to explain their superior generalization performance.

## Acknowledgements

The authors thank the reviewers for their careful reading and constructive comments, which helped improve the clarity and quality of the paper. D.H. is grateful to Wei-Liem Loh and Weihao Lu for their helpful discussions. D.H. is supported by Singapore MOE AcRF Tier-1 Grant A-8004149-00-00. Q.L. is supported by the National Natural Science Foundation of China (92370122, 11971257) and Tsinghua University Dushi Program (20251080059).

## Impact Statement

This paper presents work whose goal is to advance the field of machine learning. There are many potential societal consequences of our work, none of which we feel must be specifically highlighted here.

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

# A. Related Work

This appendix provides a comprehensive discussion of related work. We first distinguish ESD from balanced spectral cutoffs and early stopping in Appendix A.1. We then compare ESD with signal-agnostic complexity measures in Appendix A.2 and related work on target–kernel alignment in Appendix A.3. Furthermore, we discuss other estimator-guided measures in Appendix A.4. Finally, we briefly review principal component regression in Appendix A.5.

## A.1. Balanced Spectral Cutoffs and Early Stopping

The bias–variance crossing underlying ESD has classical antecedents in spectral cutoff and early-stopping theory. In the direct sequence model, and analogously for truncated-SVD estimators after accounting for the singular-value weights in inverse problems, the ESD cutoff coincides with the oracle index balancing tail bias and estimation variance. This algebraic connection is not the novelty of ESD, but rather its calibration point. Blanchard et al. (2018a;b) study oracle and data-driven stopping for fixed statistical inverse problems with a prescribed singular system, including sequential information constraints for truncated SVD, residual stopping, and oracle adaptation for spectral regularization methods. Their question is whether the performance of a stopping rule can match, or nearly match, that of the best oracle index for one fixed inverse operator under computational and information constraints. ESD addresses a different question: after fixing a target–kernel pair and a noise level, it turns the same bias–variance crossing into an alignment-sensitive measure that quantifies the statistical difficulty of estimation.

This distinction is essential for learned kernels. Our theory freezes the spectrum at each point of a learning trajectory and evaluates the resulting target–kernel pair; it does not claim a general post-selection oracle inequality for arbitrary data-dependent kernels. Through the associated span profile and ESD-bounded classes, the paper proves minimax risk characterizations of the form $\sigma^2 K$ and thereby compares the statistical difficulty induced by different spectra for the same target. The lower-bound questions are also different: the early-stopping literature studies sequential-information and stopping-rule limitations for a fixed inverse problem, whereas our lower bounds are minimax statements over ESD-bounded classes. Subsequent work in the inverse-problem and early-stopping line further develops fixed-operator stopping rules and oracle inequalities (Stankewitz, 2020; 2024; Hucker & Reiß, 2025); these results are complementary to the target–kernel alignment pursued in the current work.

## A.2. Signal-Agnostic Dimensions

Other classical complexity measures describe intrinsic properties of the kernel space rather than the alignment between the kernel and a particular target. Zhang (2005) introduces the effective dimension to quantify the complexity of a regularized method. For ridge regularization in Equation (5), the effective dimension is

$$d_{\text{eff}}(\nu) = \sum_j \frac{\lambda_j}{\lambda_j + \nu},$$

as in Zhang (2005, Proposition A.1). This quantity depends only on the spectrum $\boldsymbol{\lambda}$ and the regularization parameter $\nu$, not on the signal $\boldsymbol{\theta}^*$ or the noise level $\sigma^2$. It therefore cannot measure signal–spectrum alignment. Moreover, as a function of $\nu$, it is not itself tied to the ESD-bounded minimax classes considered here.

In linear regression, Bartlett et al. (2020) analyze the minimum-norm interpolator through the effective rank

$$r_k = \frac{\sum_{i>k} \lambda_{\pi_i}}{\lambda_{\pi_{k+1}}},$$

and $k^* = \min\{k \geq 0 : r_k(\Sigma) \geq bn\}$ for a constant $b$. Translating the sample-size scaling $n^{-1}$ into the sequence-model noise level $\sigma^2$, this condition is analogous to $\sigma^2 r_k(\Sigma) \gtrsim 1$, and the resulting risk bounds involve a leading contribution comparable to $\sigma^2 k^*$. This index resembles ESD in that it depends on the spectrum and the noise scale. However, $k^*$ does not involve the signal, and it is tailored to the minimum-norm estimator rather than to minimax risk over a signal class.

Both $d_{\text{eff}}(\nu)$ and $k^*$ are signal-agnostic: they depend on the spectrum $\boldsymbol{\lambda}$ and either $\nu$ or $\sigma^2$, but not on the allocation of the target over the eigenfunctions. If adaptive training improves alignment by reordering eigenfunctions to better match the signal while preserving the multiset of eigenvalues, then both $d_{\text{eff}}(\nu)$ and $k^*$ remain unchanged. In contrast, $d^\dagger(\sigma^2; \boldsymbol{\theta}^*, \boldsymbol{\lambda})$ changes with the target allocation and can decrease as signal-kernel alignment improves. This is why ESD can explain the generalization benefits of adaptive kernel learning in settings where signal-agnostic quantities cannot.

**A.3. Target-Kernel Alignment**

Classical kernel-target alignment measures similarity between the kernel matrix and labels, often through centered or cosine-type alignment, and has been used for kernel selection and generalization bounds (Cristianini et al., 2001; Cortes et al., 2012; Kornblith et al., 2019). However, the bounds provided by these measures are typically too loose to explain fast rates in adaptive kernel methods. More recently, Canatar et al. (2021) define the "cumulative power distribution" $C(k)$ as

$$C(k) = \frac{\sum_{k' \leq k} \lambda_{\pi_{k'}} (\theta^*_{\pi_{k'}})^2}{\sum_{k'} \lambda_{\pi_{k'}} (\theta^*_{\pi_{k'}})^2}, k = 1, 2, \ldots,$$

and they numerically demonstrate that $C(k)$ provides "a measure of the compatibility between the kernel and the target." This measure provides a useful global summaries of kernel-target alignment, but it does not encode the noise-indexed bias–variance crossing that determines the ESD.

Recently, Barzilai & Shamir (2024) extended benign-overfitting analyses (Bartlett et al., 2020; Tsigler & Bartlett, 2023) to kernel ridge regression, which may encounter saturation effects that prevent optimal rates for overly smooth target functions. Wang et al. (2024) analyze target-kernel alignment for prescribed kernels and a broad class of kernel-based methods, but their analysis relies on classical assumptions such as source conditions (see Assumptions 3.2 and 3.4 therein). ESD takes a different object as central: for an arbitrary target allocation, spectrum, and noise level, it defines the noise-indexed bias–variance crossing and the resulting span profile. This distinction is essential for learned kernels. Once training changes the spectrum or feature basis, ESD can compare the same target before and after learning, whereas fixed-kernel alignment theory describes the behavior of a method under a prescribed kernel.

**A.4. Estimator-Guided Measures**

The ESD equals the truncation point for the PC estimator where the squared tail bias first becomes comparable to the estimation variance. It may therefore seem natural to define alternative complexity indices by calibrating the bias and variance of other estimators. We focus on ESD for two reasons. First, Theorem 3.3 shows that, for the ESD-bounded classes considered in this paper, the minimax risk is calibrated by $\sigma^2 K$ up to constant factors. Thus, for these classes, ESD is the canonical alignment-sensitive dimension.

Second, an estimator-guided index may reflect the defects of the chosen estimator rather than the intrinsic difficulty of the target–kernel pair at the given noise level. Ridge regression gives a concrete example. Because of the *saturation effect*, a ridge-guided index calibrated to ridge's bias–variance balance can fail to describe the optimal estimation risk even in classical settings with source conditions. Appendix I formalizes this point.

**A.5. Principal Component Regression**

As discussed in Section 3.1, the PC estimator serves as a motivating example for the concepts of ESD and span profile due to its clear illustration of bias-variance trade-offs. However, the ESD and span profile are designed to characterize statistical difficulty determined by the target–kernel alignment at a given noise level, and their definitions do not rely on the PC estimator. Nonetheless, the analysis of PC estimators, particularly in high-dimensional linear regression, has been an active area of recent research. Below, we briefly summarize some relevant contributions to provide context.

A.5.1. PROPORTIONAL ASYMPTOTICS

Several studies analyze Principal Component Regression (PCR) in the proportional asymptotic setting where the dimension $p$ and sample size $n$ grow with $p/n \to \gamma$. In the regime $\gamma \leq 1$, Xu & Hsu (2019) study the limiting risk of PCR for Gaussian designs with diagonal covariance. They assume polynomially decaying eigenvalues or a convergent empirical spectrum, together with an isotropic prior, and they reveal a "double-descent" risk curve. Wu & Xu (2020) study generalized weighted ridge in overparameterized linear regression and, as part of their analysis, characterize double descent for PCR under an anisotropic prior satisfying $\mathbb{E}\beta_* \beta_*^\top = \Sigma_\beta$. They also derive an exact risk expression and demonstrate how "misalignment" between $\Sigma_x = \text{Cov}(x)$ and $\Sigma_\beta$ affects risk; here "alignment" refers to concordance between the orderings of their eigenvalues. Both studies assume knowledge of the eigenvectors of the population covariance matrix $\Sigma_x$ to construct the *oracle PCR*. Gedon et al. (2024) analyze the limiting risk of PCR under a latent factor model and explore the effect of distribution shift. Green & Romanov (2025) derive the exact limits of estimation risk, in-sample prediction risk, and out-of-sample prediction risk of PCR under the assumption that both the empirical spectral distribution and the distribution

of the true signal's mass over the eigenspaces of $\Sigma_x$ converge weakly.

### A.5.2. NON-ASYMPTOTIC ANALYSIS

Complementary research develops non-asymptotic guarantees. Agarwal et al. (2019) derive finite-sample upper bounds on prediction error using $\|\beta^*\|_1^2$ and the rank of the design matrix under latent factor models, and they explore the robustness of PCR to noise and missing values in the observed covariates. Bing et al. (2021) consider PCR with an adaptively selected number of components under latent factor models and provide alternative finite-sample risk bounds using $\|\beta^*\|_2^2$. Huang et al. (2022) derive non-asymptotic risk bounds for PCR in more general settings by analyzing the alignment between population and empirical principal components. Hucker & Wahl (2023) derive high-probability bounds for the conditional prediction risk of PCR in high dimensions and compare empirical PCR with the corresponding population-PC oracle under effective-rank conditions.

## B. Extension to Correlated Noise and Fixed-Design Linear Model

This section extends the concepts of ESD and span profile, developed in Section 3 for the sequence model, to the setting of fixed-design linear regression. In addition, we demonstrate how the minimax optimal prediction risk in this setting can be characterized using the span profile, paralleling the analysis in Section 4.

### B.1. Correlated Noise

Before turning to fixed-design linear regression, we first record a simple correlated-noise sequence model that illustrates how a spectral operator can arise from the geometry of the problem.

Consider
$$\mathbf{Z} = \boldsymbol{\theta}^* + \boldsymbol{\xi}, \qquad \boldsymbol{\xi} \sim \mathcal{N}_d(0, \sigma^2 \boldsymbol{\Sigma}_\xi), \tag{13}$$

where $\boldsymbol{\Sigma}_\xi \in \mathbb{R}^{d \times d}$ is known, symmetric, and positive definite. We measure error by the squared Mahalanobis loss
$$L(\widehat{\boldsymbol{\theta}}, \boldsymbol{\theta}^*) = (\widehat{\boldsymbol{\theta}} - \boldsymbol{\theta}^*)^\top \boldsymbol{\Sigma}_\xi^{-1} (\widehat{\boldsymbol{\theta}} - \boldsymbol{\theta}^*).$$

Suppose the covariance matrix $\boldsymbol{\Sigma}_\xi$ admits an eigenvalue decomposition as follows:
$$\boldsymbol{\Sigma}_\xi = \mathbf{Q} \operatorname{diag}(\rho_1, \ldots, \rho_d) \mathbf{Q}^\top, \qquad 0 < \rho_1 \le \rho_2 \le \cdots \le \rho_d.$$

Define $\lambda_j := \rho_j^{-1}$, $\boldsymbol{\lambda} = (\lambda_j)_{j=1}^d$, and $\boldsymbol{\Lambda} := \operatorname{diag}(\lambda_1, \ldots, \lambda_d)$. Then the relevant spectral operator is the precision matrix
$$\boldsymbol{\Sigma}_\xi^{-1} = \mathbf{Q} \boldsymbol{\Lambda} \mathbf{Q}^\top,$$

so larger eigenvalues correspond to lower-noise directions.

To build a connection to the standard sequence model in Equation (4), define the whitened spectral coordinates
$$\bar{\mathbf{Z}} = \boldsymbol{\Lambda}^{1/2} \mathbf{Q}^\top \mathbf{Z}, \qquad \bar{\boldsymbol{\theta}}^* = \boldsymbol{\Lambda}^{1/2} \mathbf{Q}^\top \boldsymbol{\theta}^*.$$

Then the model reduces to
$$\bar{\mathbf{Z}} = \bar{\boldsymbol{\theta}}^* + \bar{\boldsymbol{\xi}}, \qquad \bar{\boldsymbol{\xi}} \sim \mathcal{N}_d(0, \sigma^2 \mathbf{I}_d).$$

For any estimator $\widehat{\boldsymbol{\theta}}$ for $\boldsymbol{\theta}^*$, the transformed estimator is $\widehat{\bar{\boldsymbol{\theta}}} = \boldsymbol{\Lambda}^{1/2} \mathbf{Q}^\top \widehat{\boldsymbol{\theta}}$, whose loss is
$$\|\widehat{\bar{\boldsymbol{\theta}}} - \bar{\boldsymbol{\theta}}^*\|_2^2 = (\widehat{\boldsymbol{\theta}} - \boldsymbol{\theta}^*)^\top \boldsymbol{\Sigma}_\xi^{-1} (\widehat{\boldsymbol{\theta}} - \boldsymbol{\theta}^*).$$

Therefore, the correlated-noise problem under Mahalanobis loss is exactly the standard sequence model in Equation (4), with signal $\bar{\boldsymbol{\theta}}^*$ and spectrum $\boldsymbol{\lambda}$. It is then natural to define its ESD by
$$d^\dagger(\tau; \boldsymbol{\theta}^*, \boldsymbol{\Sigma}_\xi) := d^\dagger(\tau; \bar{\boldsymbol{\theta}}^*, \boldsymbol{\lambda}).$$

The same reduction also identifies the corresponding spectral estimators in the original coordinates. For a spectral filter $\psi_\nu$, define the spectral estimator for the transformed model as
$$\widehat{\bar{\theta}}_{\psi,\nu,j} = (1 - \psi_\nu(\lambda_j)) \bar{Z}_j.$$

The associated estimator before transformation is given by

$$\widehat{\boldsymbol{\theta}}_{\psi,\nu} = \mathbf{Q}\big(\mathbf{I}_d - \psi_\nu(\boldsymbol{\Lambda})\big)\mathbf{Q}^\top\mathbf{Z}.$$

For example, the ridge estimator is

$$\widehat{\boldsymbol{\theta}}_{\mathrm{R},\nu} = \mathbf{Q}\operatorname{diag}\left\{\frac{\lambda_j}{\lambda_j + \nu}\right\}\mathbf{Q}^\top\mathbf{Z} = (\mathbf{I}_d + \nu\boldsymbol{\Sigma}_\xi)^{-1}\mathbf{Z}.$$

Such a ridge estimator arises from imposing an $\ell_2$ penalty or from Bayesian inference: if $\mathbf{Z} \mid \boldsymbol{\theta} \sim \mathcal{N}_d(\boldsymbol{\theta}, \sigma^2\boldsymbol{\Sigma}_\xi)$ and $\boldsymbol{\theta} \sim \mathcal{N}_d(0, \alpha\mathbf{I}_d)$, then the posterior mean is $(\mathbf{I}_d + \nu\boldsymbol{\Sigma}_\xi)^{-1}\mathbf{Z}$ with $\nu = \frac{\sigma^2}{\alpha}$.

The PC estimator keeps precisely the low-noise eigendirections:

$$\widehat{\boldsymbol{\theta}}_{\mathrm{PC},\nu} = \mathbf{Q}\operatorname{diag}\left\{\mathbf{1}_{\{\lambda_j \geq \nu\}}\right\}\mathbf{Q}^\top\mathbf{Z} = \mathbf{Q}\operatorname{diag}\left\{\mathbf{1}_{\{\rho_j \leq \nu^{-1}\}}\right\}\mathbf{Q}^\top\mathbf{Z}.$$

In the fixed-design linear model below, our analysis follows the same principle and the relevant spectrum is given by the squared singular values of the design matrix.

## B.2. Linear Model

Consider the following fixed-design linear regression model:

$$\mathbf{Y} = \mathbf{X}\boldsymbol{\beta}^* + \boldsymbol{\epsilon}, \tag{14}$$

where $\mathbf{Y} \in \mathbb{R}^n$ is the vector of observations, $\mathbf{X} \in \mathbb{R}^{n \times p}$ is the fixed-design matrix of rank $r \leq \min(n, p)$, $\boldsymbol{\beta}^* \in \mathbb{R}^p$ is the unknown vector of true coefficients, and $\boldsymbol{\epsilon} \in \mathbb{R}^n$ is the noise vector. We assume the components of $\boldsymbol{\epsilon}$ are uncorrelated with mean zero and variance $\sigma_0^2$. For this model, we consider the loss (in-sample prediction error) $\mathcal{L}(\widehat{\boldsymbol{\beta}}; \boldsymbol{\beta}^*) = \frac{1}{n}\|\mathbf{X}\big(\widehat{\boldsymbol{\beta}} - \boldsymbol{\beta}^*\big)\|^2$ and risk $\mathcal{R}(\widehat{\boldsymbol{\beta}}; \boldsymbol{\beta}^*) = \mathbb{E}\mathcal{L}(\widehat{\boldsymbol{\beta}}; \boldsymbol{\beta}^*)$. We treat random-design linear regression as a special case of RKHS regression and discuss it in Appendix C.

To connect this model to the sequence model analysis presented earlier, we utilize the Singular Value Decomposition (SVD) of the design matrix $\mathbf{X}$ as follows:

$$\frac{1}{\sqrt{n}}\mathbf{X} = \mathbf{U}\mathbf{S}\mathbf{V}^\top, \tag{15}$$

where $\mathbf{U} \in \mathbb{R}^{n \times n}$ and $\mathbf{V} \in \mathbb{R}^{p \times p}$ are orthogonal matrices, and $\mathbf{S} \in \mathbb{R}^{n \times p}$ is a rectangular diagonal matrix with non-negative singular values $s_1 \geq s_2 \geq \cdots \geq s_r > 0$ on its diagonal, and $s_j = 0$ for $j > r$.

For any matrix $\mathbf{A}$ and subsets $R$ and $T$, we write $\mathbf{A}_{:,R}$ for the submatrix formed by the columns of $\mathbf{A}$ with indices in $R$, and write $\mathbf{A}_{T,:}$ for the submatrix formed by the rows of $\mathbf{A}$ with indices in $T$.

Multiplying the model in Equation (14) by $\frac{1}{\sqrt{n}}\mathbf{U}_{:,[r]}^\top$, we obtain an $r$-dimensional transformed model:

$$\mathbf{Z} = \boldsymbol{\theta}^* + \boldsymbol{\xi}, \tag{16}$$

where we have defined

$$\mathbf{Z} = \frac{1}{\sqrt{n}}\mathbf{U}_{:,[r]}^\top\mathbf{Y},$$

$$\boldsymbol{\theta}^* = \frac{1}{\sqrt{n}}\mathbf{U}_{:,[r]}^\top\mathbf{X}\boldsymbol{\beta}^* = \mathbf{S}_{[r],:}\mathbf{V}^\top\boldsymbol{\beta}^*,$$

$$\text{and } \boldsymbol{\xi} = \frac{1}{\sqrt{n}}\mathbf{U}_{:,[r]}^\top\boldsymbol{\epsilon}.$$

Since $\mathbf{U}$ is orthogonal, the transformed noise vector $\boldsymbol{\xi}$ still has uncorrelated components with mean zero and variance $\sigma^2 := \sigma_0^2/n$.

The transformed model in Equation (16) is analogous to the sequence model in Equation (4), where the signal is $\boldsymbol{\theta}^*$ and the noise variance for each component is $\sigma^2$. The "spectrum" relevant to this problem is derived from the singular values of $\mathbf{X}$.

Specifically, we define the eigenvalues as $\lambda_j = s_j^2$ for $j = 1, \ldots, r$, and $\lambda_j = 0$ for $j > r$. Let $\{\pi_k\}_{k=1}^r$ denote the indices corresponding to the eigenvalues sorted in descending order, $\lambda_{\pi_1} \geq \lambda_{\pi_2} \geq \cdots \geq \lambda_{\pi_r} > 0$.

For any estimator $\widehat{\boldsymbol{\beta}}$ for the linear model in Equation (14), define $\widehat{\boldsymbol{\theta}} = \mathbf{S}_{[r],.}\mathbf{V}^\top\widehat{\boldsymbol{\beta}}$. We can then write the prediction risk as $n^{-1}\mathbb{E}\|\mathbf{X}\widehat{\boldsymbol{\beta}} - \mathbf{X}\boldsymbol{\beta}^*\|^2 = \mathbb{E}\|\mathbf{U}\mathbf{S}\mathbf{V}^\top\widehat{\boldsymbol{\beta}} - \mathbf{U}\mathbf{S}\mathbf{V}^\top\boldsymbol{\beta}^*\|^2 = \mathbb{E}\|\widehat{\boldsymbol{\theta}} - \boldsymbol{\theta}^*\|^2$. Conversely, given an estimator $\widehat{\boldsymbol{\theta}}$ for the sequence model in Equation (16), we can define $\widetilde{\boldsymbol{\beta}} = \mathbf{V}\mathbf{S}^\dagger\widehat{\boldsymbol{\theta}}$ where $\mathbf{S}^\dagger \in \mathbb{R}^{p \times r}$ is a diagonal matrix whose diagonal elements are $\{1/s_j\}_{j \in [r]}$. It is easy to check that $\mathbf{S}_{[r],.}\mathbf{V}^\top\widetilde{\boldsymbol{\beta}} = \widehat{\boldsymbol{\theta}}$. Therefore, we establish an equivalence between the model in Equation (16) and the model in Equation (14).

The usual ridge regression estimator for the linear model in Equation (14) is given by

$$\widehat{\boldsymbol{\beta}}_\nu = \left(\frac{1}{n}\mathbf{X}^\top\mathbf{X} + \nu\mathbf{I}_p\right)^{-1}\left(\frac{1}{n}\mathbf{X}^\top\mathbf{Y}\right),$$

which transforms into

$$\widehat{\boldsymbol{\theta}}_\nu = \mathbf{S}_{[r],.}\mathbf{V}^\top\widehat{\boldsymbol{\beta}}_\nu = \left(\mathbf{I}_r - \psi_\nu(\mathrm{Diag}\,(\lambda_1, \ldots, \lambda_r))\right)\mathbf{Z}, \qquad \text{where } \psi_\nu(\lambda) = \frac{1}{\lambda/\nu + 1}.$$

In the above expression, we have used the identity that $\mathbf{S}_{[r],.}\mathbf{S}_{[r],.}^\top = \mathrm{Diag}\,(s_1^2, \ldots, s_r^2)$ and $\psi_\nu(\cdot)$ is applied element-wise. If we replace $\psi_\nu(\lambda)$ by other functions as discussed in Section 3, we recover other spectral methods.

### B.3. ESD for Linear Models

We can now adapt the definitions from Section 3 to linear models.

**Definition B.1** (ESD for Linear Regression). Suppose the SVD of the design matrix $\mathbf{X}$ is given in Equation (15). The Effective Span Dimension (ESD) of $\boldsymbol{\beta}^*$ with respect to the design $\mathbf{X}$ and the per component variance $\sigma_0^2/n$ is defined as

$$d^\dagger = d^\dagger(\sigma_0^2/n; \boldsymbol{\beta}^*, \mathbf{X}) = \min\{k \in [r] : \mathbf{H}_{\boldsymbol{\theta}^*,\boldsymbol{\lambda}}(k) \leq \sigma_0^2/n\},$$

where $\boldsymbol{\theta}^* = \mathbf{S}_{[r],.}\mathbf{V}^\top\boldsymbol{\beta}^*$ and $\lambda_j = s_j^2$.

The Principal Component Regression (PCR) estimator for $\boldsymbol{\beta}^*$ corresponds to the Principal Component (PC) estimator for the transformed model in Equation (16). Specifically, for any $k \in [r]$, define $\widehat{\boldsymbol{\beta}}^{\mathrm{PC},k} = \frac{1}{\sqrt{n}}\mathbf{V}\mathbf{S}_k^\dagger\mathbf{U}_{:,[r]}^\top\mathbf{Y}$, where $\mathbf{S}_k^\dagger \in \mathbb{R}^{p \times r}$ is a diagonal matrix whose diagonal elements are $\{\frac{1}{s_j}\mathbf{1}_{\{s_j \geq s_{\pi_k}\}}\}$. In the $\mathbf{Z}$ space, this means

$$\hat{\theta}_j^{\mathrm{PC},k} = \mathbf{1}\{s_j \geq s_{\pi_k}\}Z_j, \;\; j \in [r]. \tag{17}$$

By analogy with Proposition 3.2, the minimal prediction risk achievable by PCR over $k$ is characterized by the ESD.

**Proposition B.2** (Optimal PCR Prediction Risk). *Let $\widehat{\boldsymbol{\beta}}^{\mathrm{PC},k}$ be the PCR estimator using the first $k$ principal components. Let $\mathcal{R}_*^{\mathrm{PC}}$ be the minimal possible prediction risk over $k \in [r]$, i.e., $\mathcal{R}_*^{\mathrm{PC}} = \min_{k \in [r]} \mathcal{R}(\widehat{\boldsymbol{\beta}}^{\mathrm{PC},k}; \boldsymbol{\beta}^*)$. It holds that*

$$(d^\dagger - 1)\sigma_0^2/n \leq \mathcal{R}_*^{\mathrm{PC}} \leq 2d^\dagger\sigma_0^2/n,$$

*where $d^\dagger = d^\dagger(\sigma_0^2/n; \boldsymbol{\beta}^*, \mathbf{X})$ is the ESD defined in Definition B.1.*

Proposition B.2 directly follows from Proposition 3.2 and its proof is omitted. This result shows that the optimal prediction risk for PCR is determined by the ESD $d^\dagger$, which measures the effective number of principal components needed to balance the bias-variance trade-off.

We can further extend the minimax analysis from Theorem 3.3. Let $K$ be a quota on ESD. Define a class of coefficient vectors based on this quota:

$$\mathcal{B}_K^{(n)} = \left\{\boldsymbol{\beta}^* \in \mathbb{R}^p : d^\dagger(\sigma_0^2/n; \boldsymbol{\beta}^*, \mathbf{X}) \leq K\right\}. \tag{18}$$

This class contains signals whose ESD relative to the design $\mathbf{X}$ is controlled by $K$. We can establish the minimax optimal rate for prediction over this class.

**Theorem B.3** (Minimax Prediction Risk for Linear Regression). *Suppose $K \leq r$. For the linear model in Equation* (14) *with noise variance $\sigma_0^2$, the minimax prediction risk over the class $\mathcal{B}_K^{(n)}$ defined in Equation* (18) *satisfies:*

$$\inf_{\widehat{\boldsymbol{\beta}}} \sup_{\boldsymbol{\beta}^* \in \mathcal{B}_K^{(n)}} \mathcal{R}(\widehat{\boldsymbol{\beta}}; \boldsymbol{\beta}^*) \asymp \sigma_0^2 \frac{K}{n}.$$

The proof of Theorem B.3 is essentially the same as that of Theorem 3.3 and is omitted.

Through this extension, the span profile framework connects the optimal prediction performance in fixed-design linear regression to the alignment between the signal structure (transformed via the design matrix) and the spectrum derived from the design matrix's singular values.

### B.4. Numerical Illustration

This section illustrates the ESD in fixed-design linear models in two examples. Throughout, we fix the noise variance at $\sigma_0^2 = 1$, the sample size at $n = 300$, and the dimension at $p = 400$.

**Experimental setup**   The baseline design matrix $\mathbf{X}_0$ is randomly generated with covariance matrix $\Sigma = \mathrm{Diag}\{\lambda_j\}_{j \in [p]}$ and then held fixed. We consider two cases:

1. *Geometric decay spectrum and polynomial decay signal*: $\lambda_j \propto 0.95^j$ and $\beta_j^* = j^{-0.2}$;

2. *Logarithmic decay spectrum and signal*: $\lambda_j = 1/\log(j+1)$ with $\beta_j^* = 1/\log(j+1)$.

The response is generated from $\boldsymbol{Y} = \mathbf{X}_0 \boldsymbol{\beta}^* + \boldsymbol{\epsilon}$ with random noise $\boldsymbol{\epsilon}$.

We are interested in the ESD and the minimum risk for different transformations of the design matrix. For this purpose, we introduce a class of non-orthogonal column transformations indexed by $\alpha \geq 0$ as follows:

$$\mathbf{A}(\alpha) = \mathrm{Diag}\big(\exp\{\alpha\, t_j\}\big), \quad t_j = (j-1)/(p-1) - 1/2, \quad j \in [p].$$

The transformed design is $\mathbf{X}(\alpha) = \mathbf{X}_0 \mathbf{A}(\alpha)$, and the correspondingly transformed coefficient vector is $\boldsymbol{\beta}(\alpha) = \mathbf{A}(\alpha)^{-1} \boldsymbol{\beta}^*$. These transformations will change the ordering of the spectrum, from well-aligned to misaligned. We are interested in the following at each $\alpha$:

- Effective Span Dimension: $d^\dagger(\alpha) = d^\dagger\big(\sigma_0^2/n;\, \boldsymbol{\beta}(\alpha), \mathbf{X}(\alpha)\big)$;

- Minimal PCR risk: $\mathcal{R}_*(\alpha) = \min_k \mathbb{E}\big[n^{-1}\|\mathbf{X}(\alpha)\widehat{\boldsymbol{\beta}}_k - \mathbf{X}(\alpha)\boldsymbol{\beta}(\alpha)\|^2\big]$, where $\widehat{\boldsymbol{\beta}}_k$ is the $k$-component principal-component estimator based on $(\boldsymbol{Y}, \mathbf{X}(\alpha))$.

Figure 3 plots $d^\dagger(\alpha)$ (solid) and the rescaled oracle risk defined as $n\mathcal{R}_*(\alpha)/\sigma_0^2$ (dashed) against $\alpha$. The two curves track each other up to constant factors, which empirically verifies the bound in Proposition B.2 that $\mathcal{R}_*(\alpha) \asymp \frac{\sigma_0^2}{n} d^\dagger(\alpha)$. As $\alpha$ grows, the diagonal stretch $\mathbf{A}(\alpha)$ shifts signal energy towards directions that carry smaller singular values. This raises $d^\dagger$, and consequently, the achievable risk.

This experiment illustrates that ESD, rather than raw spectral decay, is the pivotal measure governing learnability.

## C. Extension to RKHS Regression

This section extends the ESD framework developed in Section 3 to the setting of RKHS regression. Since our goal is to develop a population-level complexity measure, the eigensystem used below is the Mercer eigensystem of the population kernel. We do not analyze the statistical estimation of empirical eigenvalues or empirical eigenfunctions; a thorough analysis in this direction is left for future studies.

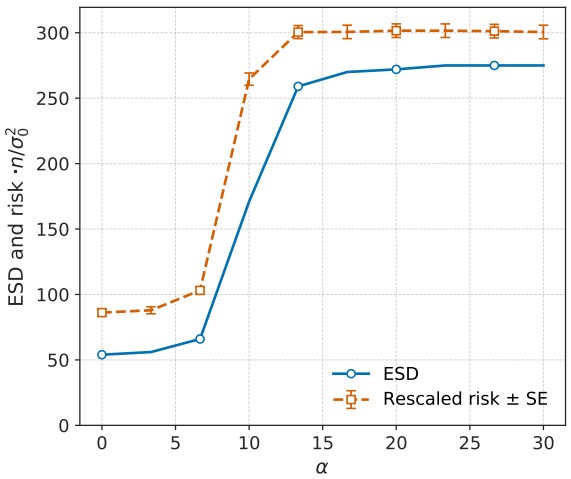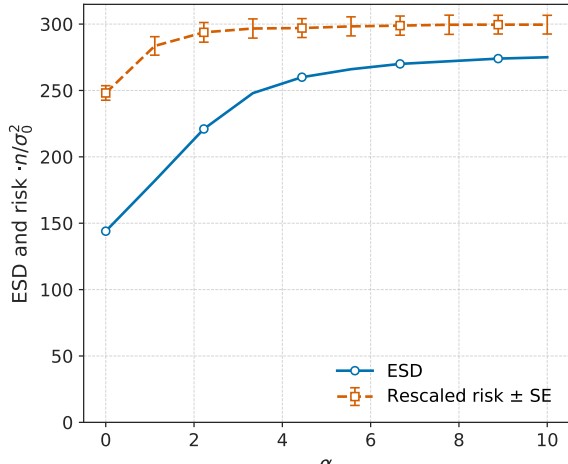

*Figure 3.* **Oracle PCR risk versus Effective Span Dimension** for (a) geometric eigen-decay and (b) logarithmic eigen-decay. The dashed line plots Risk $\times n/\sigma_0^2$; the solid line is $d^\dagger(\alpha)$. The risk is computed based on 20 replications and the error bars represent standard errors.

### C.1. RKHS Regression

We recall the standard random-design kernel regression model from Section 2:

$$y_i = f^*(x_i) + \epsilon_i, \quad \epsilon_i \text{ are i.i.d. with } \mathbb{E}[\epsilon_i] = 0, \ \mathrm{Var}(\epsilon_i) = \sigma_0^2, \quad i = 1, \ldots, n, \tag{19}$$

where $x_i \overset{\text{i.i.d.}}{\sim} \mu$, and $f^* \in L^2(\mathcal{X}, \mu)$ is the target function. We use a kernel $\mathbf{k}(\cdot, \cdot)$ with spectral decomposition $\mathbf{k}(x, x') = \sum_{j=1}^{\infty} \lambda_j \phi_j(x) \phi_j(x')$, where $\{\lambda_j\}_{j \geq 1}$ are the positive eigenvalues and $\{\phi_j\}_{j \geq 1} \subset \mathcal{H}$ is an orthonormal system in $L^2(\mathcal{X}, \mu)$. Here $\boldsymbol{\lambda} = \{\lambda_j\}_{j \geq 1}$ is not necessarily sorted. For simplicity, we assume there are no ties among the eigenvalues and let $\pi$ be the permutation that sorts them in descending order, so that $\lambda_{\pi_1} > \lambda_{\pi_2} > \cdots > \lambda_{\pi_k} > \ldots$.

Let $\mathcal{S}_{\mathbf{k}}$ denote the closed span of $\{\phi_j\}_{j=1}^{\infty}$ in $L^2(\mathcal{X}, \mu)$. To present a clear theory without unnecessary technical complications, we first focus on the case where $f^* \in \mathcal{S}_{\mathbf{k}}$. The coefficients of $f^*$ in this basis are

$$\theta_j^* = \langle f^*, \phi_j \rangle_{L^2(\mu)}, \qquad j \geq 1.$$

An estimator $\hat{f}$ has prediction risk

$$\mathcal{R}(\hat{f}; f^*) = \mathbb{E}\|\hat{f} - f^*\|_{L^2(\mu)}^2.$$

If we construct $\hat{f} \in \mathcal{S}_{\mathbf{k}}$ with coefficient

$$\hat{\theta}_j = \langle \hat{f}, \phi_j \rangle_{L^2(\mu)}, \qquad j \geq 1,$$

then Parseval's identity gives

$$\mathcal{R}(\hat{f}; f^*) = \mathbb{E}\|\hat{f} - f^*\|_{L^2(\mu)}^2 = \mathbb{E}\Big[\sum_{j=1}^{\infty} \Big(\hat{\theta}_j - \theta_j^*\Big)^2\Big] = \sum_{j=1}^{\infty} \mathbb{E}\Big(\hat{\theta}_j - \theta_j^*\Big)^2. \tag{20}$$

The expression in Equation (20) suggests that we can equivalently estimate each $\theta_j^*$ separately using the transformed observation

$$z_j = \frac{1}{n} \sum_{i=1}^{n} y_i \phi_j(x_i), \qquad j \geq 1, \tag{21}$$

as introduced in Equation (2).

We remind the reader that in RKHS, we use subscript $i$ for samples and subscript $j$ for eigen-coordinates. The subscript $j$ aligns with the use of notation in the sequence model, where we use indices $j$ to denote coordinates.

**Inflated variance in Equation (3).** Before we introduce the definition of ESD for the RKHS model, we revisit the connection to the sequence model discussed in Equation (3).

For each $j \geq 1$, we can write the transformed observation as

$$
\begin{aligned}
z_j &= \frac{1}{n} \sum_{i=1}^{n} y_i \, \phi_j(x_i) \\
&= \frac{1}{n} \sum_{i=1}^{n} \left( f^*(x_i) + \epsilon_i \right) \phi_j(x_i) \\
&= \frac{1}{n} \sum_{i=1}^{n} \left( \sum_{k \geq 1} \theta_k^* \phi_k(x_i) \right) \phi_j(x_i) \; + \; \frac{1}{n} \sum_{i=1}^{n} \epsilon_i \, \phi_j(x_i) \\
&= \sum_{k \geq 1} \theta_k^* \underbrace{\left( \frac{1}{n} \sum_{i=1}^{n} \phi_k(x_i)\phi_j(x_i) \right)}_{:=G_{kj}} \; + \; \underbrace{\frac{1}{n} \sum_{i=1}^{n} \epsilon_i \, \phi_j(x_i)}_{=\xi_j} \\
&= \sum_{k \geq 1} G_{kj} \, \theta_k^* \; + \; \xi_j,
\end{aligned}
$$

where $G_{kj} := \frac{1}{n} \sum_{i=1}^{n} \phi_k(x_i)\phi_j(x_i)$ are entries of the feature covariance matrix. For $\xi_j$, we have $\mathbb{E}(\xi_j \mid \{x_i\}_{i \in [n]}) = 0$ and $\mathbb{E}(\xi_j \xi_k \mid \{x_i\}_{i \in [n]}) = n^{-1} \sigma_0^2 G_{jk}$.

Since $x_i \overset{\text{i.i.d.}}{\sim} \mu$ and $\{\phi_j\}$ are orthonormal in $L^2(\mathcal{X}, \mu)$, we have $\mathbb{E}[G_{kj}] = \mathbf{1}_{\{k=j\}}$. Hence $\mathbb{E}[z_j] = \theta_j^*$.

We may further decompose $z_j$ as follows

$$
z_j - \theta_j^* = (G_{jj} - 1)\theta_j^* + \left( \sum_{k \geq 1, k \neq j} G_{kj} \theta_k^* \right) + \xi_j = \Delta_j + \xi_j,
$$

where we have defined $\Delta_j = (G_{jj} - 1)\theta_j^* + \left( \sum_{k \geq 1, k \neq j} G_{kj} \theta_k^* \right)$. This term does not appear in the sequence model; its randomness arises solely from the random covariates $x_i$. As $n \to \infty$, this term vanishes because $G_{jj} = n^{-1} \sum_i \phi_j(x_i)^2 \to 1$ and $G_{kj} = n^{-1} \sum_{i=1}^{n} \phi_j(x_i) \phi_k(x_i) \to 0$ for $k \neq j$. Furthermore, since $\mathbb{E}(\xi_j \mid \{x_i\}_{i \in [n]}) = 0$, we have

$$
\mathbb{E}(\Delta_j \xi_j) = \mathbb{E}(\Delta_j \mathbb{E}(\xi_j \mid \{x_i\}_{i \in [n]})) = 0.
$$

The presence of $\Delta_j$ effectively inflates the variance in $z_j$ to $\sigma_0^2/n + \mathrm{Var}(\Delta_j)$. One can show that $\mathrm{Var}(\Delta_j) = n^{-1}\mathrm{Var}(f^*(x)\phi_j(x))$, which is bounded by $n^{-1}\|f^*\|_\infty^2$. This is how we will control the impact of $\Delta_j$ in the following development.

### C.2. ESD for RKHS Regression

We start by analyzing the counterpart of the PC estimator, the Kernel Principal Component Projection Estimator (KPCPE), defined as

$$
\hat{f}_k^{\mathrm{PC}}(x) := \sum_{j:\lambda_j \geq \lambda_{\pi_k}} z_j \phi_j(x), \tag{22}
$$

where $k$ is the number of leading eigenfunctions to be included.

The risk $\mathcal{R}_k := \mathbb{E}\|\hat{f}_k^{\mathrm{PC}} - f^*\|_{L^2(\mu)}^2$ decomposes into squared bias $B(k)$ and variance $V(k)$. Since $\mathbb{E}[\phi_{j'}(x_i)\phi_j(x_i)] = \mathbf{1}_{\{j=j'\}}$, we have $\mathbb{E}[z_j] = \theta_j^*$. Therefore, the bias is due to truncation and is given by

$$
B(k) = \sum_{j=k+1}^{\infty} (\theta_{\pi_j}^*)^2. \tag{23}
$$

The integrated variance is $V(k) = \sum_{j:\lambda_j \geq \lambda_{\pi_k}} \text{Var}(z_j)$. Using the law of total variance, we have

$$\text{Var}(z_j) = \frac{1}{n}\text{Var}(y_i\phi_j(x_i)) = \frac{1}{n}(\sigma_0^2 + \tau_j^2), \quad \text{where} \quad \tau_j^2 := \text{Var}(f^*(x)\phi_j(x)). \tag{24}$$

The term $\tau_j^2$ arises from the randomness of the design $x$. To ensure $V(k)$ grows at the rate of $k/n$, we need to uniformly bound the design-induced variance $\tau_j^2$. To illustrate the idea, we present one particular way to control $\tau_j^2$. We impose the following boundedness condition.

**Assumption C.1** (Bounded target). There exists $M_f < \infty$ such that $|f^*(x)| \leq M_f$ for $\mu$-almost every $x \in \mathcal{X}$. We denote the minimal such bound by

$$\|f^*\|_\infty := \inf\{M \geq 0 : |f^*(x)| \leq M \text{ for } \mu\text{-almost every } x \in \mathcal{X}\} = \operatorname*{ess\,sup}_{x \in \mathcal{X}} |f^*(x)|.$$

Assumption C.1 is very mild: for compact $\mathcal{X}$, if $f^*$ is continuous, then $f^*$ is bounded.

Under Assumption C.1, $\tau_j^2 \leq \mathbb{E}[f^*(x)^2\phi_j(x)^2] \leq \|f^*\|_\infty^2$. Consequently, the variance is bounded by $V(k) \leq \frac{k}{n}(\sigma_0^2 + \|f^*\|_\infty^2)$. This motivates us to define the effective noise variance per component as

$$\sigma^2 := \frac{\sigma_0^2 + \|f^*\|_\infty^2}{n}. \tag{25}$$

The effective noise variance $\sigma^2$ includes the term $\|f^*\|_\infty^2/n$, which inflates the noise compared to an idealized sequence model.

*Remark* C.2 (Role of Assumption C.1). The purpose of Assumption C.1 is to control the random-design fluctuation in the transformed coordinate $z_j$ without imposing a source condition on $f^*$. Once the uniform control on $\tau_j^2$ in Equation (24) is available, the RKHS development can reuse the sequence-model accounting with the single effective noise level $\sigma^2$ from Equation (25). Consequently, we can use ESD to measure spectral alignment through the tail profile of $\theta^*$.

We can now adapt the definitions from Section 3.1 using the effective noise variance $\sigma^2$.

**Definition C.3** (ESD for RKHS Regression). The Effective Span Dimension (ESD) of $f^* \in \mathcal{S}_\mathbf{k}$ with respect to the kernel $\mathbf{k}$ and the effective noise variance $\sigma^2 = (\sigma_0^2 + \|f^*\|_\infty^2)/n$ is defined as

$$d^\dagger = d^\dagger(\sigma^2; f^*, \mathbf{k}) = \min\{k \in \mathbb{N}_+ \cup \{\infty\} : \mathbf{H}_{\boldsymbol{\theta}^*,\boldsymbol{\lambda}}(k) \leq \sigma^2\}, \tag{26}$$

where $\boldsymbol{\theta}^* = \{\theta_j^*\}_{j\geq 1}$ and $\mathbf{H}_{\boldsymbol{\theta}^*,\boldsymbol{\lambda}}(k)$ is defined as in Equation (9).

The risk of the KPCPE estimator can be bounded using this ESD.

**Proposition C.4** (Optimal KPCPE Risk Bound). Let $\hat{f}_k^{\text{PC}}$ be the KPCPE estimator defined in Equation (22). Let $\mathcal{R}_*^{\text{PC}} = \min_{k\geq 1} \mathcal{R}(\hat{f}_k^{\text{PC}}; f^*)$. Under Assumption C.1, it holds that:

$$(d^\dagger - 1)\frac{\sigma_0^2}{n} \leq \mathcal{R}_*^{\text{PC}} \leq 2d^\dagger\sigma^2 = 2d^\dagger\frac{\sigma_0^2 + \|f^*\|_\infty^2}{n}, \tag{27}$$

where $d^\dagger = d^\dagger(\sigma^2; f^*, \mathbf{k})$ is the ESD from Definition C.3. In particular, if $\|f^*\|_\infty^2 \lesssim \sigma_0^2$, we can conclude that $\mathcal{R}_*^{\text{PC}} \asymp d^\dagger\sigma_0^2/n$.

We comment that Zhang et al. (2024b) have established the minimax optimality of the oracle–tuned PC estimator. Proposition C.4 suggests that the risk of the oracle–tuned PC estimator scales as $d^\dagger/n$. Therefore, we essentially express the minimax rate therein using the ESD without reliance on the classical eigen-decay conditions or source conditions.

We can also extend the minimax framework in Section 4 to RKHS regression. Let $K$ be a quota on the ESD, and let $C_0$ be a constant. Define the class based on the span profile as follows:

$$\mathcal{F}_{K,\mathbf{k}}^{(n)} = \{f^* \in \mathcal{S}_\mathbf{k} \cap L^\infty(\mathcal{X}, \mu) : \|f^*\|_\infty \leq \sigma_0 C_0, \quad d^\dagger(\bar{\sigma}^2/n; f^*, \mathbf{k}) \leq K\}, \tag{28}$$

where $\bar{\sigma}^2 = \sigma_0^2(1 + C_0^2)$. We further impose the following assumption on the spectrum.

**Assumption C.5.** The kernel $\mathbf{k}$ is said to be $(K, n)$-regular if there are some constants $c_1 \in (0, 1)$ and $C_1$ such that $\sum_{i \leq c_1 K} \lambda_{\pi_i}^{-1} \leq C_1 n$.

*Remark* C.6 (Role of Assumption C.5). Assumption C.5 ensures that the leading $O(K)$ eigendirections used in the minimax lower bound are not too weak relative to the sample size. Assumption C.5 is a spectral condition on the population kernel and the target ESD scale $K$, not a regularity condition on $f^*$ or on the individual eigenfunctions. This keeps the RKHS minimax lower bound free of additional eigenfunction geometry assumptions: the packing functions are controlled through the population spectrum and the Mercer bound, without imposing eigenfunction incoherence, sign cancellation, or uniform eigenfunction boundedness. The price of this generality is conservativeness. In particular, under polynomial eigen-decay, Example C.9 recovers the usual source condition rate in the regime $s \geq 1$; extending the RKHS lower bound to $0 < s < 1$ is possible but would require additional eigenfunction geometry assumptions.

**Theorem C.7** (Minimax Risk over Span Profile Classes). *If $\mathbf{k}$ is $(K, n)$-regular, then the minimax risk over $\mathcal{F}_{K,\mathbf{k}}^{(n)}$ satisfies:*

$$\inf_{\hat{f}} \sup_{f^* \in \mathcal{F}_{K,\mathbf{k}}^{(n)}} \mathcal{R}(\hat{f}; f^*) \asymp \frac{\sigma_0^2 K}{n}, \tag{29}$$

*where the infimum is over all estimators $\hat{f}$.*

Combining Theorem C.7 and Proposition C.4 shows that the oracle–tuned KPCPE is minimax rate optimal over $\mathcal{F}_{K,\mathbf{k}}^{(n)}$, with rate $\sigma_0^2 K/n$.

The KPCPE serves as a simple benchmark for spectral methods. This analysis, via the ESD, characterizes the performance of the oracle–tuned KPCPE based directly on the properties of the specific signal $f^*$ (via $\theta^*$) and kernel spectrum $\boldsymbol{\lambda}$, without requiring standard assumptions like source conditions or polynomial eigenvalue decay. Therefore, we consider the ESD evaluated at the design-adjusted noise level $\sigma^2$ as a key measure of statistical complexity in RKHS regression. In summary, the span profile framework provides a unified perspective on the generalization performance of spectral methods across a variety of models.

**Minimax convergence rates.** Following the framework in Section 4, we can quantify a class of populations using a quota sequence $\boldsymbol{K} = \{K_n\}_{n=1}^{\infty}$. For some $n_0 \in \mathbb{N}_+$, define

$$\mathcal{F}_{\boldsymbol{K},\mathbf{k}} = \{f^* \in \mathcal{S}_{\mathbf{k}} \cap L^{\infty}(\mathcal{X}, \mu) : \|f^*\|_{\infty} \leq \sigma_0 C_0, \quad d^{\dagger}(\bar{\sigma}^2/n; f^*, \mathbf{k}) \leq K_n, \forall n \geq n_0\}, \tag{30}$$

where $\bar{\sigma}^2 = \sigma_0^2(1 + C_0^2)$. For a sample $\{(x_i, y_i)\}_{i=1}^n$ drawn from the model in Equation (19) and any estimator $\hat{f}$, we aim to determine the optimal convergence rate of the following minimax risk:

$$\inf_{\hat{f}} \sup_{f^* \in \mathcal{F}_{\boldsymbol{K},\mathbf{k}}} \mathcal{R}(\hat{f}, f^*). \tag{31}$$

We have the following result.

**Theorem C.8.** *Suppose Condition 4.1 holds for a quota sequence $\boldsymbol{K} = \{K_n\}_{n=1}^{\infty}$. Furthermore, suppose $\mathbf{k}$ is $(K_n, n)$-regular for all $n \geq n_0$. If $\{(x_i, y_i)\}_{i=1}^n$ is drawn from the model in Equation (19), it holds that*

$$\inf_{\hat{f}} \sup_{f^* \in \mathcal{F}_{\boldsymbol{K},\mathbf{k}}} \mathcal{R}(\hat{f}, f^*) \asymp \bar{\sigma}^2 \frac{K_n}{n}.$$

Assumption C.5 is a mild condition. The following is an example where we use Theorem C.8 to recover the minimax convergence rate derived under the classical polynomial eigen-decay conditions and source conditions.

*Example* C.9. Suppose $\mathbf{k}$ admits spectrum such that $\lambda_{\pi_i} \asymp i^{-\beta}$ with $\beta > 0$. Let $K_n = \lfloor n^{\frac{1}{1+s\beta}} \rfloor$ for any $s \geq 1$. It is easy to see that

$$\sum_{i \leq c_1 K_n} \lambda_{\pi_i}^{-1} \asymp (c_1 K_n)^{\beta+1} \asymp n^{\frac{\beta+1}{s\beta+1}} \lesssim n.$$

Therefore, the minimax optimal rate for the class $\mathcal{F}_{\boldsymbol{K},\mathbf{k}}$ is $\frac{\bar{\sigma}^2 K_n}{n} \asymp \sigma_0^2 n^{-\frac{s\beta}{1+s\beta}}$, which is the same as the optimal rate given by the source condition with smoothness parameter $s$.

**Extension beyond the range of the integral operator.** The above discussion makes use of the simplifying assumption that $f^* \in \mathcal{S}_{\mathbf{k}}$. The framework can also be extended to cover general $f^* \in L^2(\mathcal{X}, \mu)$, provided that the component of the target outside $\mathcal{S}_{\mathbf{k}}$ is treated as an irreducible approximation error. Let $\mathbf{P}_{\mathbf{k}}$ denote the $L^2$-orthogonal projection onto $\mathcal{S}_{\mathbf{k}}$. For a target function $f^* \in L^2(\mathcal{X}, \mu)$, write

$$\theta^*_{\pi_i} := \langle f^*, \phi_{\pi_i} \rangle_{L^2(\mathcal{X},\mu)}, \qquad i = 1, \ldots, d.$$

The component of $f^*$ outside $\mathcal{S}_{\mathbf{k}}$ induces the systematic squared bias

$$\Delta_{f^*,\mathbf{k}} := \|(\mathbf{I} - \mathbf{P}_{\mathbf{k}})f^*\|^2_{L^2(\mu)} = \|f^*\|^2_{L^2(\mu)} - \sum_{i=1}^{\infty} |\theta^*_{\pi_i}|^2.$$

This term is irreducible for any spectral estimator whose output lies in $\mathcal{S}_{\mathbf{k}}$, irrespective of the choice of regularization parameter. Indeed, for the estimator $\hat{f} = \sum_j \hat{\theta}_j \phi_j$, Equation (20) becomes

$$\mathcal{R}(\hat{f}; f^*) = \Delta_{f^*,\mathbf{k}} + \sum_{j=1}^{\infty} \mathbb{E}\left(\hat{\theta}_j - \theta^*_j\right)^2.$$

Accordingly, we may modify the definition of ESD by adding the systematic bias to the squared tail sum. Specifically, we modify the definition of ESD in Definition C.3 as

$$\bar{d}^\dagger = \bar{d}^\dagger(\sigma^2; f^*, \mathbf{k}) = \min\left\{ j \in [d] : \frac{1}{j}\left[\Delta_{f^*,\mathbf{k}} + \sum_{i=j+1}^{\infty} \left(\theta^*_{\pi_i}\right)^2\right] \leq \sigma^2 \right\}.$$

Note that when $d < \infty$ and $\Delta_{f^*,\mathbf{k}} > d\sigma^2$, the set in the above display is empty; in this case, $\bar{d}^\dagger$ is defined as $\lceil \frac{\Delta_{f^*,\mathbf{k}}}{\sigma^2} \rceil$. This convention emphasizes the nature of ESD as a risk-equivalent dimension rather than as a feasible spectral cutoff.

When $\bar{d}^\dagger \leq d$, $\sigma^2 \bar{d}^\dagger$ characterizes the risk of the oracle–tuned PC estimator as in Proposition C.4. This characterization continues to hold in the empty-set case. Using the same variance bound as in Proposition C.4,

$$\mathcal{R}^{\mathrm{PC}}_* \leq \Delta_{f^*,\mathbf{k}} + B(d) + V(d) = \Delta_{f^*,\mathbf{k}} + d\sigma^2 < 2\Delta_{f^*,\mathbf{k}} \leq 2\sigma^2 \bar{d}^\dagger.$$

Conversely, any spectral estimator whose output lies in $\mathcal{S}_{\mathbf{k}}$ has risk at least $\Delta_{f^*,\mathbf{k}}$. By the definition of $\bar{d}^\dagger$ as $\lceil \frac{\Delta_{f^*,\mathbf{k}}}{\sigma^2} \rceil$, we have $\Delta_{f^*,\mathbf{k}} > (\bar{d}^\dagger - 1)\sigma^2$.

### C.3. Connection to Random Design Linear Regression

The analysis in this section also covers an important case: random-design linear regression (as opposed to fixed-design linear regression). This is because linear regression can be viewed as an RKHS regression w.r.t. any positive-definite linear kernel $\mathbf{k}(x, x') = x^\top \mathbf{K} x'$ for $x, x' \in \mathbb{R}^p$, where $\mathbf{K} \in \mathbb{R}^{p \times p}$ is positive definite.

Let the support $\mathcal{X}$ be a compact subset of $\mathbb{R}^p$. Suppose $\boldsymbol{\Sigma}_x = \mathbb{E}_{x \sim \mu}(xx^\top)$ is positive definite and $\mathbf{L}$ is a symmetric square root of $\boldsymbol{\Sigma}_x$. Let the eigen-decomposition of $\mathbf{LKL}$ be

$$\mathbf{LKL} = \sum_{j=1}^{p} \lambda_j v_j v_j^\top = \mathbf{V}\boldsymbol{\Lambda}\mathbf{V}^\top,$$

where $\mathbf{V}$ is the matrix with columns formed by $v_j$ and $\boldsymbol{\Lambda} = \mathrm{Diag}(\lambda_1, \ldots, \lambda_p)$.

Define $\boldsymbol{\Psi} = \mathbf{L}^{-1}\mathbf{V}$. Then $\boldsymbol{\Psi}^\top \boldsymbol{\Sigma}_x \boldsymbol{\Psi} = \mathbf{V}^\top\mathbf{V} = \mathbf{I}_p$. Furthermore, we have

$$\mathbf{K} = \mathbf{L}^{-1}\mathbf{V}\boldsymbol{\Lambda}\mathbf{V}^\top\mathbf{L}^{-1} = \boldsymbol{\Psi}\boldsymbol{\Lambda}\boldsymbol{\Psi}^\top,$$

Suppose the columns of $\boldsymbol{\Psi}$ are $\phi_j$. We can then write

$$\mathbb{E}_{x \sim \mu}\left[\langle \phi_j, x \rangle \langle \phi_k, x \rangle\right] = \left(\boldsymbol{\Psi}^\top \boldsymbol{\Sigma}_x \boldsymbol{\Psi}\right)_{jk} = \mathbf{1}_{\{j=k\}}, \quad \forall j, k \in [p],$$

so $\{\phi_j\}$ is an orthonormal system in $L^2(\mathcal{X}, \mu)$. Furthermore, the kernel can be expressed as

$$\mathbf{k}(x, x') = x^\top \mathbf{K} x' = \sum_{j=1}^{p} \lambda_j \langle \phi_j, x \rangle \langle \phi_j, x' \rangle.$$

Hence, the pair $(\{\lambda_i\}, \{\phi_j\})$ forms the eigensystem for $\mathbf{k}$.

Consider linear regression where $y = f^*(x) + \epsilon$ and $f^*(x) = \langle \beta^*, x \rangle$. Define $\theta^* = \mathbf{\Psi}^{-1}\beta^* = \mathbf{V}^\top \mathbf{L}\beta^*$. We can write

$$f^*(x) = \langle \beta^*, x \rangle = \langle \mathbf{\Psi}^{-1}\beta^*, \mathbf{\Psi}^\top x \rangle = \sum_{j=1}^{p} \theta_j^* \langle \phi_j, x \rangle.$$

It is also clear that $\|f^*\|_\infty \leq \sup_{x \in \mathcal{X}} |\langle \beta^*, x \rangle| \leq \|\beta^*\|_2 C_\mathcal{X} < \infty$, where $C_\mathcal{X}$ is finite and depends on $\mathcal{X}$. We can define the effective noise level $\sigma^2 = n^{-1}(\sigma_0^2 + \|f^*\|_\infty^2)$.

Therefore, with respect to the kernel $\mathbf{k}$ and the basis $\{\phi_j\}$, we define the ESD exactly as in Definition C.3 using the coefficients $\{\theta_j^*\}$ and eigenvalues $\{\lambda_j\}$.

### C.4. Numerical Illustration

This section provides numerical validation of the relationship between the ESD and the risk of the oracle–tuned KPCPE, mirroring the setup for linear models in Appendix B.4. We use the cosine basis eigenfunctions $\phi_j(x) = \sqrt{2}\cos(2\pi j x)$ on the domain $[0, 1]$ with inputs sampled as $x_i \overset{\text{i.i.d.}}{\sim} \text{Unif}[0, 1]$. The sample size is fixed at $n = 400$, and for numerical purposes, we consider the first $J = 800$ eigenfunctions. The noise variance is set as $\sigma_0^2 = 1$.

**Experimental Setup:** We set the baseline kernel eigenvalue spectrum as $\lambda_{j,0} = j^{-1.1}$ and the fixed signal coefficients as $\theta_j^* = j^{-4}$. To study the impact of misalignment between the kernel spectrum and the signal, we introduce a severity parameter $\alpha \geq 0$ and define the modified eigenvalue spectrum as

$$\lambda_j(\alpha) = \lambda_{j,0} \exp(\alpha t_j), \quad t_j = \frac{j-1}{D-1} \text{ for } j \leq D, \text{ and } t_j = 0 \text{ otherwise,}$$

with $D = 80$. As $\alpha$ increases, the leading $D$ eigenvalues become progressively magnified, with the largest index having the most significant increase. Consequently, the modified kernel places more emphasis on directions that receive less signal energy, so the optimal KPCPE selects more principal components.

As the severity parameter $\alpha$ grows, only the first $D$ eigenvalues are changed while the rest of the spectrum is untouched. Among the changed ones, the leading eigenvalues are magnified by a smaller constant, so that the resulting kernel has its leading subspaces aligned with the directions in which the signal has less of its energy and thus increases the misalignment.

For each $\alpha$ in a specified grid, we compute two quantities:

- The Effective Span Dimension:

$$d_{\text{eff}}^\dagger(\alpha) = d^\dagger\left(\sigma_{\text{eff}}^2; f^*, \lambda(\alpha)\right), \quad \sigma_{\text{eff}}^2 := \frac{\sigma_0^2 + \|f^*\|_\infty^2}{n},$$

  where $\|f^*\|_\infty$ is estimated numerically on a dense grid from the fixed coefficients of $f^*$.

- The risk of the oracle–tuned KPCPE:

$$\mathcal{R}_*(\alpha) = \min_k \mathbb{E}\|\hat{f}_k^{\text{PC}}(\alpha) - f^*\|_{L^2(\mu)}^2,$$

  where the estimator $\hat{f}_k^{\text{PC}}(\alpha)$ is computed using the spectrum $\lambda(\alpha)$ and the expectation is estimated by averaging prediction error over $B = 10$ Monte Carlo replications.

Figure 4 plots the empirically computed optimal KPCPE risk (orange solid line) alongside the theoretical lower bound $(d^\dagger - 1)\sigma_0^2/n$ (blue dashed line) and upper bound $2d^\dagger \sigma_{\text{eff}}^2$ (green dotted line). The empirical risk consistently lies between the two theoretical curves, confirming the validity of the bounds derived in Proposition C.4.

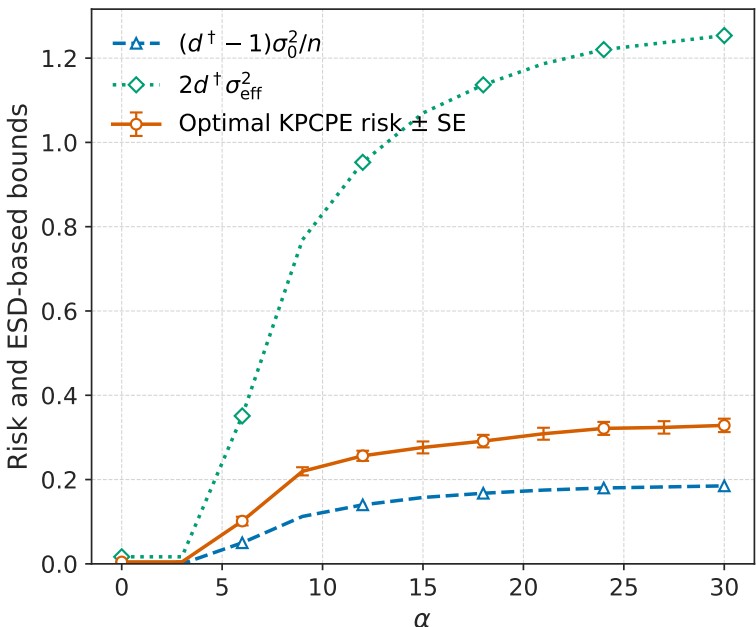

*Figure 4.* Optimal KPCPE risk and ESD-based bounds. The blue dashed curve plots the lower bound $(d^\dagger(\alpha) - 1)\sigma_0^2/n$. The green dotted curve plots the ESD-based upper curve $2d^\dagger(\alpha)(\sigma_0^2 + \|f^*\|_\infty^2)/n$. The orange curve with error bars plots the Monte Carlo estimate of the risk of the oracle–tuned KPCPE. The risk is averaged over 10 replications, and the error bars represent one standard error.

As the severity parameter $\alpha$ increases, the resulting spectral perturbation shifts energy into higher-index eigenfunctions. This inflates the ESD and consequently the minimal achievable risk.

Overall, this experiment demonstrates that our span profile framework provides an accurate and robust characterization of generalization performance in RKHS regression, consistent with earlier observations made for the sequence and linear regression models.

## D. Measuring Alignment via ESD

We illustrate how the notion of ESD can be used to measure the alignment between the signal and the kernel.

### D.1. An Example Comparing Signal-Spectrum Alignment

The following simple example illustrates how to compare signal-spectrum alignment across the spectra discussed in Section 3.2.

Suppose $d < \infty$, $\boldsymbol{\theta}^*$ is $s$-sparse with support $S \subset [d]$, and $s = |S| \ll d$. Consider the following two spectra with the same set of eigenvalues but different allocations:

(1) The $s$ largest eigenvalues of $\boldsymbol{\lambda}^{(1)}$ are located on $S$;
(2) The $d - s$ largest eigenvalues of $\boldsymbol{\lambda}^{(2)}$ are located on $S^c = [d] \setminus S$.

Intuitively, $\boldsymbol{\lambda}^{(1)}$ aligns better with $\boldsymbol{\theta}^*$ than $\boldsymbol{\lambda}^{(2)}$. However, a quantitative analysis is not straightforward without using the notion of ESD.

First, we note that the effective dimension (Zhang, 2005) is the same for both spectra because they share the same set of eigenvalues. Similarly, the covariance-splitting index $k^*$ (Bartlett et al., 2020) is the same for both spectra. Thus, these signal-agnostic complexity measures do not distinguish signal-spectrum alignment between the two spectra.

Next, we consider the ESD and the span profile. Rigorously, we can show that for any $\tau$,

$$\mathbf{D}_{\boldsymbol{\theta}^*, \boldsymbol{\lambda}^{(1)}}(\tau) \leq s, \text{ and } \mathbf{D}_{\boldsymbol{\theta}^*, \boldsymbol{\lambda}^{(2)}}(\tau) \geq \min\left(d - s, \|\boldsymbol{\theta}^*\|^2/\tau\right).$$

Indeed, under $\boldsymbol{\lambda}^{(1)}$, all nonzero coordinates of $\theta^*$ are contained among the first $s$ eigencoordinates. Hence the tail energy after the first $s$ coordinates is zero, so $\mathbf{D}_{\boldsymbol{\theta}^*, \boldsymbol{\lambda}^{(1)}}(\tau) \leq s$.

Under $\boldsymbol{\lambda}^{(2)}$, the first $d - s$ eigencoordinates all lie in $S^c$, and therefore carry no signal. Hence for every $k \leq d - s$,

$$\mathbf{H}_{\boldsymbol{\theta}^*, \boldsymbol{\lambda}^{(2)}}(k) = \frac{1}{k} \sum_{i > k} (\theta_{\pi_i}^*)^2 = \frac{\|\theta^*\|_2^2}{k}.$$

Therefore, for any $k < \min\{d - s, \|\theta^*\|_2^2/\tau\}$, we have $\mathbf{H}_{\boldsymbol{\theta}^*, \boldsymbol{\lambda}^{(2)}}(k) > \tau$. This shows that $\mathbf{D}_{\boldsymbol{\theta}^*, \boldsymbol{\lambda}^{(2)}}(\tau) \geq \min\left(d - s, \|\boldsymbol{\theta}^*\|^2/\tau\right)$.

Hence, for sufficiently small $\tau$, their ratio

$$r(\tau) = \mathbf{D}_{\boldsymbol{\theta}^*, \boldsymbol{\lambda}^{(1)}}(\tau)/\mathbf{D}_{\boldsymbol{\theta}^*, \boldsymbol{\lambda}^{(2)}}(\tau) \leq s/(d - s) \ll 1.$$

In view of Theorem 3.3, this suggests that the minimax risk under $\boldsymbol{\lambda}^{(1)}$ is substantially lower than under $\boldsymbol{\lambda}^{(2)}$ when the noise level is small. Therefore, spectral estimators using $\boldsymbol{\lambda}^{(1)}$ are preferred.

### D.2. Pathwise ESD for Learned Kernels

Section 5 analyzes eigenvalue learning because OP-GF admits tractable dynamics under a fixed eigenbasis. This limitation is specific to that analysis rather than to the definition of ESD. For learned kernels with evolving eigenfunctions, ESD can still be used as a pathwise descriptor after fixing the realized kernel at each training time. This subsection defines that pathwise formulation and illustrates, through an experiment, how changes in the learned spectrum and eigenfunctions can be tracked by ESD.

Let $\mathbf{k}_t$ be the kernel learned at training time $t$, with eigenvalues $\{\lambda_j(t)\}$ (sorted in decreasing order) and eigenfunctions $\{\phi_j^{(t)}\}$ that are orthonormal in $L^2(\mathcal{X}, \mu)$. To understand how the signal-kernel alignment evolves, we define the pathwise ESD as

$$d^\dagger(t) := d^\dagger(\sigma^2; f^*, \mathbf{k}_t), t \geq 0,$$

where we have followed Definition C.3 to define $d^\dagger(\sigma^2; f^*, \mathbf{k}_t)$ as the ESD of $f^*$ w.r.t. the kernel $\mathbf{k}_t$ using $\theta_j^{*,(t)} = \langle f^*, \phi_j^{(t)} \rangle$ and $\sigma^2 = n^{-1}(\sigma_0^2 + \|f^*\|_\infty^2)$.

Let $\mathbf{H}_t(k) := \frac{1}{k} \sum_{i > k} \left[\theta_i^{*,(t)}\right]^2$. If training aligns the leading eigenfunctions $\phi_j^{(t)}$ better with $f^*$, then $\{\theta_j^{*,(t)}\}$ concentrates more on the leading indices, and thus $\mathbf{H}_t(k)$ decreases for all $k$, which implies a decrease in $d^\dagger(t)$.

**Experiment on Deep Linear Networks.** To demonstrate this pathwise perspective, we simulate a random-design linear regression.

Each covariate coordinate is drawn independently from $\{\pm 1\}$, so $\boldsymbol{\Sigma}_x = \mathbb{E}(XX^\top) = \mathbf{I}_p$ and $\|X\|_\infty = 1$. We set $p = 900$, and specify the true parameters as follows: $\beta^*$ follows a power-law decay with $\beta_j^* = j^{-1.1}$ for $1 \leq j \leq 200$ and $\beta_j^* = 0$ for $j > 200$. The response is $Y = \langle \beta^*, X \rangle + \varepsilon$ with $\varepsilon \sim N(0, \sigma_0^2)$ and $\sigma_0 = 0.1$.

We draw $n = 1000$ samples and train a deep *linear network* with $D = 4$ hidden affine layers without bias using full-batch Adam with learning rate $10^{-4}$. The hidden weight matrices of the network are $\mathbf{W}_\ell(t) \in \mathbb{R}^{p \times p}$ for $\ell = 1, \ldots, D$ (using a near-identity initialization), and the weight of the final linear layer is $w(t) \in \mathbb{R}^p$.

The estimated function at time $t$ is given by $f_t(x) = w(t)^\top \mathbf{A}(t)x$, where $\mathbf{A}(t) := \mathbf{W}_D(t) \cdots \mathbf{W}_1(t)$. We form the learned kernel $\mathbf{k}_t(x, x') = \langle \mathbf{A}(t)x, \mathbf{A}(t)x' \rangle = x^\top \mathbf{G}_t x'$, where $G_t := \mathbf{A}(t)^\top \mathbf{A}(t)$. We then follow the derivation in Appendix C.3 and define the ESD $d^\dagger(t)$ of $f^*$ w.r.t. the kernel $\mathbf{k}_t$. Since $\|X\|_\infty = 1$ $\mu$-a.s., we have $\|f^*\|_\infty^2 = \|\beta^*\|_1^2$; this is used in computing the effective noise level.

Figure 5 illustrates that adaptive representation learning can reduce the pathwise ESD $d^\dagger(t)$ along training, together with the risk $\|\mathbf{A}(t)^\top w(t) - \beta^*\|_2^2$. This supports the use of ESD for capturing the alignment between signal and kernel in evolving-eigenfunction settings.

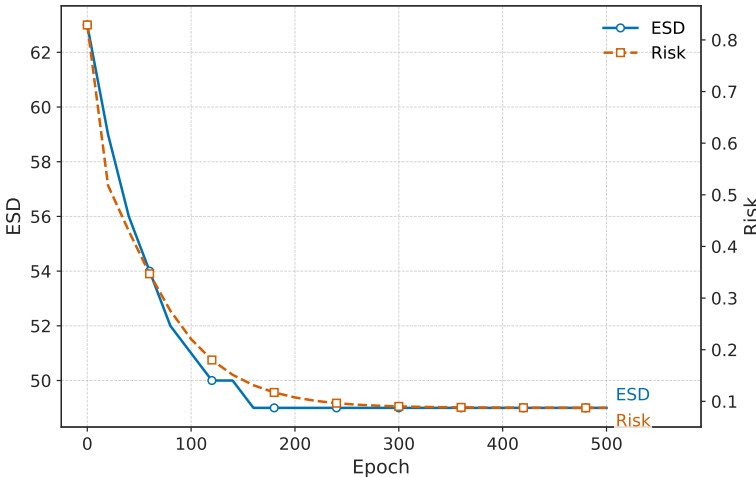

*Figure 5.* Pathwise ESD and risk under a learned kernel using a 4-layer linear network.

## E. Proof

### E.1. Proofs of Results on ESD of Sequence Models

*Proof of Proposition 3.2.* For any $\nu > 0$, define

$$k_{\boldsymbol{\lambda}}(\nu) = \#\{ j : \lambda_j \geq \nu \},$$

which counts how many eigenvalues exceed the threshold $\nu$. The PC estimator sets

$$\hat{\theta}_i^{\nu} = \mathbf{1}_{\{\lambda_i \geq \nu\}} z_i, \ i \in [d].$$

Its squared bias and variance are given by

$$B^{\mathrm{PC}}(\nu) = \sum_{i:\lambda_i < \nu} (\theta_i^*)^2, \ V^{\mathrm{PC}}(\nu) = \sum_{i:\lambda_i \geq \nu} \sigma^2 = k_{\boldsymbol{\lambda}}(\nu)\,\sigma^2.$$

For any threshold $\nu$, we can reparameterize the bias and variance using $k = k_{\boldsymbol{\lambda}}(\nu)$ as

$$B^{\mathrm{PC}}(k) = \sum_{i=k+1}^{d} \left(\theta_{\pi_i}^*\right)^2, \quad \text{and} \quad V^{\mathrm{PC}}(k) = k\,\sigma^2, k = 0, 1, \ldots, d.$$

The function $B^{\mathrm{PC}}(k)$ decreases in $k$, while $V^{\mathrm{PC}}(k)$ increases in $k$. The risk function is given by $\mathcal{R}^{\mathrm{PC}}(k) = B^{\mathrm{PC}}(k) + V^{\mathrm{PC}}(k)$.

For any integer $k \geq 1$, we have

$$\mathcal{R}^{\mathrm{PC}}(k) = k \left( \sigma^2 + \frac{1}{k} \sum_{i:\lambda_i < \lambda_{\pi_k}} (\theta_i^*)^2 \right).$$

**Upper bound** For $k = d^{\dagger}$, we have $\frac{1}{k} \sum_{i:\lambda_i < \lambda_{\pi_k}} (\theta_i^*)^2 \leq \sigma^2$. By definition of the optimal risk, we have

$$\mathcal{R}_*^{\mathrm{PC}} \leq \mathcal{R}^{\mathrm{PC}}(d^{\dagger}) \leq 2 d^{\dagger} \sigma^2.$$

Therefore, the upper bound is proved.

**Lower bound**  Without loss of generality, assume $d^\dagger \geq 2$. For any $k \leq d^\dagger - 1$, we have

$$\mathcal{R}^{\mathrm{PC}}(k) \geq B^{\mathrm{PC}}(k) = \sum_{i=k+1}^{d} \left(\theta_{\pi_i}^*\right)^2 \geq \sum_{i=d^\dagger}^{d} \left(\theta_{\pi_i}^*\right)^2 > (d^\dagger - 1)\sigma^2,$$

where the last inequality comes from the definition of $d^\dagger$. For any $k \in [d^\dagger, d]$, we have

$$\mathcal{R}^{\mathrm{PC}}(k) \geq k\sigma^2 \geq d^\dagger \sigma^2.$$

Therefore, the lower bound is proved. $\qquad\qquad\qquad\qquad\qquad\qquad\qquad\qquad\qquad\qquad\qquad\qquad$ □

## E.2. Proof of Minimax Results

To prove minimax lower bounds, it suffices to focus on Gaussian models where the noise variables are $\xi_j \overset{\text{i.i.d.}}{\sim} N(0, \sigma^2)$.

*Proof of Theorem 4.3.* Throughout the proof, the quota sequence is fixed. For the upper bound, use the PC estimator retaining the top $K_n$ eigencoordinates. Write this estimator as

$$\widehat{\theta}_{\pi_i}^{\mathrm{PC}, K_n} = \begin{cases} z_{\pi_i}, & i \leq K_n, \\ 0, & i > K_n. \end{cases}$$

The cutoff $K_n$ is the class parameter at the fixed sample size $n$ and does not depend on the unknown signal. For any $\boldsymbol{\theta} \in \mathcal{F}_{\boldsymbol{K}, \boldsymbol{\lambda}}$, the class definition gives $d^\dagger(\sigma_0^2/n; \boldsymbol{\theta}, \boldsymbol{\lambda}) \leq K_n$, and thus $\mathbf{H}_{\boldsymbol{\theta}, \boldsymbol{\lambda}}(K_n) \leq \sigma_0^2/n$. The risk of this fixed-cutoff PC estimator is the sum of its squared tail bias and its variance:

$$\mathcal{R}(\widehat{\boldsymbol{\theta}}^{\mathrm{PC}, K_n}, \boldsymbol{\theta}) = \sum_{i > K_n} \theta_{\pi_i}^2 + K_n \frac{\sigma_0^2}{n}$$

$$= K_n \mathbf{H}_{\boldsymbol{\theta}, \boldsymbol{\lambda}}(K_n) + K_n \frac{\sigma_0^2}{n} \leq 2\sigma_0^2 \frac{K_n}{n}.$$

Taking the supremum over $\boldsymbol{\theta} \in \mathcal{F}_{\boldsymbol{K}, \boldsymbol{\lambda}}$ gives the upper bound.

It remains to prove the lower bound stated in Theorem 4.3. Recall the definition of $M_k$ in Condition 4.1. Define $s(k) = \sigma_0^2 \frac{k}{M_k}$ for any $k \in \bar{K}$. Also define $\mathbf{S}_{\boldsymbol{\theta}, \boldsymbol{\lambda}}(k) = k \mathbf{H}_{\boldsymbol{\theta}, \boldsymbol{\lambda}}(k)$.

For simplicity, we assume $n_0 = 1$; the other cases follow a similar argument.

We can express $\mathcal{F}_{\boldsymbol{K}, \boldsymbol{\lambda}}$ as follows.

**Lemma E.1.** *Under Condition 4.1, we have*

$$\mathcal{F}_{\boldsymbol{K}, \boldsymbol{\lambda}} = \left\{ \boldsymbol{\theta} \in \mathbb{R}^\infty : \mathbf{S}_{\boldsymbol{\theta}, \boldsymbol{\lambda}}(k) \leq s(k) \text{ for all } k \in [\bar{K}] \right\}.$$

*Proof of Lemma E.1.* Observe the relation that

$$\boldsymbol{\theta} \in \mathcal{F}_{\boldsymbol{K}, \boldsymbol{\lambda}} \quad \Longleftrightarrow \quad \mathbf{D}_{\boldsymbol{\theta}, \boldsymbol{\lambda}}\left(\frac{\sigma_0^2}{n}\right) \leq K_n, \ \forall n \geq 1 \quad \Longleftrightarrow \quad \mathbf{S}_{\boldsymbol{\theta}, \boldsymbol{\lambda}}(K_n) \leq \sigma_0^2 \frac{K_n}{n}, \ \forall n \geq 1.$$

By (1) of Condition 4.1, we have

$$\mathbf{S}_{\boldsymbol{\theta}, \boldsymbol{\lambda}}(K_n) \leq \sigma_0^2 \frac{K_n}{n}, \ \forall n \geq 1 \quad \Longleftrightarrow \quad \mathbf{S}_{\boldsymbol{\theta}, \boldsymbol{\lambda}}(k) \leq s(k) \text{ for all } k \in [\bar{K}].$$

Therefore, we can rewrite

$$\mathcal{F}_{\boldsymbol{K}, \boldsymbol{\lambda}} = \left\{ \boldsymbol{\theta} \in \mathbb{R}^\infty : \mathbf{S}_{\boldsymbol{\theta}, \boldsymbol{\lambda}}(k) \leq s(k) \text{ for all } k \in [\bar{K}] \right\}.$$

$\qquad\qquad\qquad\qquad\qquad\qquad\qquad\qquad\qquad\qquad\qquad\qquad\qquad\qquad\qquad\qquad\qquad\qquad\qquad\qquad\qquad$ □

Fix any $n$. Define $\delta = \sqrt{c\sigma_0^2/n}$ with the constant $c = \frac{1}{4}$. Consider assigning nonzero signals on the block $B_n = \{\pi_1, \dots, \pi_{K_n}\}$ to construct a subset of populations.

Specifically, we define the collection of hypercube vertices $\mathcal{V} = \{-1, 1\}^{K_n}$. For every vertex $v \in \mathcal{V}$, define a parameter vector $\boldsymbol{\theta}^{(v)} = (\theta_j^{(v)})_{j=1}^d$ as follows:

$$\theta_{\pi_i}^{(v)} = \delta\, v_i, \text{ for } i = 1, \dots, K_n, \qquad \text{and } \theta_j^{(v)} = 0 \text{ for } j \notin B_n. \tag{32}$$

There are $2^{K_n}$ such vectors $\{\boldsymbol{\theta}^{(v)}\}$, and they satisfy the following property.

**Lemma E.2.** *For any $v \in \mathcal{V}$, the parameter vector $\boldsymbol{\theta}^{(v)}$ constructed in Equation* (32) *lies in $\mathcal{F}_{\boldsymbol{K},\boldsymbol{\lambda}}$.*

*Proof of Lemma E.2.* For any $k \in [\bar{K}]$, if $k \geq K_n$, then $\mathbf{S}_{\boldsymbol{\theta}^{(v)},\boldsymbol{\lambda}}(k)$ is 0.

If $1 \leq k \leq K_n - 1$, then $\mathbf{S}_{\boldsymbol{\theta}^{(v)},\boldsymbol{\lambda}}(k) \leq K_n \delta^2 = c\sigma_0^2 K_n/n$. Denote $k_0 = K_n - 1$ and $L = 1 + M_{k_0}$. By definition of $M_{k_0}$, we have $n \geq L$. Since $k \leq k_0$, we have $L \geq 1 + M_k$. We have

$$\begin{aligned}
\frac{K_n}{n} &\leq \frac{K_n}{L} \\
&= \frac{k_0 + 1}{1 + M_{k_0}} \\
&\leq 2\frac{k_0}{M_{k_0}} \\
&\leq 2\frac{k}{M_k},
\end{aligned} \tag{33}$$

where the second last inequality is because $(1 + k_0)/(1 + m) \leq 2k_0/m \Leftrightarrow m + k_0 m \leq 2k_0 + 2k_0 m$ and the last inequality is due to (2) of Condition 4.1. Since $2c < 1$, we see that $\sigma_0^2 c K_n/n \leq s(k)$ for all $k < K_n$.

In either case, we have $\mathbf{S}_{\boldsymbol{\theta}^{(v)},\boldsymbol{\lambda}}(k) \leq s(k)$ for all $k \in [\bar{K}]$, and thus $\boldsymbol{\theta}^{(v)} \in \mathcal{F}_{\boldsymbol{K},\boldsymbol{\lambda}}$. $\qquad\square$

For each $v \in \mathcal{V}$, let $P_v$ be the sampling distribution of the sequence model in Equation (4) with $\boldsymbol{\theta}^* = \boldsymbol{\theta}^{(v)}$, $\sigma^2 = \sigma_0^2/n$, and i.i.d. normal noise variables $\{\xi_j\}_{j \in [d]}$ from $N(0, \sigma^2)$. Let $\rho$ be the Hamming distance on $\mathcal{V}$. If $v$ and $w \in \mathcal{V}$ differ in exactly one coordinate (i.e., $\rho(v, w) = 1$), then

- $\|\boldsymbol{\theta}^{(v)} - \boldsymbol{\theta}^{(w)}\|^2 \geq (2\delta)^2$, and

- the Kullback-Leibler divergence between $P_v$ and $P_w$ satisfies $\mathrm{KL}(P_v \,\|\, P_w) = \frac{1}{2\sigma^2}(2\delta)^2 = 2c \leq \frac{1}{2}$, and by Pinsker's inequality, $\|P_v \wedge P_w\| = 1 - \mathrm{TV}(P_v, P_w) \geq 1 - \sqrt{\mathrm{KL}(P_v \,\|\, P_w)/2} \geq 1/2$.

By Assouad's Lemma (Lemma 2 in Yu (1997)), for any estimator $\widehat{\boldsymbol{\theta}}$ based on a sample $\boldsymbol{Z}^{(n)}$ drawn from $P_v$, we have

$$\sup_{v \in \mathcal{V}} \mathbb{E}_v \|\widehat{\boldsymbol{\theta}} - \boldsymbol{\theta}^{(v)}\|^2 \geq K_n \frac{(2\delta)^2}{4} = c\sigma_0^2 \frac{K_n}{n}.$$

$\qquad\square$

*Proof of Theorem 3.3.* For the upper bound, use the PC estimator retaining the top $K$ eigencoordinates. Write this estimator as

$$\widehat{\theta}_{\pi_i}^{\mathrm{PC},K} = \begin{cases} z_{\pi_i}, & i \leq K, \\ 0, & i > K. \end{cases}$$

The cutoff $K$ is the class parameter in $\mathcal{F}_{K,\boldsymbol{\lambda}}^{(\sigma^2)}$ and does not depend on the unknown signal. For any $\boldsymbol{\theta}^* \in \mathcal{F}_{K,\boldsymbol{\lambda}}^{(\sigma^2)}$, the class definition gives $d^\dagger(\sigma^2; \boldsymbol{\theta}^*, \boldsymbol{\lambda}) \leq K$, and thus $\mathbf{H}_{\boldsymbol{\theta}^*,\boldsymbol{\lambda}}(K) \leq \sigma^2$. The risk of this fixed-cutoff PC estimator is the sum of its

squared tail bias and its variance:

$$\mathcal{R}(\widehat{\boldsymbol{\theta}}^{\mathrm{PC},K}, \boldsymbol{\theta}^*) = \sum_{i>K} (\theta^*_{\pi_i})^2 + K\sigma^2$$
$$= K\mathbf{H}_{\boldsymbol{\theta}^*,\boldsymbol{\lambda}}(K) + K\sigma^2 \leq 2K\sigma^2.$$

Taking the supremum over $\boldsymbol{\theta}^* \in \mathcal{F}^{(\sigma^2)}_{K,\boldsymbol{\lambda}}$ gives the upper bound.

For the lower bound, the main idea is the same as the proof for Theorem 4.3.

Let $\delta = \sqrt{c\sigma^2}$ with the constant $c = \frac{1}{4}$. Let $B = \{\pi_1, \ldots, \pi_K\}$. We define the collection of hypercube vertices $\mathcal{V} = \{-1,1\}^K$. For every vertex $v \in \mathcal{V}$, define a parameter vector $\boldsymbol{\theta}^{(v)} = (\theta^{(v)}_j)^d_{j=1}$ as in Equation (32). There are $2^K$ such vectors $\{\boldsymbol{\theta}^{(v)}\}$. For each $v \in \mathcal{V}$, let $P_v$ be the sampling distribution of the sequence model in Equation (4) with $\boldsymbol{\theta}^* = \boldsymbol{\theta}^{(v)}$, $\sigma^2$, and i.i.d. normal noise variables $\{\xi_j\}_{j \in [d]}$. Let $\rho$ be the Hamming distance on $\mathcal{V}$. The rest of the proof is identical to that of Theorem 4.3 and is omitted. $\square$

## E.3. Details of Examples in Example 3.4

We provide the details of Example 3.4 for illustration of the concepts of ESD and span profile through several examples.

*Example* E.3 (Polynomial signals ($\alpha > 1$)). Suppose $\theta^*_i = i^{-\alpha/2}$ for some constant $\alpha > 1$, and $\{\lambda_i\}^d_1$ are decreasing. By an integral approximation, we can get $\mathbf{H}_{\boldsymbol{\theta}^*,\boldsymbol{\lambda}}(k) \leq \frac{1}{\alpha-1} k^{-\alpha}$. Therefore, we have $\mathbf{D}_{\boldsymbol{\theta}^*,\boldsymbol{\lambda}}(\sigma^2) \lesssim [\sigma^2]^{-\frac{1}{\alpha}}$. The optimal risk of PC estimator satisfies

$$\mathcal{R}^{\mathrm{PC}}_* \leq 2\sigma^2 \mathbf{D}_{\boldsymbol{\theta}^*,\boldsymbol{\lambda}}(\sigma^2) \lesssim \min\left([\sigma^2]^{1-\frac{1}{\alpha}}, d\sigma^2\right).$$

*Example* E.4 (Polynomial signals ($\alpha = 1$)). Suppose $d < \infty$, $\theta^*_i = i^{-1/2}$, and $\{\lambda_i\}^d_1$ are decreasing. We will show below that for some constant $C$, if $d\sigma^2 \leq e$, then $\mathcal{R}^{\mathrm{PC}}_* \leq Cd\sigma^2$, and if $d\sigma^2 > e$, then $\mathcal{R}^{\mathrm{PC}}_* \leq C\log(d\sigma^2/\log(d\sigma^2))$.

*Example* E.5 (Polynomial signals ($\alpha < 1$)). Suppose $d < \infty$, $\theta^*_i = i^{-\alpha/2}$, and $\{\lambda_i\}$ is decreasing. We will show below that $\mathcal{R}^{\mathrm{PC}}_* \lesssim d\min\left(d^{-\alpha}, \sigma^2\right)$.

These examples suggest that the span-profile framework not only recovers classical results but also extends them to settings where the classical framework is inapplicable

*Details of Example E.4.* We have $\mathbf{H}_{\boldsymbol{\theta}^*,\boldsymbol{\lambda}}(k) \leq k^{-1} \int^d_k \frac{1}{x} dx = k^{-1}\left(\log d - \log k\right)$.

By dropping the term $\log k$ in the numerator, it is easy to see that a sufficient condition for $\mathbf{H}_{\boldsymbol{\theta}^*,\boldsymbol{\lambda}}(k) \leq \sigma^2$ is given by $k \geq \sigma^{-2} \log(d)$. Therefore, we have $\mathbf{D}_{\boldsymbol{\theta}^*,\boldsymbol{\lambda}}(\sigma^2) \leq \lceil \sigma^{-2} \log(d) \rceil$.

The upper bound can be improved. Suppose $A > 1$ satisfies $d\sigma^2 \leq A \log A$. If $k \geq \sigma^{-2} \log A$, then

$$\frac{k}{d} \geq \frac{\log A}{d\sigma^2} \geq \frac{d\sigma^2/A}{d\sigma^2} = \frac{1}{A},$$

which implies that $\mathbf{H}_{\boldsymbol{\theta}^*,\boldsymbol{\lambda}}(k) \leq k^{-1} \log A \leq \sigma^2$. Therefore, we have

$$\mathbf{D}_{\boldsymbol{\theta}^*,\boldsymbol{\lambda}}(\sigma^2) \leq \min\left(d, \lceil \sigma^{-2} \log A \rceil\right).$$

By elementary calculus, if $y > e$, the solution to $x \log x = y$ satisfies that $x \in (e, y)$, and thus $\log x \in (1, \log(y))$, which implies $x > y/\log(y)$ and thus $x < y/\log(y/\log(y)) = y/(\log y - \log \log y) < 2y/\log(y)$.

If $d\sigma^2 \leq e$, we can take $A = e$ and conclude

$$\mathcal{R}^{\mathrm{PC}}_* \leq 2\sigma^2 \mathbf{D}_{\boldsymbol{\theta}^*,\boldsymbol{\lambda}}(\sigma^2) \lesssim d\sigma^2.$$

If $d\sigma^2 > e$, then $\log(d\sigma^2) > 1$ and we can take $A = 2d\sigma^2/\log(d\sigma^2)$, which implies that

$$\mathcal{R}^{\mathrm{PC}}_* \leq 2\sigma^2 \mathbf{D}_{\boldsymbol{\theta}^*,\boldsymbol{\lambda}}(\sigma^2) \lesssim \log(d\sigma^2) - \log(\log(d\sigma^2)).$$

$\square$

*Details of Example E.5.* By an integral approximation, we see that

$$\mathbf{H}_{\boldsymbol{\theta}^*,\boldsymbol{\lambda}}(k) \asymp k^{-1}\left(d^{1-\alpha} - k^{1-\alpha}\right).$$

**Case 1:** $\sigma^2 d^\alpha < 2$. We have the default bound $\mathbf{D}_{\boldsymbol{\theta}^*,\boldsymbol{\lambda}}(\sigma^2) \le d$.

**Case 2:** $\sigma^2 d^\alpha \ge 2$. If $k \ge d^{1-\alpha}/\sigma^2$, then $\mathbf{H}_{\boldsymbol{\theta}^*,\boldsymbol{\lambda}}(k) \le \sigma^2$. Therefore, we have $\mathbf{D}_{\boldsymbol{\theta}^*,\boldsymbol{\lambda}}(\sigma^2) \le \lceil d^{1-\alpha}/\sigma^2 \rceil$, which is not larger than $\lceil d/2 \rceil$.

Combining both cases, we have $\mathbf{D}_{\boldsymbol{\theta}^*,\boldsymbol{\lambda}}(\sigma^2) \lesssim d\min(1/(d^\alpha\sigma^2), 1)$. Multiplying by $2\sigma^2$ on both sides, we have

$$\mathcal{R}_*^{\mathrm{PC}} \le 2\sigma^2 \mathbf{D}_{\boldsymbol{\theta}^*,\boldsymbol{\lambda}}(\sigma^2) \lesssim d\min\left(d^{-\alpha}, \sigma^2\right).$$

$\square$

### E.4. Details of Example 3.5

In Example 3.5, we show that a classical source ellipsoid is embedded into an ESD class of the corresponding size, and that this size is sharp in the worst case.

**Proposition E.6.** *Recall the notations in Example 3.5. We have*

$$\Theta_s(R) \subseteq \mathcal{F}_{K_{\mathrm{src}},\boldsymbol{\lambda}}(\sigma^2).$$

*Moreover,*

$$\sup_{\boldsymbol{\theta} \in \Theta_s(R)} d^\dagger(\sigma^2; \boldsymbol{\theta}, \boldsymbol{\lambda}) \asymp K_{\mathrm{src}}, \tag{34}$$

*where the constants depend only on $s$ and $\beta$.*

*Proof of Proposition E.6.* We first prove the embedding. For any $\boldsymbol{\theta} \in \Theta_s(R)$ and any $k \in [d]$,

$$
\begin{aligned}
H_{\boldsymbol{\theta},\boldsymbol{\lambda}}(k) &= \frac{1}{k} \sum_{i=k+1}^{d} \theta_i^2 \\
&= \frac{1}{k} \sum_{i=k+1}^{d} i^{-s\beta} i^{s\beta} \theta_i^2 \\
&\le \frac{k^{-s\beta}}{k} \sum_{i=k+1}^{d} i^{s\beta} \theta_i^2 \\
&\le Rk^{-(1+s\beta)}.
\end{aligned}
$$

Let

$$A_\sigma = \left(\frac{R}{\sigma^2}\right)^{1/(1+s\beta)}.$$

If $K_{\mathrm{src}} < d$, then

$$K_{\mathrm{src}} = \lceil A_\sigma \rceil \ge A_\sigma,$$

and hence

$$H_{\boldsymbol{\theta},\boldsymbol{\lambda}}(K_{\mathrm{src}}) \le RK_{\mathrm{src}}^{-(1+s\beta)} \le RA_\sigma^{-(1+s\beta)} = \sigma^2.$$

Therefore,

$$d^\dagger(\sigma^2; \boldsymbol{\theta}, \boldsymbol{\lambda}) \le K_{\mathrm{src}}.$$

Note that this inequality also holds trivially when $K_{\mathrm{src}} = d$. Therefore, the upper bound in Equation (34) holds and we have

$$\Theta_s(R) \subseteq \mathcal{F}_{K_{\mathrm{src}},\boldsymbol{\lambda}}(\sigma^2).$$

The rest is devoted to proving the matching lower bound in Equation (34), i.e., there exists some $\boldsymbol{\theta} \in \Theta_s(R)$ such that

$$d^\dagger(\sigma^2; \boldsymbol{\theta}, \boldsymbol{\lambda}) \gtrsim K_{\mathrm{src}},$$

where the constant only depends on $s$ and $\beta$.

If $K_{\mathrm{src}}$ is bounded by an absolute constant, then the claim is immediate up to constants, since the ESD is at least one. Hence we assume below that $K_{\mathrm{src}}$ is sufficiently large.

Specifically, we pick $c > 0$ sufficiently small so that

$$2^{s\beta} c^{1+s\beta} \leq 1.$$

Define

$$L_\sigma = \min\{d - 1, A_\sigma\},$$

with the convention that $d - 1 = \infty$ when $d = \infty$. In the nontrivial regime where $K_{\mathrm{src}}$ is sufficiently large,

$$L_\sigma \asymp K_{\mathrm{src}}, \text{ and } cL_\sigma \geq 1.$$

Set

$$m = \lfloor cL_\sigma \rfloor.$$

We have $m \geq 1$, $m + 1 \leq 2cL_\sigma$, and $m + 1 \leq d$. Define

$$\theta_{m+1}^{*2} = m\sigma^2, \text{ and } \qquad \theta_i^* = 0, \quad i \neq m + 1.$$

We first check that $\boldsymbol{\theta}^* \in \Theta_s(R)$. Its source norm is

$$\sum_{i=1}^d i^{s\beta} \theta_i^{*2} = (m + 1)^{s\beta} m\sigma^2.$$

Since $m \leq cL_\sigma$ and $m + 1 \leq 2cL_\sigma$, we have

$$\begin{aligned}
(m + 1)^{s\beta} m\sigma^2 &\leq (2cL_\sigma)^{s\beta}(cL_\sigma)\sigma^2 \\
&= 2^{s\beta} c^{1+s\beta} L_\sigma^{1+s\beta} \sigma^2 \\
&\leq 2^{s\beta} c^{1+s\beta} A_\sigma^{1+s\beta} \sigma^2 \\
&= 2^{s\beta} c^{1+s\beta} R.
\end{aligned}$$

We obtain

$$\sum_{i=1}^d i^{s\beta} \theta_i^{*2} \leq R.$$

Thus $\boldsymbol{\theta}^* \in \Theta_s(R)$.

Next, we compute the ESD of $\boldsymbol{\theta}^*$. For every $k \leq m$, the only nonzero coordinate $m + 1$ lies in the tail after $k$, so

$$H_{\boldsymbol{\theta}^*, \boldsymbol{\lambda}}(k) = \frac{1}{k} \sum_{i=k+1}^d \theta_i^{*2} = \frac{m\sigma^2}{k}.$$

Thus

$$H_{\boldsymbol{\theta}^*, \boldsymbol{\lambda}}(m) = \sigma^2,$$

whereas for every $k < m$,

$$H_{\boldsymbol{\theta}^*, \boldsymbol{\lambda}}(k) = \frac{m\sigma^2}{k} > \sigma^2.$$

For every $k \geq m + 1$, the tail after $k$ is zero:

$$H_{\boldsymbol{\theta}^*, \boldsymbol{\lambda}}(k) = 0.$$

By the definition of ESD,

$$d^\dagger(\sigma^2; \boldsymbol{\theta}^*, \boldsymbol{\lambda}) = m.$$

Since

$$m = \lfloor cL_\sigma \rfloor \asymp L_\sigma \asymp K_{\mathrm{src}},$$

we have

$$\sup_{\boldsymbol{\theta} \in \Theta_s(R)} d^\dagger(\sigma^2; \boldsymbol{\theta}, \boldsymbol{\lambda}) \geq d^\dagger(\sigma^2; \boldsymbol{\theta}^*, \boldsymbol{\lambda}) \asymp K_{\mathrm{src}}.$$

Combining this lower bound with the upper bound proves

$$\sup_{\boldsymbol{\theta} \in \Theta_s(R)} d^\dagger(\sigma^2; \boldsymbol{\theta}, \boldsymbol{\lambda}) \asymp K_{\mathrm{src}}.$$

$\square$

### E.5. Details of Example 4.4

Let $f(x) = \sigma_0^2 x e^{-x^b}$. Then $(\theta_{j+1}^*)^2 = f(j) - f(j+1)$ for $j \geq 1$. Since $\theta_1^* = 0$, for any $k \geq 1$, the tail sum is

$$\sum_{j=k+1}^{\infty} (\theta_j^*)^2 = \sum_{j=k}^{\infty}(f(j) - f(j+1)) = f(k) = \sigma_0^2 k e^{-k^b},$$

since $f(N) \to 0$. As $\{\lambda_j\}$ is assumed to be decreasing, the trade-off function is $\mathbf{H}_{\boldsymbol{\theta}^*,\boldsymbol{\lambda}}(k) = \frac{1}{k}\sum_{j=k+1}^{\infty}(\theta_j^*)^2 = \sigma_0^2 e^{-k^b}$.

For any $n \geq 3$, let $k = K_n$. By definition of the ceiling function, $k \geq (\log n)^{1/b}$, which implies $k^b \geq \log n$, and thus $e^{k^b} \geq n$. Then, $\mathbf{H}_{\boldsymbol{\theta}^*,\boldsymbol{\lambda}}(k) = \sigma_0^2 e^{-k^b} \leq \sigma_0^2/n$. By Proposition 3.7, we have $\mathbf{D}_{\boldsymbol{\theta}^*,\boldsymbol{\lambda}}(\sigma_0^2/n) \leq k = K_n$.

Since this holds for all sufficiently large $n$, we conclude that $\boldsymbol{\theta}^* \in \mathcal{F}_K$. Theorem 4.3 guarantees the optimal convergence rate is $\Theta(\sigma_0^2 K_n/n) = \Theta(\sigma_0^2 (\log n)^{1/b}/n)$.

Lastly, we consider the standard source condition that for some $s > 0$, there is some constant $R_s$ such that

$$\sum_{j=1}^{\infty} \lambda_j^{-s}(\theta_j^*)^2 \leq R_s. \tag{35}$$

Assume a polynomial eigenvalue decay $\lambda_j \asymp j^{-\beta}$ for some $\beta > 0$. Let $S$ be the left-hand side of Equation (35). Since $\theta_1^* = 0$, we have

$$S = \sum_{j=2}^{\infty}(j^{-\beta})^{-s}(\theta_j^*)^2 = \sum_{j=2}^{\infty} j^{s\beta}(\theta_j^*)^2$$

$$= \sum_{k=1}^{\infty}(k+1)^{s\beta}(\theta_{k+1}^*)^2.$$

Using $(\theta_{k+1}^*)^2 = f(k) - f(k+1)$ with $f(x) = \sigma_0^2 x e^{-x^b}$, we obtain

$$S = \sum_{k=1}^{\infty}(k+1)^{s\beta}(f(k) - f(k+1)).$$

Using summation by parts, we have

$$S = (1+1)^{s\beta}f(1) - \lim_{N \to \infty}(N+1)^{s\beta}f(N+1) + \sum_{k=1}^{\infty}((k+2)^{s\beta} - (k+1)^{s\beta})f(k+1).$$

Since $\lim_{N \to \infty}(N+1)^{s\beta}Ne^{-N^b} = 0$ for $b \geq 1$, the limit term vanishes. $f(1) = \sigma_0^2 e^{-1}$. The difference term $(k+2)^{s\beta} - (k+1)^{s\beta} > 0$. $f(k+1) = \sigma_0^2(k+1)e^{-(k+1)^b} > 0$. The sum $\sum_{k=1}^{\infty}((k+2)^{s\beta} - (k+1)^{s\beta})f(k+1)$ converges

because $f(k+1)$ decays faster than any polynomial grows. Specifically, $(k+2)^{s\beta} - (k+1)^{s\beta} \approx s\beta k^{s\beta-1}$, and the sum $\sum k^{s\beta-1}(k+1)e^{-(k+1)^b}$ converges. Therefore, $S$ converges for any $s > 0$ and any $\beta > 0$.

Given this source condition, the classical theory yields the minimax optimal rate at the order $n^{-\frac{s\beta}{s\beta+1}}$. Furthermore, the source condition holds for arbitrarily large $s$. By taking $\gamma = s\beta$, one can see that the classical convergence rate for this particular signal can be expressed as $n^{-\frac{\gamma}{1+\gamma}}$ for any $\gamma > 0$. However, this rate ignores the logarithmic factor $(\log n)^{1/b}$ present in the true optimal rate $\Theta(\sigma_0^2(\log n)^{1/b}/n)$. Thus, the traditional convergence analysis based on the source condition is not sharp for this signal.

# F. Proofs for Results in Appendix C

*Proof of Proposition C.4.* **Upper bound**: Take $k = d^\dagger$, and we have $B(k) = k\mathbf{H}_{f^*,\lambda}(k) \le k\sigma^2$. The variance $V(k) = \sum_{j=1}^{k}(\sigma_0^2 + \tau_j^2)/n \le k(\sigma_0^2 + \|f^*\|_\infty^2)/n = k\sigma^2$. Thus $\mathcal{R}_*^{\mathrm{PC}} \le \mathcal{R}_k = B(k) + V(k) \le 2k\sigma^2 = 2d^\dagger\sigma^2$.

**Lower bound**: Let $k^*$ be the optimal tuning parameter. If $k^* \ge d^\dagger$, then $\mathcal{R}_* \ge d^\dagger\sigma_0^2/n$. If $k^* \le d^\dagger - 1$, by definition of ESD, we have $\mathcal{R}_* \ge B(k^*) \ge B(d^\dagger - 1) \ge (d^\dagger - 1)\sigma^2 \ge (d^\dagger - 1)\sigma_0^2/n$.

$\square$

*Proof of Theorem C.7.*
**Upper bound**: Use the KPCPE estimator retaining the top $K$ eigenfunctions, with the transformed coefficients $z_j$ defined in Equation (21). Write this estimator as

$$\hat{f}_K^{\mathrm{PC}}(x) := \sum_{i \le K} z_{\pi_i}\phi_{\pi_i}(x).$$

The cutoff $K$ is the class parameter in $\mathcal{F}_{K,\mathbf{k}}^{(\sigma^2)}$ and does not depend on the unknown target. For any $f^* \in \mathcal{F}_{K,\mathbf{k}}^{(\sigma^2)}$, the class definition gives

$$d^\dagger(\bar{\sigma}^2/n; f^*, \mathbf{k}) \le K,$$

and thus

$$\mathbf{H}_{\boldsymbol{\theta}^*,\boldsymbol{\lambda}}(K) \le \bar{\sigma}^2/n.$$

Moreover, $\|f^*\|_\infty^2 \le \sigma_0^2 C_0^2$, so Equation (24) gives $V(K) \le K\bar{\sigma}^2/n$. The risk of this fixed-cutoff KPCPE estimator is the sum of its squared tail bias and its variance:

$$\mathcal{R}(\hat{f}_K^{\mathrm{PC}}; f^*) = B(K) + V(K)$$

$$\le K\mathbf{H}_{\boldsymbol{\theta}^*,\boldsymbol{\lambda}}(K) + K\frac{\bar{\sigma}^2}{n} \le 2\bar{\sigma}^2\frac{K}{n}.$$

Taking the supremum over $f^* \in \mathcal{F}_{K,\mathbf{k}}^{(\sigma^2)}$ gives the upper bound, which is a constant multiple of $\sigma_0^2 K/n$ because $\bar{\sigma}^2 = \sigma_0^2(1 + C_0^2)$.

**Lower bound**: We establish the lower bound using Assouad's method. The lower bound is proved by restricting to Gaussian noise within the model class.

Let $m = \lfloor c_1 K \rfloor$. Consider the first $m$ eigenfunctions $\{\phi_{\pi_j}\}_{j \le m}$ corresponding to the largest eigenvalues $\{\lambda_{\pi_j}\}_{j \le m}$. Define the collection of hypercube vertices $\mathcal{V} = \{-1, 1\}^m$. For every vertex $v \in \mathcal{V}$, define a function

$$f^{(v)}(x) = \gamma \sum_{j=1}^{m} v_j\phi_{\pi_j}(x), \tag{36}$$

where the amplitude $\gamma$ is to be chosen. Since $\mathbf{k}$ is $(K, n)$-regular, we have

$$f^{(v)}(x)^2 \le \gamma^2 \sum_{j \le m} \lambda_{\pi_j}^{-1} \sum_{j \le m} \lambda_j\phi_{\pi_j}^2(x) \le \gamma^2 C_1 n\kappa^2, \tag{37}$$

where $\kappa^2 = \sup_x \mathbf{k}(x, x) < \infty$ by assumption.

We choose

$$\gamma^2 = n^{-1} \min\left(\frac{\bar{\sigma}^2}{4(1+C_0^2)}, \frac{\sigma_0^2 C_0^2}{C_1 \kappa^2}\right).$$

It then follows that $\|f^{(v)}\|_\infty^2 \leq \sigma_0^2 C_0^2$.

For each $v \in \mathcal{V}$, let $P_v$ be the sampling distribution of $\{(x_i, y_i)\}_{i \leq n}$ from the regression model Equation (19) with $f^* = f^{(v)}$. Let $\rho$ be the Hamming distance on $\mathcal{V}$. If $v$ and $w \in \mathcal{V}$ differ in exactly one coordinate (i.e., $\rho(v, w) = 1$), then

- $\|f^{(v)} - f^{(w)}\|_{L^2(\mu)}^2 \geq (2\gamma)^2$, and

- the Kullback-Leibler divergence between $P_v$ and $P_w$ satisfies $\mathrm{KL}(P_v \| P_w) = \frac{n}{2\sigma_0^2}(2\gamma)^2 \leq \frac{1}{2}$, where the last inequality is due to the definition of the constant $c$. By Pinsker's inequality, $\|P_v \wedge P_w\| = 1 - \mathrm{TV}(P_v, P_w) \geq 1 - \sqrt{\mathrm{KL}(P_v \| P_w)/2} = 1/2$.

By Assouad's Lemma (Lemma 2 in Yu (1997)), for any estimator $\widehat{f}$ based on a sample $\{(x_i, y_i)\}_{i \leq n}$ drawn from $P_v$, we have

$$\sup_{v \in \mathcal{V}} \mathbb{E}_v \|\widehat{f} - f^{(v)}\|_{L^2(\mu)}^2 \geq m\frac{(2\gamma)^2}{4} = c\frac{\sigma_0^2 K}{n},$$

where $c$ is a constant that depends on $C_0$, $\kappa$, $c_1$, and $C_1$. $\qquad\square$

*Proof of Theorem C.8.* For the upper bound at the fixed sample size $n$, use the KPCPE estimator retaining the top $K_n$ eigenfunctions, with the transformed coefficients $z_j$ defined in Equation (21). Write this estimator as

$$\hat{f}_{K_n}^{\mathrm{PC}}(x) := \sum_{i \leq K_n} z_{\pi_i} \phi_{\pi_i}(x).$$

The cutoff $K_n$ is the class parameter at sample size $n$ and does not depend on the unknown target. For any $f^* \in \mathcal{F}_{\mathbf{K},\mathbf{k}}$, the definition of $\mathcal{F}_{\mathbf{K},\mathbf{k}}$ gives

$$d^\dagger(\bar{\sigma}^2/n; f^*, \mathbf{k}) \leq K_n,$$

and thus

$$\mathbf{H}_{\boldsymbol{\theta}^*, \boldsymbol{\lambda}}(K_n) \leq \bar{\sigma}^2/n.$$

Moreover, $\|f^*\|_\infty^2 \leq \sigma_0^2 C_0^2$, so Equation (24) gives $V(K_n) \leq K_n \bar{\sigma}^2/n$. The risk of this fixed-cutoff KPCPE estimator is the sum of its squared tail bias and its variance:

$$\mathcal{R}(\hat{f}_{K_n}^{\mathrm{PC}}; f^*) = B(K_n) + V(K_n)$$

$$\leq K_n \mathbf{H}_{\boldsymbol{\theta}^*, \boldsymbol{\lambda}}(K_n) + K_n \frac{\bar{\sigma}^2}{n} \leq 2\bar{\sigma}^2 \frac{K_n}{n}.$$

Taking the supremum over $f^* \in \mathcal{F}_{\mathbf{K},\mathbf{k}}$ gives the upper bound, which is a constant multiple of $\sigma_0^2 K_n/n$ because $\bar{\sigma}^2 = \sigma_0^2(1 + C_0^2)$.

The lower bound follows the same argument as in the proof of Theorem 4.3, but replace the construction of parameter vectors in Equation (32) by the construction of functions in Equation (36). Following the proof for Theorem 4.3, we use Condition 4.1 to ensure the constructed functions all belong to $\mathcal{F}_{\mathbf{K},\mathbf{k}}$. Then the lower bound is given using Assouad's Lemma as in the proof of Theorem C.7. Below, we provide the details for completeness.

Mercer's theorem yields

$$\mathbf{k}(\boldsymbol{x}, \boldsymbol{x}') = \sum_{j=1}^\infty \lambda_j \phi_j(\boldsymbol{x}) \phi_j(\boldsymbol{x}'), \quad \boldsymbol{x}, \boldsymbol{x}' \in \mathcal{X}, \tag{38}$$

where $\{\phi_j\}_{j \geq 1}$ is an $L^2(\mathcal{X}, \mu)$-orthonormal eigenbasis. Without loss of generality, assume the sequence $\{\lambda_j\}$ is sorted in a decreasing order.

Fix $n$ and set $m := \lfloor c_1 K_n \rfloor$ where $c_1$ comes from Assumption C.5.

For a sign vector $v = (v_j)_{j \leq m} \in \{-1, +1\}^m$, define the sequence of coefficients as

$$\theta_j^{(v)} := \begin{cases} \gamma\, v_j, & j \leq m, \\ 0, & j > m, \end{cases} \qquad f_v(x) := \sum_{j \geq 1} \theta_j^{(v)}\, \phi_j(x) = \gamma \sum_{j \leq m} v_j\, \phi_j(x).$$

Since $\mathbf{k}$ is $(K_n, n)$-regular, Equation (37) holds and reads as

$$f^{(v)}(x)^2 \leq \gamma^2 C_1 n \kappa^2.$$

If $\gamma^2 C_1 n \kappa^2 \leq \sigma_0^2 C_0^2$, then $\|f^{(v)}\|_\infty^2 \leq \sigma_0^2 C_0^2$. Furthermore, if $m\gamma^2 \leq (2n)^{-1} \sigma_0^2 K_n$, we can use the same argument in Lemma E.2 (in particular, using Condition 4.1 to derive Equation (33)) to show that $f^{(v)} \in \mathcal{F}_{\mathbf{K},\mathbf{k}}$.

We choose

$$\gamma^2 = n^{-1} \min\left( \sigma_0^2, \frac{\bar{\sigma}^2}{4(1 + C_0^2)}, \frac{\sigma_0^2 C_0^2}{C_1 \kappa^2} \right),$$

which implies $f^{(v)} \in \mathcal{F}_{\mathbf{K},\mathbf{k}}$.

We then follow the same argument in the proof of lower bound in Theorem C.7 to obtain

$$\sup_{v \in \mathcal{V}} \mathbb{E}_v \big\| \widehat{f} - f^{(v)} \big\|_{L^2(\mu)}^2 \geq m \frac{(2\gamma)^2}{4} = c \frac{\sigma_0^2 K}{n},$$

where $c$ is a constant that depends on $C_0, \kappa, c_1, C_1$.

$\square$

# G. Proofs for Results on Over-Parameterized Gradient Flow

In this section, we prove Theorem 5.2. Before giving the proof, we explain the high-level idea: To show the ESD decreases, it is enough to show that the squared signal tail sorted by the learned eigenvalues at the new time is smaller than that at the old time. The key step is to study how the gradient flow changes the eigenvalues depending on the signal's strength. Our analysis reveals that under the stated conditions, eigenvalues associated with the strong signal coordinates will often grow much faster than those associated with weak ones. Consequently, more of the largest learned eigenvalues correspond to the strong signals. This implies a reduction of the signal tail sorted by the learned kernel and hence an ESD reduction.

The proof below uses the scalar flow

$$\begin{aligned} \dot{a}_j &= -\nabla_{a_j} L_j = b_j^D \beta_j (z_j - \theta_j), \\ \dot{b}_j &= -\nabla_{b_j} L_j = D a_j b_j^{D-1} \beta_j (z_j - \theta_j), \\ \dot{\beta}_j &= -\nabla_{\beta_j} L_j = a_j b_j^D (z_j - \theta_j), \\ a_j(0) &= \lambda_j^{\frac{1}{2}} > 0, \quad b_j(0) = b_0 > 0, \quad \beta_j(0) = 0, \end{aligned} \tag{39}$$

where $L_j = \frac{1}{2}(z_j - \theta_j)^2$. We write $\ln^+(x) = \ln(\max(e, x))$ for any $x \geq 0$.

This section is organized as follows. First, we record the concentration event and two elementary ODE comparison lemmas. Next, we prove Theorem 5.2 by reducing it to endpoint ordering relations and a top-$m$ set exchange. We then collect the scalar conservation identities and one-coordinate flow estimates used later. Finally, we give the deferred proofs of the endpoint ordering lemmas.

## G.1. Concentration and Elementary ODE Comparison Lemmas

Lemma G.1 defines the concentration event $\mathcal{E}$ used in the proof of Theorem 5.2.

**Lemma G.1.** *Recall the set $S$ defined in Assumption 5.1 and let $C = 2C_{proxy}^{1/2}$. For $k \in S$, we introduce the events $\{E_k\}$ as follows:*

$$E_k := \{ |\xi_k| \leq C n^{-1/2} \sqrt{\ln n} \}. \tag{40}$$

*For $k \in S^c$, we introduce the events $\{E_k\}$ as follows:*

$$E_k := \left\{ |\xi_k| \leq Cn^{-1/2}\sqrt{\ln(en\tilde{k})} \right\}, \tag{41}$$

*where $\tilde{k} := \tilde{d}/\lambda_k$ and $\tilde{d} := \sum_j \lambda_j$.*

*Then, with probability at least $1 - \frac{4}{n}$, all events $E_k, k \in [d]$ hold simultaneously.*

*Proof.* By Assumption 5.1, the noise $\xi_k$ is sub-Gaussian with variance proxy $C_{\text{proxy}}/n$. Therefore, $\mathbf{P}\left(|\xi_k| \geq s\right) \leq 2\exp(-ns^2/(2C_{\text{proxy}}))$.

If $k \in S$, we have

$$\mathbf{P}\left\{ |\xi_k| \geq 2C_{\text{proxy}}^{1/2}\sqrt{\frac{\ln n}{n}} \right\} \leq 2\exp\left(-2(\ln n)\right).$$

By the union bound, we have

$$\begin{aligned}
\mathbf{P}\left\{\cap_{k \in S} E_k\right\} &\geq 1 - \sum_{k \in S} \mathbf{P}\left\{ |\xi_k| \geq 2C_{\text{proxy}}^{1/2}\sqrt{\frac{\ln n}{n}} \right\} \\
&\geq 1 - |S|2\exp\left(-2(\ln n)\right) \\
&\geq 1 - \frac{2}{n},
\end{aligned} \tag{42}$$

where the last inequality follows from $|S|2\exp\left(-2(\ln n)\right) \leq 2n^{-1}$.

If $k \in S^c$, we have

$$\mathbf{P}\left\{ |\xi_k| \geq 2C_{\text{proxy}}^{1/2}\sqrt{\frac{\ln(en\tilde{k})}{n}} \right\} \leq 2\exp\left(-2\ln(en\tilde{k})\right) \leq 2\exp\left(-\ln(en\tilde{k})\right) = \frac{2}{en\tilde{k}} \leq \frac{2}{n} \cdot \frac{\lambda_k}{\sum_j \lambda_j}, \tag{43}$$

where we recall that $\tilde{k} := \tilde{d}/\lambda_k$ and $\tilde{d} := \sum_j \lambda_j$.

By the union bound, we have

$$\begin{aligned}
\mathbf{P}\left\{\cap_{k \in S^c} E_k\right\} &\geq 1 - \sum_{k \in S^c} \mathbf{P}\left\{ |\xi_k| \geq 2C_{\text{proxy}}^{1/2}\sqrt{\frac{\ln(en\tilde{k})}{n}} \right\} \\
&\geq 1 - \sum_{k \in S^c} \frac{2}{n} \cdot \frac{\lambda_k}{\sum_j \lambda_j} \\
&\geq 1 - \frac{2}{n}.
\end{aligned} \tag{44}$$

Combining Equation (42) and Equation (44) gives the result.

$\square$

The following two lemmas provide convenient upper bounds on hitting times of ODE solutions.

**Lemma G.2.** *Let $k > 0$ and $p > 1$.*

- *Consider the ODE*

$$\dot{x} \geq kx^p, \quad x(0) = x_0 > 0$$

*Then we have*

$$x(t) \geq \left( x_0^{-(p-1)} - (p-1)kt \right)^{-\frac{1}{p-1}}$$

*and thus for any $M \geq 0$,*

$$\inf\{t \geq 0 : x(t) \geq M\} \leq \left[ (p-1)kx_0^{p-1} \right]^{-1}. \tag{45}$$

- *Consider the ODE*

$$\dot{x} \leq -kx^p, \quad x(0) = x_0 > 0.$$

*Then we have*

$$x(t) \leq \left( x_0^{-(p-1)} + (p-1)kt \right)^{-\frac{1}{p-1}},$$

*and thus for any $M > 0$,*

$$\inf\{t \geq 0 : x(t) \leq M\} \leq \left[ (p-1)kM^{p-1} \right]^{-1}. \tag{46}$$

**Lemma G.3.** *Let $k > 0$ and $x_0 > 0$.*

1. *If*

$$\dot{x}(t) \geq k\,x(t), \qquad x(0) = x_0,$$

*then for all $t \geq 0$, it holds that*

$$x(t) \geq x_0\,e^{kt},$$

*and for every $M \geq x_0$, we have*

$$\inf\{t \geq 0 : x(t) \geq M\} \leq \frac{1}{k}\ln\!\left(\frac{M}{x_0}\right).$$

2. *If*

$$\dot{x}(t) \leq -k\,x(t), \qquad x(0) = x_0,$$

*then for all $t \geq 0$, it holds that*

$$x(t) \leq x_0\,e^{-kt},$$

*and for every $0 < M \leq x_0$, we have*

$$\inf\{t \geq 0 : x(t) \leq M\} \leq \frac{1}{k}\ln\!\left(\frac{x_0}{M}\right).$$

## G.2. Proof of Theorem 5.2: Endpoint Ordering and Block Exchange

We prove Theorem 5.2 on the concentration event $\mathcal{E} := \cap_{k \in [d]} E_k$ from Lemma G.1. Lemma G.1 gives $\mathbb{P}(\mathcal{E}) \geq 1 - 4/n$. Throughout this subsection we work deterministically on $\mathcal{E}$. Since $\mathcal{E}$ is fixed, all bounds furnished by the events $E_k$ may be used simultaneously and no additional union bound over $t_1$ is needed.

Fix a time $t_1 \in [0, t_2)$ with $d^\dagger(t_1) < \infty$ satisfying the hypotheses of Theorem 5.2. Put

$$m := d^\dagger(t_1), \qquad T_1 := \{i : \pi_{t_1}^{-1}(i) \leq m\}, \qquad T_2 := \{i : \pi_{t_2}^{-1}(i) \leq m\}. \tag{47}$$

$m$ is the ESD cutoff selected by the learned spectrum at time $t_1$. $T_1$ is the set of coordinates occupying the first $m$ positions at time $t_1$, and $T_2$ is the set occupying the first $m$ positions at time $t_2$ when the same cutoff $m$ is used. Define the corresponding tail-energy difference

$$\Delta_m := \sum_{i : \pi_{t_1}^{-1}(i) > m} |\theta_i^*|^2 - \sum_{i : \pi_{t_2}^{-1}(i) > m} |\theta_i^*|^2. \tag{48}$$

Thus $\Delta_m \geq 0$ means that the signal energy outside the fixed cutoff $m$ is no larger at time $t_2$ than at time $t_1$. Equivalently, because the same cutoff $m$ is used on both sides, $\Delta_m \geq 0$ is the desired fixed-cutoff ESD comparison

$$\mathbf{H}_{\boldsymbol{\theta}^*, \tilde{\boldsymbol{\lambda}}(t_2)}(m) \leq \mathbf{H}_{\boldsymbol{\theta}^*, \tilde{\boldsymbol{\lambda}}(t_1)}(m). \tag{49}$$

Indeed, by definition of $m = d^\dagger(t_1)$, $\mathbf{H}_{\boldsymbol{\theta}^*, \tilde{\boldsymbol{\lambda}}(t_1)}(m) \leq \sigma^2$. Thus Equation (49) implies $\mathbf{H}_{\boldsymbol{\theta}^*, \tilde{\boldsymbol{\lambda}}(t_2)}(m) \leq \sigma^2$, so $m$ is feasible for the ESD at time $t_2$, and hence $d^\dagger(t_2) \leq m = d^\dagger(t_1)$.

**Proof structure.** The proof of Theorem 5.2 consists of three stages:

1. **Theorem-local setup and scales.** Define the strong and weak blocks within $T_1$ and outside $T_1$ (namely $A_1, A_2, A_3, B_1, B_2$) and the threshold scales $\tau_A, \tau_w, \Lambda_s$; the tiny weak set $T_{\text{tiny}}$ is introduced in Lemma G.4.

2. **Endpoint ordering relations.** Prove that at time $t_2$, (1) strong tail coordinates have larger learned eigenvalues than small weak top coordinates, (2) strong coordinates are ordered according to their true amplitudes, (3) weak tail coordinates either stay outside $T_2$ or are counted explicitly as weak entrants, and (4) coordinates in $A_3$ have larger learned eigenvalues than coordinates in $A_1$.

3. **Top-$m$ set exchange.** Use the ordering relations at time $t_2$ to compare the top-$m$ sets $T_1$ and $T_2$. The comparison splits into $|B_2| \leq |A_1|$ and $|B_2| > |A_1|$. In both cases the entering strong energy compensates all outgoing weak energy, which implies $\Delta_m \geq 0$.

### G.2.1. Theorem-Local Setup and Scales

The theorem constants are chosen so that $C_M > c'$. Since $M = C_M \varepsilon$ and $\tilde{\sigma} = c' \varepsilon$, this gives $M > \tilde{\sigma}$. Under Conditions (1)–(2) of Theorem 5.2, every coordinate in $S$ is strong, $|\theta_i^*| \geq M$, and every coordinate in $S^c$ is weak, $|\theta_i^*| \leq \tilde{\sigma}$. Because the two thresholds are disjoint, the time-$t_1$ top block $T_1$ is the following disjoint union:

$$\begin{aligned}
A_1 &:= \{i \in T_1 : \tilde{\lambda}_i(t_1) < c b_0^{2D} D^{-D/(D+2)} M^{2/(D+2)}, \ |\theta_i^*| \leq \tilde{\sigma}\}, \\
A_2 &:= \{i \in T_1 : |\theta_i^*| \geq M\}, \\
A_3 &:= \{i \in T_1 : |\theta_i^*| \leq \tilde{\sigma}\} \setminus A_1.
\end{aligned} \tag{50}$$

Their interpretations are:

1. $A_1$ contains the weak coordinates already in the time-$t_1$ top block whose learned eigenvalues are below the displayed $M$-scale threshold. These are the small weak top coordinates that may be displaced by strong tail coordinates.

2. $A_2$ contains the strong coordinates already in the time-$t_1$ top block.

3. $A_3$ contains the remaining weak coordinates in the time-$t_1$ top block; equivalently, these are weak top coordinates whose learned eigenvalues at time $t_1$ are not as small as those in $A_1$.

The time-$t_1$ tail is also a disjoint union:

$$B_1 := \{i \notin T_1 : |\theta_i^*| \leq \tilde{\sigma}\},$$
$$B_2 := \{i \notin T_1 : |\theta_i^*| \geq M\}. \tag{51}$$

$B_1$ contains the weak coordinates outside the time-$t_1$ top block. $B_2$ contains the strong coordinates outside the time-$t_1$ top block; when such coordinates enter $T_2$, their signal energy offsets outgoing weak coordinates in the final comparison of $T_1$ and $T_2$. Therefore

$$A_2 \cup B_2 \subseteq S, \qquad A_1 \cup A_3 \cup B_1 \subseteq S^c.$$

We refer to the four clauses inside Condition (4) of Theorem 5.2 as Conditions (4-i)–(4-iv). The local definitions below identify the sets used in Condition (4). Since $m = d^\dagger(t_1)$ and

$$T_1 = \{i : \pi_{t_1}^{-1}(i) \leq d^\dagger(t_1)\},$$

the present set $A_1$ is exactly the $A_1$ in Condition (4), namely

$$\{i \in S^c : \pi_{t_1}^{-1}(i) \leq d^\dagger(t_1), \ \tilde{\lambda}_i(t_1) < c b_0^{2D} D^{-D/(D+2)} M^{2/(D+2)}\}.$$

Similarly, the present $B_1$ and $B_2$ are exactly the two tail sets in Condition (4):

$$B_1 = \{i \in S^c : \pi_{t_1}^{-1}(i) > d^\dagger(t_1)\},$$
$$B_2 = \{i \in S : \pi_{t_1}^{-1}(i) > d^\dagger(t_1)\}. \tag{52}$$

The sets $A_2$ and $A_3$ are proof-only refinements of the time-$t_1$ top block: $A_2$ is the strong part of $T_1$, and $A_3$ is the weak part of $T_1$ that is not in $A_1$.

We write

$$C_{B_1} := \min\{(|A_1| - |B_2|)_+, |B_1|\}.$$

Condition (4-iv) of Theorem 5.2 gives the weak-coordinate energy-compensation inequality

$$(|B_2| + C_{B_1})\tilde{\sigma}^2 \leq |B_2|M^2. \tag{53}$$

We use the following threshold scales throughout the proof:

$$\begin{aligned}
s_M &:= D^{-D/(D+2)} M^{2/(D+2)}, \\
\tau_A &:= c b_0^{2D} s_M = c b_0^{2D} D^{-D/(D+2)} M^{2/(D+2)}, \\
\tau_w &:= c b_0^{2D} D^{-D/(D+2)} \varepsilon^{2/(D+2)}, \\
\Lambda_s &:= c_s D^{D/(D+2)} M^{2(D+1)/(D+2)}, \\
U_w &:= C_w D^{-D/(D+2)} (2c'\varepsilon)^{2/(D+2)}.
\end{aligned} \tag{54}$$

The constant $c_s$ depends only on $D$. We choose $c_s > 0$ so that

$$c_s \leq (3/4)^{2(D+1)/(D+2)}. \tag{55}$$

This is the constant used in the strong endpoint lower bound in Equation (59). The constant $C_w \geq 1$, depending only on $D$ and the one-dimensional flow estimate Equation (100), is chosen so that Equation (100) gives

$$|\theta(t)| \leq 2c'\varepsilon \implies \beta^2(t) \leq C_w D^{-D/(D+2)} (2c'\varepsilon)^{2/(D+2)}.$$

### G.2.2. ENDPOINT ORDERING RELATIONS AT TIME $t_2$

Lemmas G.4 to G.9 prove the learned-eigenvalue ordering relations at time $t_2$ that are summarized in Proposition G.10 and used in Proposition G.11. For two coordinate sets $U$ and $V$, write $U \succ_{t_2} V$ if every coordinate in $U$ has larger learned eigenvalue at time $t_2$ than every coordinate in $V$. In Lemmas G.4 to G.9, we keep the following setup fixed: the concentration event $\mathcal{E}$ holds, $t_1$ satisfies the hypotheses of Theorem 5.2, Conditions (1)–(4) of Theorem 5.2 are in force, and the sets and scales are those in Equations (47), (50), (51) and (54). The displayed assumptions inside each lemma list the additional scale and large-sample inequalities used in that lemma.

**Lemma G.4** (Tiny weak coordinates). *Assume the concentration event $\mathcal{E}$ and Condition (2) of Theorem 5.2. Choose once and for all $\delta_{\text{tiny}} \in ((D+1)/(D+2), 1)$. There exists a constant $K_{\text{tiny}}$, independent of $n$, such that, after possibly increasing $n_0$, for every $n \geq n_0$, with*

$$T_{\text{tiny}} := \{i \in S^c : \lambda_i < n^{-K_{\text{tiny}}}\},$$

*every $i \in T_{\text{tiny}}$ satisfies*

$$\tilde{\lambda}_i(t_1) \vee \tilde{\lambda}_i(t_2) < n^{-\delta_{\text{tiny}}}.$$

*The same choice of $n_0$ can also be made to satisfy*

$$n^{-\delta_{\text{tiny}}} < \frac{1}{2}\min\{\tau_A, \tau_w, \Lambda_s\}. \tag{56}$$

*For all non-tiny weak coordinates $i \in S^c \setminus T_{\text{tiny}}$,*

$$|\xi_i| \leq \kappa_w := 2C_{\text{proxy}}^{1/2} n^{-1/2}\sqrt{\ln(en\tilde{d}\, n^{K_{\text{tiny}}})},$$

*and the same choice of $n_0$ can be made so that $\kappa_w \leq c'\varepsilon$.*

In the rest of this subsection, $T_{\text{tiny}}$ denotes the set defined in Lemma G.4, and the conclusions of Lemma G.4 are used whenever $T_{\text{tiny}}$ appears.

After the constants appearing in Lemmas G.5 to G.9 and the constant $K_{\text{tiny}}$ in Lemma G.4 are fixed, the sample-size threshold $n_0$ is enlarged so that, for every $n \geq n_0$,

$$M \geq 8\varepsilon', \qquad \kappa_w \leq c'\varepsilon. \tag{57}$$

The inequality $M \geq 8\varepsilon'$ follows from

$$\frac{M}{\varepsilon'} = C_M \frac{\varepsilon}{\varepsilon'} = C_M \sqrt{\ln(e + n\tilde{d})} \to \infty,$$

while $\kappa_w \leq c'\varepsilon$ is the large-sample conclusion in Lemma G.4.

**Lemma G.5** (Strong endpoint scale). *Assume the following large-sample and constant inequalities:*

$$M \geq 8\varepsilon', \qquad t_2 \geq t(\varepsilon), \qquad c_s \leq (3/4)^{2(D+1)/(D+2)}, \qquad (c+\eta)(\bar{c}_b + \eta)^D < c_s. \tag{58}$$

*Suppose $k \in S$ and $|\theta_k^*| \geq M$. Then*

$$\tilde{\lambda}_k(t_2) \geq \Lambda_s := c_s D^{D/(D+2)} M^{2(D+1)/(D+2)}. \tag{59}$$

*Furthermore, if $\lambda_k < cs_M$ and $D^{-1}b_0^2 \leq \bar{c}_b s_M$, then*

$$\beta_k^2(t_2) > \eta s_M. \tag{60}$$

*Under the same additional hypotheses,*

$$\frac{D^{-1}b_0^2}{\beta_k^2(t_2)} < \frac{\bar{c}_b}{\eta}. \tag{61}$$

**Lemma G.6** (Strong tail is ordered above small weak top). *With $T_{\text{tiny}}$ fixed as in Lemma G.4, assume Equation (57), Equation (58), and*

$$t_2 \geq t(\varepsilon), \qquad C_M > 4C_D c', \qquad D^{-1}b_0^2 \leq \bar{c}_b s_M,$$

$$\rho_D \in (1, C_D^2), \qquad \frac{1 + \bar{c}_b/\eta}{C_D^{-2} + \bar{c}_b/\eta} \geq \rho_D, \qquad c < c_1 := \eta(\rho_D^D - C_D^{-2}). \tag{62}$$

*For every $j \in B_2$ and every non-tiny $i \in A_1 \setminus T_{\text{tiny}}$,*

$$\tilde{\lambda}_j(t_2) > \tilde{\lambda}_i(t_2).$$

**Lemma G.7** (Strong pairwise ordering inside $A_2 \cup B_2$). *Assume the large-sample bounds in Equation* (57) *and the following constant inequalities:*

$$t_2 \geq t(\varepsilon), \qquad c_s \leq (3/4)^{2(D+1)/(D+2)}, \qquad (c+\eta)(\bar{c}_b + \eta)^D < c_s,$$

$$\frac{1}{C_D^{-2} + L^{-1}} > \frac{1}{C^*}, \qquad C^* := \frac{1}{2} C_D^{2D}, \tag{63}$$

$$\left(\frac{1+r_0}{C_D^{-2} + r_0}\right)^D \geq C^*, \qquad c \leq \eta/L, \qquad \bar{c}_b/\eta \leq r_0, \qquad D^{-1} b_0^2 \leq \bar{c}_b s_M.$$

*Assume also the following two large-sample comparison inequalities:*

$$\frac{C_\eta \varepsilon - 4\varepsilon'}{C_{\max} C_M \varepsilon + 2\varepsilon'} \geq \gamma_D,$$

$$\frac{(1 + c_\eta/D) C_M \varepsilon - 2\varepsilon'}{C_M \varepsilon + 2\varepsilon'} \geq C_D. \tag{64}$$

*Let $i \in A_2$ and $j \in B_2$. If $|\theta_i^*| > |\theta_j^*|$, then*

$$\tilde{\lambda}_i(t_2) > \tilde{\lambda}_j(t_2).$$

**Lemma G.8** (Moderate weak top is ordered above small weak top). *With $T_{\text{tiny}}$ fixed as in Lemma G.4, assume Equation* (57), *Equation* (58), *and*

$$0 < \delta_{\text{w}} < 1, \qquad (cs_M + U_w)(b_0^2 + DU_w)^D \leq (1 + \delta_{\text{w}}/2)cs_M b_0^{2D},$$

$$\Lambda_{\text{s}} > 2\tau_A. \tag{65}$$

*Then*

$$\tilde{\lambda}_j(t_2) \geq (1 + \delta_{\text{w}})\tau_A \qquad \text{for every } j \in A_3, \tag{66}$$

*and*

$$A_3 \succ_{t_2} A_1, \qquad A_2 \cup B_2 \succ_{t_2} A_1.$$

**Lemma G.9** (Weak tail exclusion: $B_1$). *With $T_{\text{tiny}}$ fixed as in Lemma G.4, assume Equation* (57), *Equation* (58), *and*

$$0 < \delta_{\text{w}} < 1, \qquad U_w \leq b_0^2/D,$$

$$U_w \left(\frac{2^D b_0^{2D}}{\tau_w} + \frac{D^2}{b_0^2}\right) \leq \ln(1 + \delta_{\text{w}}/2), \tag{67}$$

$$\left(cD^{-D/(D+2)}\varepsilon^{2/(D+2)} + U_w\right)(b_0^2 + DU_w)^D < \min\{\Lambda_{\text{s}}/2, (1 + \delta_{\text{w}}/2)\tau_A\}.$$

*Let $j \in B_1 \setminus T_{\text{tiny}}$. Then the following implications hold.*

1. *If $\tilde{\lambda}_j(t_1) \geq \tau_w$ and*

$$\min_{\ell \in T_1} \tilde{\lambda}_\ell(t_1) \geq (1 + \delta_{\text{w}})\tilde{\lambda}_j(t_1),$$

   *then $j \notin T_2$.*

2. *If $\tilde{\lambda}_j(t_1) < \tau_w$, then*

$$\tilde{\lambda}_j(t_2) < \min\{\Lambda_{\text{s}}/2, (1 + \delta_{\text{w}}/2)\tau_A\}.$$

   *Thus $A_2 \cup B_2 \cup A_3 \succ_{t_2} \{j\}$.*

The proofs of Lemmas G.4 to G.9 are deferred to Appendix G.6, after the auxiliary dynamic estimates used in those proofs are established.

**Feasibility of the displayed ordering assumptions.** We now choose the constants so that the displayed assumptions in Equations (58), (62) to (65) and (67) hold simultaneously. The constants $c_s$ and $C_w$ have already been fixed after Equation (54). Choose $\delta_w \in (0, 1)$, and choose $c_B > 0$ small. Choose

$$C_D = 1 + \gamma_D > 2^{1/(2D+2)}, \qquad C^* := \frac{1}{2} C_D^{2D},$$

then choose $\rho_D \in (1, C_D^2)$. Choose $L$ large enough and $r_0 > 0$ small enough so that

$$\frac{1}{C_D^{-2} + L^{-1}} > \frac{1}{C^*}, \qquad \left( \frac{1 + r_0}{C_D^{-2} + r_0} \right)^D \geq C^*.$$

The second inequality is feasible because the left-hand side converges to $C_D^{2D} > C^*$ as $r_0 \downarrow 0$.

Choose $\eta > 0$ so that $2^{D+1} \eta^{D+1} < c_s$. Then choose $\bar{c}_b > 0$ small enough that

$$\frac{1 + \frac{\bar{c}_b}{\eta}}{C_D^{-2} + \frac{\bar{c}_b}{\eta}} \geq \rho_D, \qquad \frac{\bar{c}_b}{\eta} \leq r_0, \qquad \bar{c}_b \leq \eta.$$

Set $c_1 := \eta(\rho_D^D - C_D^{-2})$, and choose $c > 0$ so that

$$c < c_1, \qquad c \leq \frac{\eta}{L}, \qquad c \leq \eta.$$

These choices give the constant inequalities among those displayed in Equations (58), (62) and (63) that are independent of $c', C_M, C_{\max}, c_\eta, C_\eta$, the horizon constant $C$, and the final sample-size threshold $n_0$.

Next choose $c' > 0$ small enough so that the two inequalities

$$U_w \leq \frac{b_0^2}{D}, \qquad U_w \left( \frac{2^D b_0^{2D}}{\tau_w} + \frac{D^2}{b_0^2} \right) \leq \ln(1 + \delta_w/2)$$

in Equation (67) hold. After $c'$ is fixed, choose $C_M$ large enough so that $C_M > c'$, $C_M > 4C_D c'$, the inequality $D^{-1} b_0^2 \leq \bar{c}_b s_M$ in Equations (62) and (63) holds, and the scale comparisons in Equations (65) and (67) hold:

$$\Lambda_s > 2\tau_A,$$

$$(c s_M + U_w)(b_0^2 + D U_w)^D \leq (1 + \delta_w/2) c s_M b_0^{2D},$$

and

$$\left( c D^{-D/(D+2)} \varepsilon^{2/(D+2)} + U_w \right) (b_0^2 + D U_w)^D < \min\{\Lambda_s/2, (1 + \delta_w/2)\tau_A\}.$$

Indeed, after substituting $b_0 = c_B D^{(D+1)/(D+2)} \varepsilon^{1/(D+2)}$ and $M = C_M \varepsilon$, these comparisons reduce to inequalities involving only the fixed constants $D, c_B, c, c', C_M, c_s, C_w, \bar{c}_b, \eta$. The choice of $c'$ makes the terms containing $\frac{U_w}{b_0^2}$ small; then increasing $C_M$ makes the terms containing $\frac{U_w}{c s_M}$ small and makes $\frac{\Lambda_s}{\tau_A}$ large.

Finally choose $C_{\max}$, choose $c_\eta > 2D\gamma_D$, and choose $C_\eta > 2\gamma_D C_{\max} C_M$. After increasing $n_0$, the two inequalities in Equation (64) hold. Choose the theorem horizon constant $C$ large enough that $t_2 \geq t(\varepsilon)$, where $t(\varepsilon)$ is the time threshold in Proposition G.18. Once Proposition G.19 supplies the constant $K_{\text{tiny}}$ used in Lemma G.4, increase $n_0$ again, if needed, so that Equation (57), Equation (56), and the large-sample requirements in Propositions G.18 and G.19 all hold. These choices are made before $t_1$ and the learned-spectrum state are fixed, except for the final threshold $n_0$, which may depend on the fixed initial spectrum through $\tilde{d}$.

We next collect the endpoint ordering relations for learned eigenvalues at time $t_2$.

**Proposition G.10** (Ordering relations at time $t_2$). *With the notation in Equations (47), (50), (51) and (54) and Lemma G.4, assume the displayed large-sample and constant inequalities in Equations (57), (58), (62) to (65) and (67). Let*

$$A_1^\circ := A_1 \setminus T_{\text{tiny}},$$

*and split the non-tiny weak tail coordinates according to their learned eigenvalue at time $t_1$:*

$$B_1^{<\tau_w} := \{j \in B_1 \setminus T_{\text{tiny}} : \tilde{\lambda}_j(t_1) < \tau_w\}, \qquad B_1^{\geq \tau_w} := (B_1 \setminus T_{\text{tiny}}) \setminus B_1^{<\tau_w}.$$

*The following endpoint relations at time $t_2$ hold:*

$$
\begin{aligned}
B_2 &\succ_{t_2} A_1^\circ, \\
A_2 \cup B_2 &\succ_{t_2} A_1, \\
A_3 &\succ_{t_2} A_1, \\
A_2 \cup B_2 \cup A_3 &\succ_{t_2} B_1^{<\tau_w}, \\
B_1^{\geq \tau_w} \cap T_2 &= \emptyset, \\
A_2 \cup A_3 \cup B_2 &\succ_{t_2} (B_1 \cap T_{\text{tiny}}).
\end{aligned}
\tag{68}
$$

*The following strong-coordinate implication also holds:*

$$\text{if } i \in A_2, \ j \in B_2, \text{ and } |\theta_i^*| > |\theta_j^*|, \text{ then } \tilde{\lambda}_i(t_2) > \tilde{\lambda}_j(t_2). \tag{69}$$

*Proof.* The relation $B_2 \succ_{t_2} A_1^\circ$ is Lemma G.6. The two relations

$$A_2 \cup B_2 \succ_{t_2} A_1, \qquad A_3 \succ_{t_2} A_1$$

are conclusions of Lemma G.8. If $j \in B_1^{<\tau_w}$, then Lemma G.9 gives

$$\tilde{\lambda}_j(t_2) < \min\{\Lambda_s/2, (1 + \delta_w/2)\tau_A\},$$

which is below the learned eigenvalue of every coordinate in $A_2 \cup B_2 \cup A_3$ by Lemmas G.5 and G.8. This proves

$$A_2 \cup B_2 \cup A_3 \succ_{t_2} B_1^{<\tau_w}.$$

If $j \in B_1^{\geq \tau_w}$, then $j \in B_1$ and $\tilde{\lambda}_j(t_1) \geq \tau_w$. Since

$$\tau_w = c b_0^{2D} D^{-D/(D+2)} \varepsilon^{2/(D+2)},$$

the first alternative in Condition (4-ii) of Theorem 5.2 is false for this $j$. Therefore, the second alternative in Condition (4-ii) gives

$$\min_{\ell \in T_1} \tilde{\lambda}_\ell(t_1) \geq (1 + \delta_w)\tilde{\lambda}_j(t_1).$$

The coordinate $j$ therefore satisfies all hypotheses in the first item of Lemma G.9 for coordinates with $\tilde{\lambda}_j(t_1) \geq \tau_w$, which gives $j \notin T_2$. Since $j \in B_1^{\geq \tau_w}$ was arbitrary, we have $B_1^{\geq \tau_w} \cap T_2 = \emptyset$.

We next prove $(A_2 \cup A_3 \cup B_2) \succ_{t_2} (B_1 \cap T_{\text{tiny}})$. Lemma G.4 gives $\tilde{\lambda}_j(t_2) < n^{-\delta_{\text{tiny}}}$ for $j \in B_1 \cap T_{\text{tiny}}$. By Equation (56),

$$n^{-\delta_{\text{tiny}}} < \frac{1}{2}\min\{\tau_A, \tau_w, \Lambda_s\}.$$

For every $k \in A_2 \cup B_2$, Lemma G.5 gives

$$\tilde{\lambda}_k(t_2) \geq \Lambda_s > n^{-\delta_{\text{tiny}}} > \tilde{\lambda}_j(t_2).$$

For every $k \in A_3$, Equation (66) gives

$$\tilde{\lambda}_k(t_2) \geq (1 + \delta_w)\tau_A > n^{-\delta_{\text{tiny}}} > \tilde{\lambda}_j(t_2).$$

Hence $A_2 \cup A_3 \cup B_2 \succ_{t_2} (B_1 \cap T_{\text{tiny}})$.

Equations (57), (63) and (64) are assumed here, so Lemma G.7 applies to every $i \in A_2$, $j \in B_2$ with $|\theta_i^*| > |\theta_j^*|$. Its conclusion is exactly Equation (69). □

### G.2.3. Top-$m$ Set Exchange and Tail-Energy Comparison

Recall from Equation (48) that $\Delta_m$ is the tail-energy difference at the fixed cutoff $m = d^\dagger(t_1)$: it is the time-$t_1$ tail signal energy minus the time-$t_2$ tail signal energy. We now prove $\Delta_m \geq 0$. This is the place where the ordering relations at time $t_2$ are converted into the ESD conclusion.

**Proposition G.11** (Top-$m$ set exchange implies $\Delta_m \geq 0$). *With $T_1, T_2, A_1, A_2, A_3, B_1, B_2, C_{B_1}$, and $T_{\text{tiny}}$ fixed as above, assume the budget condition Equation (53), the endpoint relations in Equation (68), the strong-coordinate implication in Equation (69), and the conclusions of Lemma G.4. Then $\Delta_m \geq 0$.*

*Proof.* Although $\Delta_m$ was defined in Equation (48) by comparing tail energies, the total signal energy is fixed. Hence

$$
\begin{aligned}
\Delta_m &= \sum_{i \in T_2} |\theta_i^*|^2 - \sum_{i \in T_1} |\theta_i^*|^2 \\
&= \sum_{i \in T_2 \setminus T_1} |\theta_i^*|^2 - \sum_{i \in T_1 \setminus T_2} |\theta_i^*|^2.
\end{aligned}
\tag{70}
$$

Our proof identifies the entering set $T_2 \setminus T_1$ and the outgoing set $T_1 \setminus T_2$, and then compares their signal energies. This proof splits according to whether there are enough small weak top coordinates $A_1$ to be replaced by the strong tail coordinates $B_2$.

We repeatedly use the following elementary consequence of the definition of $T_2$: if $q \in T_2$ and $p \notin T_2$, then

$$
\tilde{\lambda}_q(t_2) \geq \tilde{\lambda}_p(t_2).
\tag{71}
$$

The symbols $A_1^\circ, B_1^{<\tau_w}$, and $B_1^{\geq \tau_w}$ are taken from Proposition G.10. Recall also that

$$
B_1 = B_1^{<\tau_w} \dot{\cup} B_1^{\geq \tau_w} \dot{\cup} (B_1 \cap T_{\text{tiny}}).
\tag{72}
$$

Set

$$
P := A_2 \cup A_3 \cup B_2.
\tag{73}
$$

*Part 1. The case $|B_2| \leq |A_1|$.* By Equation (73), in this case

$$
|P| = |A_2| + |A_3| + |B_2| \leq |A_2| + |A_3| + |A_1| = |T_1| = m.
\tag{74}
$$

The endpoint relations in Equation (68) give

$$
P \succ_{t_2} A_1, \qquad P \succ_{t_2} B_1^{<\tau_w}, \qquad P \succ_{t_2} (B_1 \cap T_{\text{tiny}}).
\tag{75}
$$

Together with $B_1^{\geq \tau_w} \cap T_2 = \emptyset$, these relations force

$$
P \subseteq T_2.
\tag{76}
$$

To prove Equation (76), suppose that $p \in P \setminus T_2$. Since

$$
T_1 = A_1 \dot{\cup} A_2 \dot{\cup} A_3, \qquad T_1^c = B_1 \dot{\cup} B_2,
$$

and $P = A_2 \cup A_3 \cup B_2$ by Equation (73), every coordinate outside $P$ belongs to $A_1 \dot{\cup} B_1$. Using Equation (72), this gives

$$
[d] \setminus P = A_1 \dot{\cup} B_1^{<\tau_w} \dot{\cup} B_1^{\geq \tau_w} \dot{\cup} (B_1 \cap T_{\text{tiny}}).
$$

Therefore every coordinate in $T_2 \setminus P$ belongs to one of the four sets in the last display. The set $B_1^{\geq \tau_w}$ has empty intersection with $T_2$, and the other three possibilities are strictly below every coordinate of $P$ by Equation (75). Hence no coordinate in $T_2 \setminus P$ can satisfy Equation (71) with this $p$. Therefore

$$
T_2 \subseteq P \setminus \{p\}.
$$

This contradicts

$$
m = |T_2| \leq |P| - 1 \leq m - 1,
$$

where the last inequality uses Equation (74). This proves Equation (76).

By Equation (76) and Equation (73), $A_2 \cup A_3 \cup B_2 \subseteq T_2$. Also $T_2 \setminus T_1 \subseteq T_1^c = B_1 \dot\cup B_2$. Since $A_2 \cup A_3 \subseteq T_2$ and $T_1 = A_1 \dot\cup A_2 \dot\cup A_3$, every coordinate in $T_1 \setminus T_2$ lies in $A_1$. To sum up, we have

$$A_2 \cup A_3 \cup B_2 \subseteq T_2, \qquad T_2 \setminus T_1 \subseteq B_2 \cup B_1, \qquad T_1 \setminus T_2 \subseteq A_1 \subseteq S^c. \tag{77}$$

Define

$$W_{11} := (T_1 \cap S^c) \setminus T_2, \qquad B_{11} := (T_2 \cap B_1) \setminus T_1 = (T_2 \setminus T_1) \cap B_1, \tag{78}$$

where the last equality uses $B_1 \subseteq T_1^c$.

By Equation (77), we have $B_2 \subseteq A_2 \cup A_3 \cup B_2 \subseteq T_2$ and $B_2 \subseteq T_1^c$, so $B_2 \subseteq T_2 \setminus T_1$. Using $T_2 \setminus T_1 \subseteq B_2 \cup B_1$ from Equation (77), we have

$$T_2 \setminus T_1 = B_2 \dot\cup \big((T_2 \setminus T_1) \cap B_1\big) = B_2 \dot\cup B_{11}.$$

Similarly, Equation (77) gives $T_1 \setminus T_2 \subseteq S^c$. Therefore

$$T_1 \setminus T_2 = (T_1 \setminus T_2) \cap S^c = (T_1 \cap S^c) \setminus T_2 = W_{11}.$$

Thus

$$T_2 \setminus T_1 = B_2 \dot\cup B_{11}, \qquad T_1 \setminus T_2 = W_{11}. \tag{79}$$

Because $|T_1| = |T_2| = m$, the two set differences in Equation (79) have the same cardinality, hence

$$|W_{11}| = |B_2| + |B_{11}|. \tag{80}$$

Also, $T_1 \setminus T_2 \subseteq A_1$ by Equation (77), so $|W_{11}| \leq |A_1|$. Combining this with Equation (80) gives

$$|B_{11}| \leq |A_1| - |B_2|.$$

Since $B_{11} \subseteq B_1$, we obtain

$$|B_{11}| \leq \min\{(|A_1| - |B_2|)_+, |B_1|\} = C_{B_1}. \tag{81}$$

Using Equations (70) and (79), we get

$$\Delta_m = \|\theta_{B_2}^*\|_2^2 + \|\theta_{B_{11}}^*\|_2^2 - \|\theta_{W_{11}}^*\|_2^2. \tag{82}$$

Every coordinate in $B_2$ has squared signal at least $M^2$. Every coordinate in $W_{11} \cup B_{11}$ is weak, hence has squared signal at most $\tilde\sigma^2$. Therefore

$$\begin{aligned}
\Delta_m &\geq |B_2| M^2 + \|\theta_{B_{11}}^*\|_2^2 - |W_{11}| \tilde\sigma^2 \\
&\geq |B_2| M^2 - |W_{11}| \tilde\sigma^2 \\
&= |B_2| M^2 - (|B_2| + |B_{11}|) \tilde\sigma^2 \\
&\geq |B_2| M^2 - (|B_2| + C_{B_1}) \tilde\sigma^2 \geq 0,
\end{aligned} \tag{83}$$

where the equality uses Equation (80), the third inequality uses Equation (81), and the last inequality follows from Equation (53).

*Part 2. The case $|B_2| > |A_1|$.* We first remove a small ambiguity caused by tiny weak top coordinates. If $i \in A_3 \cap T_{\text{tiny}}$, then Lemma G.4 gives $\tilde\lambda_i(t_1) < n^{-\delta_{\text{tiny}}}$. By Equation (56), this is smaller than $\tau_A$. Since $i \in A_3 \subseteq T_1 \cap S^c$, the definition of $A_1$ would then put $i$ in $A_1$, contradicting $A_3 \cap A_1 = \emptyset$. Thus

$$A_3 \cap T_{\text{tiny}} = \emptyset. \tag{84}$$

By Equation (73), the assumption $|B_2| > |A_1|$ gives

$$|P| = |A_2| + |A_3| + |B_2| > |A_2| + |A_3| + |A_1| = |T_1| = m. \tag{85}$$

Next we show that no weak tail coordinate enters $T_2$. The endpoint relation $B_1^{\geq \tau_w} \cap T_2 = \emptyset$ in Equation (68) excludes $B_1^{\geq \tau_w}$. The endpoint relation $(A_2 \cup B_2 \cup A_3) \succ_{t_2} B_1^{< \tau_w}$, together with Equation (73), gives

$$P \succ_{t_2} B_1^{< \tau_w}.$$

Equation (68) also gives

$$P \succ_{t_2} (B_1 \cap T_{\text{tiny}}).$$

We claim that neither $B_1^{< \tau_w}$ nor $B_1 \cap T_{\text{tiny}}$ can meet $T_2$. Let $\mathcal{W}$ be $B_1^{< \tau_w}$ or $B_1 \cap T_{\text{tiny}}$. In both cases, Equation (68) together with Equation (73) gives $P \succ_{t_2} \mathcal{W}$. If some $j \in \mathcal{W} \cap T_2$, then every $p \in P$ satisfies $\tilde{\lambda}_p(t_2) > \tilde{\lambda}_j(t_2)$. Hence every $p \in P$ must also belong to $T_2$; otherwise Equation (71), applied with $q = j$ and this $p \notin T_2$, would give $\tilde{\lambda}_j(t_2) \geq \tilde{\lambda}_p(t_2)$. Thus $P \subseteq T_2$, which contradicts $|P| > m = |T_2|$. Therefore $\mathcal{W} \cap T_2 = \emptyset$. Using the partition in Equation (72), we obtain

$$T_2 \cap B_1 = \emptyset. \tag{86}$$

The same rank-count argument excludes $A_1$ from $T_2$. Indeed, Equation (68) gives $A_2 \cup B_2 \succ_{t_2} A_1$ and $A_3 \succ_{t_2} A_1$, so

$$P \succ_{t_2} A_1.$$

Since $|P| > m$, no coordinate of $A_1$ can belong to $T_2$:

$$A_1 \cap T_2 = \emptyset. \tag{87}$$

Define

$$B_{21} := \{i \in B_2 : \pi_{t_2}^{-1}(i) > m\},$$
$$A_{21} := \{i \in A_2 : \pi_{t_2}^{-1}(i) > m\},$$
$$A_{31} := \{i \in A_3 : \pi_{t_2}^{-1}(i) > m\},$$
$$Q_{\text{tiny}}^{(2)} := (T_1 \cap S^c \cap T_{\text{tiny}}) \setminus T_2,$$

and let $B_{22} := B_2 \setminus B_{21} = B_2 \cap T_2$. Since $T_1^c = B_1 \dot\cup B_2$, Equation (86) gives

$$T_2 \setminus T_1 = T_2 \cap B_2 = B_{22}. \tag{88}$$

We next identify the outgoing part of $A_1$. Since $A_1 \subseteq T_1 \cap S^c$, we have $A_1 \cap T_{\text{tiny}} \subseteq T_1 \cap S^c \cap T_{\text{tiny}}$. Conversely, if $i \in T_1 \cap S^c \cap T_{\text{tiny}}$, then $i$ is a weak top coordinate. The relation $A_3 \cap T_{\text{tiny}} = \emptyset$ in Equation (84) gives $i \notin A_3$, and $i \in S^c$ gives $i \notin A_2$. Since $T_1 = A_1 \dot\cup A_2 \dot\cup A_3$, we must have $i \in A_1$. Thus

$$A_1 \cap T_{\text{tiny}} = T_1 \cap S^c \cap T_{\text{tiny}}.$$

This identity identifies the tiny part of $A_1$. After removing coordinates that remain in $T_2$, it gives

$$Q_{\text{tiny}}^{(2)} = (T_1 \cap S^c \cap T_{\text{tiny}}) \setminus T_2 = (A_1 \cap T_{\text{tiny}}) \setminus T_2 = A_1 \cap T_{\text{tiny}},$$

where the last equality uses $A_1 \cap T_2 = \emptyset$ from Equation (87). Thus $Q_{\text{tiny}}^{(2)}$ is the tiny part of the outgoing set $A_1 \setminus T_2$, while $A_1^\circ = A_1 \setminus T_{\text{tiny}}$ is its non-tiny part. Therefore

$$A_1 \setminus T_2 = A_1 = A_1^\circ \dot\cup Q_{\text{tiny}}^{(2)}. \tag{89}$$

Hence

$$T_1 \setminus T_2 = A_1^\circ \dot\cup A_{21} \dot\cup A_{31} \dot\cup Q_{\text{tiny}}^{(2)}. \tag{90}$$

The sets in Equations (88) and (90) have the same cardinality because $|T_1| = |T_2| = m$. Therefore

$$|B_{22}| = |A_1^\circ| + |A_{21}| + |A_{31}| + |Q_{\text{tiny}}^{(2)}|. \tag{91}$$

Using Equations (70), (88) and (90), we get

$$\Delta_m = \|\theta^*_{B_{22}}\|^2_2 - \|\theta^*_{A^\circ_1}\|^2_2 - \|\theta^*_{A_{21}}\|^2_2 - \|\theta^*_{A_{31}}\|^2_2 - \sum_{i \in Q^{(2)}_{\text{tiny}}} |\theta^*_i|^2. \tag{92}$$

We next compare the strong coordinates in $B_{22}$ with the strong coordinates in $A_{21}$. By Equation (91), choose $B^{(2)}_{22} \subseteq B_{22}$ with

$$|B^{(2)}_{22}| = |A_{21}|.$$

The remaining coordinates in $B_{22}$ can be partitioned into disjoint sets $B^{(1)}_{22}, B^{(3)}_{22}, B^{(4)}_{22}$ satisfying

$$|B^{(1)}_{22}| = |A^\circ_1|, \qquad |B^{(3)}_{22}| = |A_{31}|, \qquad |B^{(4)}_{22}| = |Q^{(2)}_{\text{tiny}}|. \tag{93}$$

The set $B^{(2)}_{22}$ may be chosen arbitrarily: every coordinate in $B_{22}$ has true signal at least as large as every coordinate in $A_{21}$. This can be proved by contradiction. Suppose that $i \in A_{21}$ and $j \in B_{22}$ satisfy $|\theta^*_i| > |\theta^*_j|$. Since $A_{21} \subseteq A_2$ and $B_{22} \subseteq B_2$, the strong-coordinate implication Equation (69) gives

$$\tilde{\lambda}_i(t_2) > \tilde{\lambda}_j(t_2).$$

This contradicts Equation (71), because $j \in B_{22} \subseteq T_2$ and $i \in A_{21} \subseteq T^c_2$.

Therefore, we have

$$\|\theta^*_{B^{(2)}_{22}}\|^2_2 \geq \|\theta^*_{A_{21}}\|^2_2. \tag{94}$$

Using the partition of $B_{22}$ and then Equation (94), the identity Equation (92) gives

$$\begin{aligned}
\Delta_m &= \|\theta^*_{B^{(1)}_{22}}\|^2_2 + \|\theta^*_{B^{(2)}_{22}}\|^2_2 + \|\theta^*_{B^{(3)}_{22}}\|^2_2 + \|\theta^*_{B^{(4)}_{22}}\|^2_2 \\
&\quad - \|\theta^*_{A^\circ_1}\|^2_2 - \|\theta^*_{A_{21}}\|^2_2 - \|\theta^*_{A_{31}}\|^2_2 - \sum_{i \in Q^{(2)}_{\text{tiny}}} |\theta^*_i|^2 \\
&\geq \|\theta^*_{B^{(1)}_{22}}\|^2_2 + \|\theta^*_{B^{(3)}_{22}}\|^2_2 + \|\theta^*_{B^{(4)}_{22}}\|^2_2 - \|\theta^*_{A^\circ_1}\|^2_2 - \|\theta^*_{A_{31}}\|^2_2 - \sum_{i \in Q^{(2)}_{\text{tiny}}} |\theta^*_i|^2.
\end{aligned} \tag{95}$$

The sets $B^{(1)}_{22}, B^{(3)}_{22}, B^{(4)}_{22}$ are subsets of $B_2$, so each of their coordinates has squared signal at least $M^2$. The sets $A^\circ_1, A_{31}, Q^{(2)}_{\text{tiny}}$ consist of weak coordinates, so each of their coordinates has squared signal at most $\tilde{\sigma}^2$. Therefore, using Equation (93),

$$\Delta_m \geq (|A^\circ_1| + |A_{31}| + |Q^{(2)}_{\text{tiny}}|)(M^2 - \tilde{\sigma}^2) \geq 0. \tag{96}$$

This proves $\Delta_m \geq 0$ in Case 2 as well. $\qquad\square$

### G.2.4. CONCLUSION OF THEOREM 5.2

By Proposition G.11, $\Delta_m \geq 0$. Hence

$$\sum_{i:\pi^{-1}_{t_2}(i)>m} |\theta^*_i|^2 \leq \sum_{i:\pi^{-1}_{t_1}(i)>m} |\theta^*_i|^2.$$

Since $m = d^\dagger(t_1) \in [d]$, we have $m \geq 1$. Dividing by $m$ gives

$$\mathbf{H}_{\theta^*, \tilde{\lambda}(t_2)}(m) \leq \mathbf{H}_{\theta^*, \tilde{\lambda}(t_1)}(m) \leq \sigma^2.$$

Therefore $m = d^\dagger(t_1)$ is feasible for the ESD at time $t_2$, and

$$d^\dagger(t_2) \leq m = d^\dagger(t_1).$$

Since the proof was carried out on $\mathcal{E}$, and $\mathbb{P}(\mathcal{E}) \geq 1 - 4/n$, this completes the proof of Theorem 5.2.

### G.3. Scalar Conservation Identities

We omit the subscript $j$ in this subsection because the same calculation applies to every $j \in [d]$. Throughout we assume $D \geq 1$ and write $z := z_j = \theta_j^* + \xi_j$. By Equation (39), direct differentiation gives

$$\frac{d}{dt}a^2 = \frac{1}{D}\frac{d}{dt}b^2 = \frac{d}{dt}\beta^2 = 2ab^D\beta(\theta^* - \theta + \xi).$$

Consequently, we have

$$a^2(t) - \beta^2(t) \equiv a_0^2, \quad b^2(t) - D\beta^2(t) \equiv b_0^2. \tag{97}$$

Using Equation (97), we see that

$$a(t) = \left(\beta^2(t) + a_0^2\right)^{1/2}, \quad b(t) = \left(D\beta^2(t) + b_0^2\right)^{1/2} > 0.$$

Using these conservation quantities, we can prove the following bounds in terms of $\beta$:

$$\begin{aligned} \max\left(a_0, |\beta|\right) \leq a \leq \sqrt{2}\max\left(a_0, |\beta|\right), \\ \max\left(b_0, \sqrt{D}|\beta|\right) \leq b \leq \sqrt{2}\max\left(b_0, \sqrt{D}|\beta|\right). \end{aligned} \tag{98}$$

These bounds also imply that $|\theta| = \left|ab^D\beta\right| \geq D^{D/2}|\beta|^{D+2}$. For the evolution of $\theta$, a direct computation gives, for $\beta(t) \neq 0$,

$$\begin{aligned} \dot{\theta} &= \dot{a}b^D\beta + aDb^{D-1}\dot{b}\beta + ab^D\dot{\beta} \\ &= \left[\left(b^D\beta\right)^2 + \left(Dab^{D-1}\beta\right)^2 + \left(ab^D\right)^2\right](\theta^* - \theta + \xi) \\ &= \theta^2\left(a^{-2} + D^2b^{-2} + \beta^{-2}\right)(\theta^* - \theta + \xi). \end{aligned} \tag{99}$$

The last expression can be extended by continuity to $\beta = 0$.

We also have

$$|\theta| = \left|ab^D\beta\right| \geq D^{D/2}|\beta|^{D+2} \implies |\beta| \leq \left(D^{-D/2}|\theta|\right)^{1/(D+2)}. \tag{100}$$

Therefore,

$$\theta^2\left(a^{-2} + D^2b^{-2} + \beta^{-2}\right) \geq \theta^2\beta^{-2} \geq D^{-\frac{D}{D+2}}|\theta|^{\frac{2D+2}{D+2}}. \tag{101}$$

### G.4. One-Coordinate Flow Lemmas

We study the scalar flow for a fixed coordinate $j$. Throughout this subsection, we assume that $|\xi_j| \leq \kappa_j$ for some $\kappa_j > 0$. This assumption is verified on the concentration event in Lemma G.1.

Before the analysis, we streamline some notation. Since $j$ is given, we drop the subscript $j$ to simplify the exposition throughout this subsection; for example, we write $\lambda$ for $\lambda_j$, $\theta^*$ for $\theta_j^*$, and $z$ for $z_j = \theta_j^* + \xi_j = \theta^* + \xi$.

**Lemma G.12** (Monotonicity from equation)**.** *Consider Equation (39). Let $s = \mathrm{sign}(z)$, with the convention $s = 1$ if $z = 0$.*

1. *$a(t)$ is nondecreasing, and $s\beta(t)$ and $s\theta(t)$ are non-negative and nondecreasing.*

2. *We have*
$$|z| \geq s\theta(t) \geq 0 \quad \forall t \geq 0.$$

3. *Since $z = \theta^* + \xi$ and $|\xi| \leq \kappa$, we have*

$$|\theta^* - \theta(t)| \leq |\theta^*| + \kappa, \quad \forall t \geq 0.$$

4. $|\theta^* - \theta(t)|$ *is decreasing provided that* $|\theta^* - \theta(t)| > \kappa$.

5. *If* $|\theta^* - \theta(t_1)| \leq \kappa$ *for some* $t_1$, *we have*

$$|\theta^* - \theta(t)| \leq \kappa \text{ for all } t \geq t_1.$$

*Proof.* If $z = 0$, then $\beta(t) = \theta(t) = 0$ and the claims are immediate. Otherwise, replace $(\beta, \theta, \theta^*, \xi)$ by $(s\beta, s\theta, s\theta^*, s\xi)$. The transformed system has positive response $sz = |z|$, so Items 1 and 2 follow from Equation (39) applied to the transformed variables. Item 3 is implied by Item 2.

To prove Item 4, consider Equation (99), from which we have

$$\dot{\theta} = \theta^2 \left( a^{-2} + D^2 b^{-2} + \beta^{-2} \right) (z - \theta),$$

which, after the same sign normalization, implies $s\dot{\theta} \geq 0$ as long as $s\theta \leq |z|$.

Since $|\theta^* - \theta(t)| > \kappa$, in the sign-normalized coordinates we have either $s\theta(t) > s\theta^* + \kappa$ or $s\theta(t) < s\theta^* - \kappa$.

In the sign-normalized coordinates, the overshoot case is not possible; otherwise, we have $0 < |z| = s\xi + s\theta^* \leq \kappa + s\theta^* < s\theta \leq |z|$, which is a contradiction. In the second case, we have $|\theta^* - \theta(t)| = s\theta^* - s\theta(t)$, which is decreasing because $s\dot{\theta} \geq 0$.

Item 5 is implied by Item 4. □

**Lemma G.13** (Approaching from below). *Consider Equation* (39). *Suppose* $|\theta^*| \geq 8\kappa$ *and set* $s = \text{sign}(\theta^*)$. *Suppose* $t_0 \geq 0$ *satisfies that* $0 \leq s\theta(t_0) < \frac{1}{4}|\theta^*|$. *Define*

$$T^{sig} = \inf \left\{ u \geq 0 : s\theta(t_0 + u) \geq |\theta^*|/4 \right\}.$$

*This is the extra time needed from $t_0$ for $s\theta$ to reach $|\theta^*|/4$. We have*

$$T^{sig} \leq \begin{cases} C_1 |\theta^*|^{-1} b_0^{-D} \ln(\frac{eb_0}{a_0\sqrt{D}}) & a_0 \leq b_0/\sqrt{D}; \\ C_1 |\theta^*|^{-1} a_0^{-1} \ln(\frac{ea_0\sqrt{D}}{b_0}) & a_0 > b_0/\sqrt{D}, \text{ and } D = 1; \\ C_1 |\theta^*|^{-1} D^{-\frac{1}{2}} a_0^{-1} b_0^{-D+1} & a_0 > b_0/\sqrt{D}, \text{ and } D > 1. \end{cases} \tag{102}$$

*Proof.* By replacing $(\beta, \theta, \theta^*, \xi)$ with $(s\beta, s\theta, s\theta^*, s\xi)$, it suffices to prove the case $\theta^* > 0$. In this normalized case, since $|z - \theta^*| = |\xi| \leq \kappa$ and $\theta^* \geq 8\kappa$, we have $z \geq 7\kappa > 0$. Therefore, $\theta(t) \in [0, z]$. For any $t \leq t_0 + T^{sig}$, we use $\theta^* \geq 8\kappa$ to show that

$$z - \theta(t) = \theta^* - \theta(t) + \xi \geq \frac{3}{4}\theta^* - \kappa \geq \frac{1}{2}\theta^*.$$

Let $r = \min\left(a_0, b_0/D^{\frac{1}{2}}\right)$ and $R = \max\left(a_0, b_0/D^{\frac{1}{2}}\right)$. Define the following time points if they exist:

$$T^{pos,1} = \inf\left\{s \geq 0 : \beta(t_0 + s) \geq r\right\}; \quad T^{pos,2} = \inf\left\{s \geq 0 : \beta(t_0 + s) \geq R\right\}$$

$$T^{sig} = \inf\left\{s \geq 0 : |\theta^* - \theta(t_0 + s)| \leq \frac{3}{4}\theta^*\right\}.$$

If $\beta(t_0)$ has already crossed $r$ or $R$, the corresponding stage time is set to zero and the relevant stage below is skipped. This only decreases the remaining hitting time, so the same upper bound derived from the zero initial time remains valid from the arbitrary state $t_0$.

We will first bound both $T^{pos,1}$ and $T^{pos,2}$.

From Equation (39), we have

$$\dot{\beta}(t) = a(t)b^D(t)[\theta^* + \xi - \theta(t)] \geq \frac{1}{4}\theta^* a(t)b^D(t), \quad \text{for} \quad t \leq t_0 + T^{\text{sig}}. \tag{103}$$

**Stage 1:** $0 \leq s \leq T^{\text{pos},1}$. Note that $\sqrt{2}a_0 > a(t) > a_0$, and $b(t)^D = (D\beta(t)^2 + b_0^2)^{\frac{D}{2}} \geq b_0^D$. We have

$$\dot{\beta}(t_0 + s) \geq \frac{1}{4}\theta^* a_0 b_0^D \geq \frac{1}{4}\theta^* a_0 b_0^D,$$

which shows that $\beta$ increases at least linearly. Therefore, we have

$$T^{\text{pos},1} \leq 4r\left(\theta^* a_0 b_0^D\right)^{-1}. \tag{104}$$

**Stage 2:** $T^{\text{pos},1} \leq s \leq T^{\text{pos},2}$. Consider two cases.

Case 1: If $a_0 \leq b_0/\sqrt{D}$, $r = a_0$ and $R = b_0/\sqrt{D}$. Note that $a \geq \beta$ by Equation (98). We use Equation (103) to get

$$\dot{\beta}(t_0 + s) \geq \frac{1}{4}\theta^* b_0^D |\beta(t_0 + s)|,$$

By Grönwall's inequality, we have

$$T^{\text{pos},2} - T^{\text{pos},1} \leq 4\left(\theta^* b_0^D\right)^{-1} \ln \frac{b_0}{a_0\sqrt{D}}. \tag{105}$$

Case 2: If $a_0 > b_0/\sqrt{D}$, $R = a_0$ and $r = b_0/\sqrt{D}$. We use $b \geq \sqrt{D}\beta$ in Equation (98), together with $a > a_0$, $b > \sqrt{D}|\beta|$, and Equation (103), to get

$$\dot{\beta}(t_0 + s) \geq \frac{1}{4}\theta^* a_0 D^{\frac{D}{2}} |\beta(t_0 + s)|^D.$$

Applying Lemma G.3 when $D = 1$ and the first item of Lemma G.2 with $p = D$ when $D \geq 2$, starting from $r = b_0/\sqrt{D}$ and stopping at $R = a_0$, we have

$$T^{\text{pos},2} - T^{\text{pos},1} \leq \begin{cases} 4\left(\theta^* a_0\right)^{-1} \ln \frac{a_0\sqrt{D}}{b_0}, & \text{if} \quad D = 1; \\ 4\left((D-1)\theta^* a_0 D^{D/2}\right)^{-1} \left[\left(b_0/\sqrt{D}\right)^{-(D-1)} - a_0^{-(D-1)}\right], & \text{if} \quad D \geq 2. \end{cases} \tag{106}$$

**Stage 3:** If $T^{\text{sig}} \leq T^{\text{pos},2}$, then we can use the Stage 2 upper bound as a bound for $T^{\text{sig}}$. Now, we consider the case $T^{\text{pos},2} < T^{\text{sig}}$. We combine Equation (98) with $a > |\beta|$, $b > \sqrt{D}|\beta|$, and Equation (103) to get

$$\dot{\beta}(t_0 + T^{\text{pos},2} + s) \geq \frac{1}{4}\theta^* D^{D/2}|\beta(t_0 + T^{\text{pos},2} + s)|^{D+1}, \quad \text{for} \quad s \in \left[0, T^{\text{sig}} - T^{\text{pos},2}\right].$$

Moreover, we have $\beta(t_0 + T^{\text{pos},2}) = R > 0$. By Lemma G.2, we have

$$T^{\text{sig}} - T^{\text{pos},2} \leq 4D^{-\frac{D+2}{2}}(\theta^*)^{-1}R^{-D}. \tag{107}$$

We now bound $T^{\text{sig}}$ by summing Equation (104), Equation (107), and Equation (105) if $a_0 \leq b_0/\sqrt{D}$, or by summing Equation (104), Equation (107), and Equation (106) if $a_0 > b_0/\sqrt{D}$.

If $a_0 \leq b_0/\sqrt{D}$, we can bound the right-hand sides of Equation (104) and Equation (107) by $4(\theta^*)^{-1}b_0^{-D}$ and $4(\theta^*)^{-1}b_0^{-D}$ respectively.

If $a_0 > b_0/\sqrt{D}$, we can bound the right-hand sides of Equation (104) and Equation (107) by $4(\theta^*)^{-1}D^{-\frac{1}{2}}a_0^{-1}b_0^{-D+1}$ and $4(\theta^*)^{-1}D^{-\frac{1}{2}}a_0^{-1}b_0^{-D+1}$ respectively. Furthermore, if $D > 1$, we can bound Equation (106) by $4(\theta^*)^{-1}(D-1)^{-1}D^{-\frac{1}{2}}a_0^{-1}b_0^{-D+1}$.

This leads to

$$T^{\text{sig}} \leq \begin{cases} 4(\theta^*)^{-1}b_0^{-D}\left(3 + \ln(\frac{b_0}{a_0\sqrt{D}})\right) & a_0 \leq b_0/\sqrt{D}; \\ 4(\theta^*)^{-1}a_0^{-1}\left(3 + \ln(\frac{a_0\sqrt{D}}{b_0})\right) & a_0 > b_0/\sqrt{D}, \text{ and } D = 1; \\ 16(\theta^*)^{-1}D^{-\frac{1}{2}}a_0^{-1}b_0^{-D+1} & a_0 > b_0/\sqrt{D}, \text{ and } D > 1. \end{cases} \tag{108}$$

$\square$

**Lemma G.14** (Approximation time near $\theta^*$). *Consider Equation* (39) *and set* $s = \text{sign}(\theta^*)$. *Suppose* $|\theta^*| \geq 8\kappa$. *Suppose that, for some* $t_0 \geq 0$,

$$\frac{1}{4}|\theta^*| \leq s\theta(t_0) \leq |\theta^*| - \kappa.$$

*Then, for any* $\delta > 0$, *we have*

$$|\theta^* - \theta(t)| \leq \kappa + \delta, \quad \forall t \geq t_0 + 4^{\frac{2D+2}{D+2}} D^{\frac{D}{D+2}}|\theta^*|^{-\frac{2D+2}{D+2}} \ln^+ \frac{|\theta^* - \theta(t_0)| - \kappa}{\delta}.$$

*Proof.* After sign normalization it suffices to treat $\theta^* > 0$. Given any $\delta > 0$, if $s\theta(t_0) \geq |\theta^*| - \kappa - \delta$, we have $|\theta^* - \theta(t)| \leq \kappa + \delta$ for all $t \geq t_0$ by Lemma G.12 (Items 4 and 5) and the desired result is proved.

Next, suppose $s\theta(t_0) < |\theta^*| - \kappa - \delta$. Define

$$T^{\text{app}} = \inf\left\{s \geq 0 : |\theta^* - \theta(t_0 + s)| \leq \kappa + \delta\right\}.$$

By Lemma G.12 (Item 4) again, it suffices to provide an upper bound on $T^{\text{app}}$.

For all $t \geq t_0$, we have $\frac{1}{4}|\theta^*| \leq s\theta(t)$ by Lemma G.12 (Item 1). Consequently, Equation (101) implies that

$$\theta^2\left(a^{-2} + D^2b^{-2} + \beta^{-2}\right) \geq D^{-\frac{D}{D+2}}|\theta|^{\frac{2D+2}{D+2}} \geq 4^{-\frac{2D+2}{D+2}}D^{-\frac{D}{D+2}}|\theta^*|^{\frac{2D+2}{D+2}} := c_0.$$

Furthermore, by Equation (99), we have

$$s\dot{\theta} = \theta^2\left(a^{-2} + D^2b^{-2} + \beta^{-2}\right)(s\theta^* - s\theta + s\xi) \geq c_0(|\theta^*| - \kappa - s\theta).$$

Let $x(u) := |\theta^*| - \kappa - s\theta(t_0 + u)$ with $x(0) = |\theta^*| - \kappa - s\theta(t_0)$. Note that $T^{\text{app}}$ is the hitting time at which $x(u)$ reaches $\delta$.

Note that $\dot{x}(u) = -s\dot{\theta}(t_0 + u) \leq -c_0 x(u)$. Applying Lemma G.3 to $x(s)$, we have

$$T^{\text{app}} \leq c_0^{-1} \ln^+ \frac{|\theta^* - \theta(t_0)| - \kappa}{\delta}.$$

$\square$

**Lemma G.15.** *Consider Equation* (39) *with arbitrary sign of* $\theta^*$, *and set* $s = \text{sign}(\theta^*)$. *Suppose* $|\theta^*| \geq 8\kappa$. *Since* $|\xi| \leq \kappa$ *and* $|\theta^*| \geq 8\kappa$, $z$ *has the same sign as* $\theta^*$.

*There exist two absolute constants* $C_1, C_2$ *such that, for every* $\delta > 0$, *we have*

$$|\theta^* - \theta(t)| \leq \kappa + \delta, \quad \forall t \geq \bar{T}^{app}(\delta),$$

*where*

$$\bar{T}^{app}(\delta) := \bar{T}^{sig} + C_2 D^{\frac{D}{D+2}}|\theta^*|^{-\frac{2D+2}{D+2}} \ln^+ \frac{|\theta^*|}{\delta}, \tag{109}$$

*and*

$$\bar{T}^{sig} := \begin{cases} C_1|\theta^*|^{-1}b_0^{-D}\ln(\frac{eb_0}{a_0\sqrt{D}}) & a_0 \leq b_0/\sqrt{D}; \\ C_1|\theta^*|^{-1}a_0^{-1}\ln(\frac{ea_0\sqrt{D}}{b_0}) & a_0 > b_0/\sqrt{D}, \text{ and } D = 1; \\ C_1|\theta^*|^{-1}D^{-\frac{1}{2}}a_0^{-1}b_0^{-D+1} & a_0 > b_0/\sqrt{D}, \text{ and } D > 1. \end{cases} \tag{110}$$

*Proof.* We will repeatedly apply the monotonicity of Lemma G.12.

Recall $T^{\mathrm{sig}}$ defined in Lemma G.13 with $t_0 = 0$ and let $\tau_{\mathrm{sig}}$ be the upper bound on $T^{\mathrm{sig}}$ we found therein. Then $s\theta(\tau_{\mathrm{sig}}) \geq \frac{|\theta^*|}{4}$.

If $s\theta(\tau_{\mathrm{sig}}) > |\theta^*| - \kappa$, then Lemma G.12 gives $|\theta^* - \theta(\tau_{\mathrm{sig}})| \leq \kappa$, and Item 5 of Lemma G.12 implies $|\theta^* - \theta(t)| \leq \kappa \leq \kappa + \delta$ for all $t \geq \tau_{\mathrm{sig}}$.

If $s\theta(\tau_{\mathrm{sig}}) \leq |\theta^*| - \kappa$, then the hypotheses of Lemma G.14 hold with $t_0 = \tau_{\mathrm{sig}}$. Applying Lemma G.14, we get $|\theta^* - \theta(t)| \leq \kappa + \delta$ for all $t \geq \tau_{\mathrm{sig}} + \tau_{\mathrm{app}}$, where

$$\tau_{\mathrm{app}} = 4^{\frac{2D+2}{D+2}} D^{\frac{D}{D+2}} |\theta^*|^{-\frac{2D+2}{D+2}} \ln^+ \frac{|\theta^* - \theta(\tau_{\mathrm{sig}})| - \kappa}{\delta}.$$

In both cases, $|\theta^* - \theta(\tau_{\mathrm{sig}})| - \kappa \leq |\theta^*|$. We complete the proof by defining $\bar{T}^{\mathrm{sig}} = \tau_{\mathrm{sig}}$ and $\bar{T}^{app}(\delta) = \tau_{\mathrm{sig}} + 4^{\frac{2D+2}{D+2}} D^{\frac{D}{D+2}} |\theta^*|^{-\frac{2D+2}{D+2}} \ln^+ \frac{|\theta^*|}{\delta}$. $\qquad \square$

**Lemma G.16.** *Consider Equation* (39). *Denote* $r' = \min\{a_0, b_0/D\}$, $R' = \max\{a_0, b_0/D\}$. *Define* $T_1 = \inf\{t : |\beta(t)| > r'\}$, *and* $T_2 = \inf\{t : |\beta(t)| > R'\}$. *If* $D \geq 1$ *and* $t$ *satisfies the following condition,*

$$\sqrt{2e} a_0 b_0^D \int_0^t (|\theta^*| + |\xi|) \mathrm{d}s \leq \min(a_0, b_0/D),$$

*then we have*

$$|\theta(t)| \leq 2e \cdot a_0^2 b_0^{2D} \int_0^t (|\theta^*| + |\xi|) \mathrm{d}s. \tag{111}$$

*Moreover, if* $a_0 \leq b_0/D$ *and* $T_1 \leq u \leq T_2$ *satisfies the following condition,*

$$\sqrt{2e} b_0^D \int_{T_1}^u (|\theta^*| + |\xi|) \mathrm{d}s \leq \ln \frac{b_0/D}{a_0},$$

*then we have*

$$|\beta(u)| \leq a_0 \exp\left(\sqrt{2e} b_0^D \int_{T_1}^u (|\theta^*| + |\xi|) \mathrm{d}s\right);$$

$$|\theta(u)| \leq \sqrt{2e} a_0^2 b_0^D \exp\left(2\sqrt{2e} b_0^D \int_{T_1}^u (|\theta^*| + |\xi|) \mathrm{d}s\right).$$

*Proof.* From Equation (39), we have

$$|\beta(t)| \leq \int_0^t a(s) b^D(s) (|\theta^*| + |\xi|) ds.$$

Consider $t \leq T_1$. We use Equation (98) to get $a(t) \leq \sqrt{2} a_0$; by Equation (97), $b^2(t) - D\beta^2(t) \equiv b_0^2$. Consequently, we have $b^2(t) \leq (1 + \frac{1}{D}) b_0^2$, and thus

$$|\beta(t)| \leq \sqrt{2e} a_0 b_0^D \int_0^t (|\theta^*| + |\xi|) dt, \tag{112}$$

which implies Equation (111) using $|\theta| = |ab^D \beta|$. Furthermore, Equation (112) implies that

$$T_1 \geq \inf\left\{t \geq 0 : \sqrt{2e} a_0 b_0^D \int_0^t (|\theta^*| + |\xi|) ds \geq r'\right\}.$$

For the case $t > T_1$, suppose in the following that $a_0 \leq b_0/D$. We have $r' = a_0$ and $R' = b_0/D$.

Consider $t \in (T_1, T_2)$. We have $a(t) \leq \sqrt{2}|\beta(t)|$ and $b(t)^2 \leq (1 + \frac{1}{D})b_0^2$. Consequently, Equation (39) implies that

$$|\dot{\beta}(t)| \leq \sqrt{2e}b_0^D|\beta(t)|(|\theta^*| + |\xi|)$$

and

$$|\beta(t)| \leq a_0 + \sqrt{2e}b_0^D \int_{T_1}^t |\beta(s)|(|\theta^*| + |\xi|)\mathrm{d}s, \tag{113}$$

for any $t \in (T_1, T_2)$ where we have used $|\beta(T_1)| = a_0$. By Grönwall's inequality, we have

$$|\beta(t)| \leq a_0 \exp\left(\sqrt{2e}b_0^D \int_{T_1}^t (|\theta^*| + |\xi|)\mathrm{d}s\right), \quad t \in (T_1, T_2).$$

By definition of $T_2$, we have

$$T_2 \geq \inf\left\{t \geq T_1 : \sqrt{2e}b_0^D \int_{T_1}^t (|\theta^*| + |\xi|)\mathrm{d}s = \ln\frac{b_0/D}{a_0}\right\}. \tag{114}$$

The bound for $\theta(t)$ now follows by using the bounds $a(t) \leq \sqrt{2}|\beta|, b^2(t) \leq (1 + \frac{1}{D})b_0^2$ to get

$$|\theta(t)| = |ab^D\beta| \leq \sqrt{2e}b_0^D|\beta|^2 \leq \sqrt{2e}a_0^2b_0^D \exp\left(2\sqrt{2e}b_0^D \int_{T_1}^t (|\theta^*| + |\xi|)\mathrm{d}s\right), \forall t \in (T_1, T_2).$$

$\qquad\square$

**Lemma G.17** (Tiny-coordinate no-escape consequence). *Consider Equation* (39). *Assume* $a_0 \leq b_0/D$. *For any* $T > 0$, *set*

$$B_T := b_0^D T(|\theta^*| + |\xi|).$$

*If*

$$\sqrt{2e}\, B_T \leq \ln\frac{b_0/D}{a_0},$$

*then* $|\beta(t)| \leq b_0/D$ *for every* $t \in [0, T]$, *and*

$$|\beta(T)| \leq a_0 \exp(\sqrt{2e}\, B_T).$$

*Proof.* Let
$$T_{\mathrm{low}} := \inf\{t : |\beta(t)| > \min(a_0, b_0/D)\}, \qquad T_{\mathrm{esc}} := \inf\{t : |\beta(t)| > b_0/D\}.$$

Since $a_0 \leq b_0/D$, before $T_{\mathrm{low}}$ we have $|\beta(t)| \leq a_0$. Hence Equation (98) gives $a(t) \leq \sqrt{2}\, a_0$, while Equation (97) gives $b^2(t) \leq (1 + D^{-1})b_0^2$ and therefore $b^D(t) \leq e^{1/2}b_0^D$. By the sign-invariant monotonicity in Lemma G.12, $|z - \theta(t)| \leq |z| \leq |\theta^*| + |\xi|$ along the trajectory. Applying Equation (39) yields

$$|\beta(t)| \leq \sqrt{2e}\, a_0 b_0^D t(|\theta^*| + |\xi|) \leq a_0 \exp(\sqrt{2e}\, B_T), \qquad t \leq T \wedge T_{\mathrm{low}}.$$

Under $a_0 \leq b_0/D$, $T_{\mathrm{low}}$ and $T_{\mathrm{esc}}$ are the two hitting times $T_1$ and $T_2$ in Lemma G.16. For every $t \in [T_{\mathrm{low}}, T \wedge T_{\mathrm{esc}}]$,

$$\sqrt{2e}\, b_0^D \int_{T_{\mathrm{low}}}^t (|\theta^*| + |\xi|)\, ds \leq \sqrt{2e}\, B_T \leq \ln\frac{b_0/D}{a_0}.$$

Hence the second estimate in Lemma G.16 gives, for every $t \in [T_{\mathrm{low}}, T \wedge T_{\mathrm{esc}}]$,

$$|\beta(t)| \leq a_0 \exp\left(\sqrt{2e}\, b_0^D (t - T_{\mathrm{low}})(|\theta^*| + |\xi|)\right) \leq a_0 \exp(\sqrt{2e}\, B_T) \leq b_0/D.$$

Since $T_{\mathrm{esc}}$ is defined using the strict event $|\beta| > b_0/D$, $T_{\mathrm{esc}}$ cannot occur before $T$. The endpoint bound follows from the same display with $t = T$. $\qquad\square$

## G.5. Endpoint Estimates for Strong and Tiny Coordinates

The proof of Theorem 5.2 uses two one-coordinate consequences of the dynamic analysis. Proposition G.18 gives approximation of strong coordinates after the shrinkage time $t(\varepsilon)$. Proposition G.19 shows that weak coordinates with sufficiently small initial eigenvalues remain polynomially small throughout the time interval considered in the theorem.

**Proposition G.18** (Strong-coordinate approximation time). *Assume Assumption 5.1 and assume that the events $E_k$ in Lemma G.1 hold for all coordinates $k$. Work under the fixed initial spectrum convention, so that $\tilde{d} = \sum_j \lambda_j$ is finite and independent of $n$. Let*

$$\varepsilon := 2C_{proxy}^{1/2}\sqrt{\frac{\ln(e + n\tilde{d})}{n}}\ln n, \qquad \varepsilon' := 2C_{proxy}^{1/2}\sqrt{\frac{\ln n}{n}}.$$

*For any $j \in S$ satisfying $|\theta_j^*| \geq 8\varepsilon'$,*

$$|\theta_j(t) - \theta_j^*| \leq 2\varepsilon', \qquad t \geq t(\varepsilon), \tag{115}$$

*where $t(\varepsilon) := Cb_0^{-D}\varepsilon^{-1}\ln n$ for an absolute constant $C$.*

*Proof.* On the event $\cap_k E_k$, Lemma G.1 gives

$$\|\xi_S\|_\infty \leq \varepsilon'. \tag{116}$$

Fix $j \in S$ with $|\theta_j^*| \geq 8\varepsilon'$. Apply Lemma G.15 with approximation parameter $\delta_{\mathrm{app}} = \varepsilon'$ and noise level $\kappa = \varepsilon'$. This gives

$$|\theta_j(t) - \theta_j^*| \leq 2\varepsilon', \qquad t \geq \bar{T}^{\mathrm{app}}(\varepsilon'),$$

where

$$\bar{T}^{\mathrm{app}}(\varepsilon') \leq T^{\mathrm{sig}} + C_2 D^{D/(D+2)}|\theta_j^*|^{-(2D+2)/(D+2)}\ln^+\frac{2|\theta_j^*|}{\varepsilon'}, \tag{117}$$

and

$$T^{\mathrm{sig}} \leq \begin{cases} C_1|\theta_j^*|^{-1}b_0^{-D}\ln\frac{eb_0}{a_{0j}\sqrt{D}}, & a_{0j} \leq b_0/\sqrt{D}, \\ C_1|\theta_j^*|^{-1}a_{0j}^{-1}\ln\frac{ea_{0j}\sqrt{D}}{b_0}, & a_{0j} > b_0/\sqrt{D}, D = 1, \\ C_1|\theta_j^*|^{-1}D^{-1/2}a_{0j}^{-1}b_0^{-D+1}, & a_{0j} > b_0/\sqrt{D}, D > 1. \end{cases} \tag{118}$$

Here $C_1$ and $C_2$ are absolute constants.

We bound the two terms in Equation (117). Since $j \in S$, Assumption 5.1 gives $|\theta_j^*| > \varepsilon$. If $a_{0j} \leq b_0/\sqrt{D}$, then Assumption 5.1 also gives $a_{0j} = \lambda_j^{1/2} > n^{-\delta/2}$, and hence

$$\ln\frac{eb_0}{a_{0j}\sqrt{D}} \lesssim \ln n, \qquad T^{\mathrm{sig}} \lesssim |\theta_j^*|^{-1}b_0^{-D}\ln n \lesssim b_0^{-D}\varepsilon^{-1}\ln n.$$

If $a_{0j} > b_0/\sqrt{D}$ and $D = 1$, write $u = a_{0j}/b_0 \geq 1$. Since $\sup_{u\geq 1}\ln(eu)/u = 1$,

$$a_{0j}^{-1}\ln\frac{ea_{0j}}{b_0} = b_0^{-1}\frac{\ln(eu)}{u} \leq b_0^{-1},$$

so $T^{\mathrm{sig}} \lesssim b_0^{-1}\varepsilon^{-1}$. If $a_{0j} > b_0/\sqrt{D}$ and $D > 1$, then

$$a_{0j}^{-1}b_0^{-D+1} \leq \sqrt{D}\,b_0^{-D},$$

and again $T^{\mathrm{sig}} \lesssim b_0^{-D}\varepsilon^{-1}$.

It remains to bound the approximation term in Equation (117). For fixed initial spectrum, $\tilde{d}$ is independent of $n$, and

$$\frac{\varepsilon}{\varepsilon'} = \sqrt{\ln(e + n\tilde{d})} \lesssim \sqrt{\ln n}$$

for all sufficiently large $n$. Therefore

$$\sup_{x\geq\varepsilon} x^{-(2D+2)/(D+2)}\ln^+\frac{2x}{\varepsilon'} \lesssim \varepsilon^{-(2D+2)/(D+2)}\ln n.$$

Indeed, writing $p = (2D + 2)/(D + 2)$ and $x = \varepsilon y$ with $y \geq 1$, the factor $y^{-p} \ln^+((2\varepsilon/\varepsilon')y)$ is bounded by a constant times $\ln^+(2\varepsilon/\varepsilon') + 1$, and this is $O(\ln n)$.

Using $b_0 = c_B D^{(D+1)/(D+2)} \varepsilon^{1/(D+2)}$, this gives

$$D^{D/(D+2)} |\theta_j^*|^{-(2D+2)/(D+2)} \ln^+ \frac{2|\theta_j^*|}{\varepsilon'} \lesssim b_0^{-D} \varepsilon^{-1} \ln n.$$

Combining the bounds for $T^{\text{sig}}$ and the approximation term yields

$$\bar{T}^{\text{app}}(\varepsilon') \leq C b_0^{-D} \varepsilon^{-1} \ln n = t(\varepsilon)$$

after enlarging the absolute constant $C$. This proves Equation (115). $\qquad\square$

**Proposition G.19** (Theorem-horizon tiny eigenvalue control). *Fix $\bar{\delta} > 0$. There exists a constant $K_{\text{prop}}$, depending only on $\bar{\delta}$ and the constants already fixed in Theorem 5.2, such that the following holds. Let $j \in S^c$ with $|\theta_j^*| \leq \tilde{\sigma}$, and assume that the events $E_k$ in Lemma G.1 hold for all coordinates $k$. For every $0 \leq T \leq t_2$, if $\lambda_j < n^{-K_{\text{prop}}}$ and $n$ is sufficiently large, then*

$$\tilde{\lambda}_j(T) < n^{-\bar{\delta}}.$$

*Proof.* Set

$$L := \ln \frac{1}{\lambda_j}, \qquad N := \ln n, \qquad \tilde{j} := \frac{\tilde{d}}{\lambda_j}.$$

The event $E_j$ gives

$$|\xi_j| \leq 2 C_{\text{proxy}}^{1/2} \sqrt{\frac{\ln(en\tilde{j})}{n}}.$$

The bound $|\theta_j^*| \leq \tilde{\sigma} = c'\varepsilon$ is part of the present proposition hypothesis. Since $T \leq t_2$, the definitions of $t_2$ and $b_0$ give

$$b_0^D T |\theta_j^*| \leq c' b_0^D t_2 \varepsilon = C c' c_B^D D^D \ln n.$$

For the noise term, using $\ln(en\tilde{j}) = \ln(en\tilde{d}) + L$ and the definition of $\varepsilon$,

$$b_0^D t_2 |\xi_j| \leq C_T^{(1)} \frac{\sqrt{\ln n}}{\sqrt{\ln(e + n\tilde{d})}} \sqrt{\ln(en\tilde{d}) + L}.$$

The initial spectrum is fixed, so $\tilde{d}$ is independent of $n$. After increasing $n_0$, the bound on $b_0^D t_2 |\xi_j|$ is at most $C_T^{(2)} \sqrt{\ln n + L}$. Hence, for a constant $C_T$ depending only on the constants already fixed in Theorem 5.2,

$$b_0^D T(|\theta_j^*| + |\xi_j|) \leq C_T \left( N + \sqrt{N + L} \right). \tag{119}$$

Set $C_T' := \sqrt{2e}\, C_T$, fix $\rho \in (0, 1/4)$, and fix $\eta_0 \in (0, D/(2(D+2)))$. We now choose $K_{\text{prop}}$ once. Take $K_{\text{prop}} \geq 1$ large enough so that

$$K_{\text{prop}} > \frac{1}{D+2}, \qquad \frac{1}{2\sqrt{2e}} \left( \frac{K_{\text{prop}}}{2} - \frac{1}{2(D+2)} \right) > 2C_T, \qquad K_{\text{prop}}(1 - 2\rho) > 2C_T' + \bar{\delta}. \tag{120}$$

This choice is independent of $n$. We then increase $n_0$ only. First, the definitions of $b_0$ and $\varepsilon$ give

$$\frac{b_0^2}{D^2} = c_{\text{init}}\, n^{-\frac{1}{D+2}} \left( \ln(e + n\tilde{d}) \ln n \right)^{\frac{1}{D+2}},$$

where

$$c_{\text{init}} := c_B^2 D^{\frac{2(D+1)}{D+2} - 2} \left( 2 C_{\text{proxy}}^{1/2} \right)^{\frac{2}{D+2}} > 0.$$

Since $K_{\mathrm{prop}} > \frac{1}{D+2}$,

$$\frac{n^{-K_{\mathrm{prop}}}}{\frac{b_0^2}{D^2}} = c_{\mathrm{init}}^{-1} n^{-\left(K_{\mathrm{prop}} - \frac{1}{D+2}\right)} \left(\ln(e + n\tilde{d}) \ln n\right)^{-\frac{1}{D+2}} \longrightarrow 0.$$

After increasing $n_0$, this gives $n^{-K_{\mathrm{prop}}} < \frac{b_0^2}{D^2}$ for all $n \geq n_0$. Hence $\lambda_j < n^{-K_{\mathrm{prop}}}$ implies $\lambda_j < \frac{b_0^2}{D^2}$.

Next, we verify the no-escape comparison. If $\lambda_j < n^{-K_{\mathrm{prop}}}$, then $L \geq K_{\mathrm{prop}} N$. Also

$$\ln \frac{b_0}{D} = -\frac{1}{2(D+2)} N + O(\ln N).$$

We now prove that, uniformly for $L \geq K_{\mathrm{prop}} N$,

$$C_T \left(N + \sqrt{N + L}\right) \leq \frac{1}{2\sqrt{2e}} \ln \left(\frac{b_0}{D\lambda_j^{1/2}}\right). \tag{121}$$

Let

$$\Phi_N(L) := \frac{1}{2\sqrt{2e}} \left(\frac{L}{2} + \ln \frac{b_0}{D}\right) - C_T \left(N + \sqrt{N + L}\right).$$

By increasing $n_0$, we have

$$\Phi_N'(L) = \frac{1}{4\sqrt{2e}} - \frac{C_T}{2\sqrt{N + L}} > 0, \qquad L \geq K_{\mathrm{prop}} N.$$

It is therefore enough to check $L = K_{\mathrm{prop}} N$. The expansion of $\ln \frac{b_0}{D}$ above gives, for a fixed constant $C_0$,

$$\Phi_N(K_{\mathrm{prop}} N) \geq \underbrace{\left[\frac{1}{2\sqrt{2e}} \left(\frac{K_{\mathrm{prop}}}{2} - \frac{1}{2(D+2)}\right) - C_T\right]}_{=: a_{\mathrm{esc}}} N - C_T \sqrt{(1 + K_{\mathrm{prop}})N} - C_0 \ln N.$$

The second inequality in Equation (120) gives $a_{\mathrm{esc}} > C_T$. After increasing $n_0$, the whole expression is nonnegative. This proves Equation (121).

With $B_T := b_0^D T(|\theta_j^*| + |\xi_j|)$, Equations (119) and (121) verify the no-escape condition in Lemma G.17. Therefore

$$|\beta_j(t)| \leq \frac{b_0}{D}, \qquad 0 \leq t \leq T,$$

and

$$|\beta_j(T)| \leq \lambda_j^{1/2} \exp\left(C_T' \left(N + \sqrt{N + L}\right)\right).$$

Since $L \geq K_{\mathrm{prop}} N$ and $K_{\mathrm{prop}} \geq 1$,

$$N + L \leq (1 + K_{\mathrm{prop}}^{-1})L \leq 2L.$$

After increasing $n_0$,

$$C_T' \sqrt{N + L} \leq C_T' \sqrt{2L} \leq \rho L.$$

Therefore, from $L = \ln(1/\lambda_j)$, we have

$$\exp\left(C_T' \left(N + \sqrt{N + L}\right)\right) \leq n^{C_T'} e^{\rho L} = n^{C_T'} \lambda_j^{-\rho},$$

and hence

$$|\beta_j(T)| \leq \lambda_j^{1/2 - \rho} n^{C_T'}. \tag{122}$$

Using

$$\tilde{\lambda}_j(T) = \left(\beta_j^2(T) + \lambda_j\right)\left(b_0^2 + D\beta_j^2(T)\right)^D, \tag{123}$$

Equation (122), together with $|\beta_j(T)| \leq b_0/D$, gives

$$\beta_j^2(T) \leq \lambda_j^{1 - 2\rho} n^{2C_T'}, \qquad \lambda_j \leq \lambda_j^{1 - 2\rho} n^{2C_T'},$$

where the second inequality holds after increasing $n_0$, because $\lambda_j < 1$, $1 - 2\rho < 1$, and $n^{2C'_T} \geq 1$. Hence

$$\beta_j^2(T) + \lambda_j \leq 2\lambda_j^{1-2\rho} n^{2C'_T}. \tag{124}$$

Also, the no-escape bound $|\beta_j(T)| \leq b_0/D$ gives

$$\left(b_0^2 + D\beta_j^2(T)\right)^D \leq \left(1 + \frac{1}{D}\right)^D b_0^{2D}. \tag{125}$$

Combining Equations (123) to (125) gives, with $C_{\text{eig}} := 2(1 + \frac{1}{D})^D$,

$$\tilde{\lambda}_j(T) \leq C_{\text{eig}} b_0^{2D} \lambda_j^{1-2\rho} n^{2C'_T}. \tag{126}$$

Since $b_0 = c_B D^{(D+1)/(D+2)} \varepsilon^{1/(D+2)}$ and the initial spectrum is fixed,

$$b_0^{2D} = O\left(\varepsilon^{2D/(D+2)}\right) = n^{-D/(D+2)} \text{polylog}(n).$$

Thus, after increasing $n_0$,

$$b_0^{2D} \leq n^{-D/(D+2)+\eta_0}. \tag{127}$$

Combining Equations (126) and (127), we obtain

$$\tilde{\lambda}_j(T) \leq C_{\text{eig}} n^{-D/(D+2)+\eta_0} \lambda_j^{1-2\rho} n^{2C'_T} \leq C_{\text{eig}} n^{-D/(D+2)+\eta_0 - K_{\text{prop}}(1-2\rho)+2C'_T}. \tag{128}$$

The third inequality in Equation (120) gives

$$-K_{\text{prop}}(1 - 2\rho) + 2C'_T < -\bar{\delta}.$$

Together with $\eta_0 < D/(2(D + 2))$, the exponent of $n$ in Equation (128) is at most $-\bar{\delta} - \Delta_{\text{exp}}$, where

$$\Delta_{\text{exp}} := \frac{D}{D + 2} - \eta_0 > 0.$$

After increasing $n_0$ so that $C_{\text{eig}} n^{-\Delta_{\text{exp}}} \leq 1$, we obtain $\tilde{\lambda}_j(T) < n^{-\bar{\delta}}$. $\qquad\square$

## G.6. Proofs of the Endpoint-Ordering Lemmas

All notation and standing hypotheses in this subsection are inherited from the proof of Theorem 5.2. In particular, we work on $\mathcal{E}$, the fixed time $t_1$ satisfies the hypotheses of Theorem 5.2, and Conditions (1)–(4) of Theorem 5.2 remain in force. The cutoff $m$ and the top-$m$ sets $T_1, T_2$ are defined in Equation (47); the blocks $A_1, A_2, A_3$ and $B_1, B_2$ are defined in Equations (50) and (51); the scales $s_M, \tau_A, \tau_w, \Lambda_s, U_w$ are defined in Equation (54); and $T_{\text{tiny}}$ is defined in Lemma G.4.

Before presenting the proofs, we record two one-coordinate estimates used later to compare learned eigenvalues at time $t_2$. Lemma G.20 converts a small learned eigenvalue at any time into a small initial eigenvalue. Lemma G.21 controls the relative growth of a weak tail coordinate under an upper bound on $\beta^2(t_2)$.

**Lemma G.20** (Learned-to-initial spectrum bridge)**.** *For any coordinate following Equation* (39)*, any time $t \geq 0$, any scale $q > 0$, and any constant $\alpha > 0$, if*

$$\tilde{\lambda}_i(t) < \alpha b_0^{2D} D^{-D/(D+2)} q^{2/(D+2)},$$

*then*

$$\lambda_i < \alpha D^{-D/(D+2)} q^{2/(D+2)}.$$

*Proof.* By Equation (97),

$$\tilde{\lambda}_i(t) = a_i^2(t) b_i^{2D}(t) = (\beta_i^2(t) + \lambda_i)(D\beta_i^2(t) + b_0^2)^D \geq \lambda_i b_0^{2D}.$$

Dividing by $b_0^{2D}$ gives the claim. $\qquad\square$

**Lemma G.21** (Local weak-tail relative growth). *Fix one coordinate following Equation* (39). *Let*

$$x(t) := \beta^2(t), \qquad F_\lambda(x) := (\lambda + x)(b_0^2 + Dx)^D, \qquad \tilde{\lambda}(t) = F_\lambda(x(t)).$$

*Let $t_a < t_b$, $\tau > 0$, $U > 0$, and $\delta_{\mathrm{rel}} > 0$. Suppose $x(t)$ is nondecreasing on $[t_a, t_b]$,*

$$F_\lambda(x(t_a)) \geq \tau, \qquad x(t_b) \leq U, \qquad U \leq b_0^2/D,$$

*and*

$$U\left(\frac{2^D b_0^{2D}}{\tau} + \frac{D^2}{b_0^2}\right) \leq \ln(1 + \delta_{\mathrm{rel}}/2).$$

*Then*

$$\tilde{\lambda}(t_b) \leq (1 + \delta_{\mathrm{rel}}/2)\tilde{\lambda}(t_a).$$

*Proof.* Let $x_a = x(t_a)$ and $x_b = x(t_b)$. On $[x_a, x_b]$, monotonicity gives $x \leq U \leq b_0^2/D$. Hence $(b_0^2 + Dx_a)^D \leq 2^D b_0^{2D}$. Since $F_\lambda(x_a) \geq \tau$,

$$\lambda + x_a \geq 2^{-D} \tau b_0^{-2D}.$$

For $x \in [x_a, x_b]$,

$$\frac{d}{dx} \ln F_\lambda(x) = \frac{1}{\lambda + x} + \frac{D^2}{b_0^2 + Dx} \leq \frac{2^D b_0^{2D}}{\tau} + \frac{D^2}{b_0^2}.$$

Integrating from $x_a$ to $x_b$ and using $x_b - x_a \leq U$ proves the claim. $\square$

*Proof of Lemma G.4.* Let $K_{\mathrm{prop}}$ be the threshold exponent supplied by Proposition G.19 with $\bar{\delta} = \delta_{\mathrm{tiny}}$. In Lemma G.4 set $K_{\mathrm{tiny}} := K_{\mathrm{prop}}$. This constant is independent of $n$. After $K_{\mathrm{tiny}}$ is fixed, the comparisons in Equation (56) and the bound $\kappa_{\mathrm{w}} \leq c'\varepsilon$ are enforced by increasing $n_0$. Let $i \in T_{\mathrm{tiny}}$. Then $\lambda_i < n^{-K_{\mathrm{tiny}}} = n^{-K_{\mathrm{prop}}}$, so the eigenvalue threshold required by Proposition G.19 is satisfied. Also $i \in S^c$, and Condition (2) of Theorem 5.2 gives $|\theta_i^*| \leq \tilde{\sigma}$. Thus the remaining hypotheses of Proposition G.19 are satisfied. Apply Proposition G.19 with the exponent $\delta_{\mathrm{tiny}}$, first with $T = t_1$ and then with $T = t_2$. This is allowed because $t_1 \in [0, t_2)$, so both choices satisfy the hypothesis $0 \leq T \leq t_2$ in Proposition G.19. This gives the displayed upper bound on $\tilde{\lambda}_i(t_1)$ and $\tilde{\lambda}_i(t_2)$. Since $\delta_{\mathrm{tiny}} > (D+1)/(D+2)$ and $\varepsilon$ is $n^{-1/2}$ times logarithmic factors, after increasing $n_0$ the quantity $n^{-\delta_{\mathrm{tiny}}}$ is below each of the scales $\tau_A, \tau_w, \Lambda_{\mathrm{s}}$ defined in Equation (54). This yields Equation (56). If $i \in S^c \setminus T_{\mathrm{tiny}}$, then $\lambda_i \geq n^{-K_{\mathrm{tiny}}}$, so Lemma G.1 gives the stated bound with $\tilde{i} = \tilde{d}/\lambda_i \leq \tilde{d}n^{K_{\mathrm{tiny}}}$. Moreover, by the definitions of $\kappa_{\mathrm{w}}$ and $\varepsilon$,

$$\frac{\kappa_{\mathrm{w}}}{\varepsilon} = \left(\frac{\ln(en\tilde{d}n^{K_{\mathrm{tiny}}})}{\ln(e + n\tilde{d})\ln n}\right)^{1/2}.$$

The fixed quantities $\tilde{d}$ and $K_{\mathrm{tiny}}$ do not depend on $n$, so the ratio in the preceding display tends to zero. Hence $\kappa_{\mathrm{w}} \leq c'\varepsilon$ after increasing $n_0$. $\square$

*Proof of Lemma G.5.* Since $|\theta_k^*| \geq M$, Equation (58) gives $|\theta_k^*| \geq 8\varepsilon'$. By Proposition G.18, the approximation bound holds for all $t \geq t(\varepsilon)$, where $t(\varepsilon)$ is the time threshold defined in that proposition. Since Equation (58) gives $t_2 \geq t(\varepsilon)$, Proposition G.18 applies at time $t_2$, which gives

$$|\theta_k(t_2) - \theta_k^*| \leq 2\varepsilon', \qquad |\theta_k(t_2)| \geq M - 2\varepsilon' \geq 3M/4.$$

Using Equation (97), we have $\beta_k^2 < a_k^2$ and $\beta_k^2 < D^{-1}b_k^2$. Therefore

$$|\theta_k(t_2)| = a_k(t_2)b_k^D(t_2)|\beta_k(t_2)| \leq D^{-D/(2D+2)}\left(a_k(t_2)b_k^D(t_2)\right)^{(D+2)/(D+1)}.$$

Solving for $a_k(t_2)b_k^D(t_2)$ and squaring gives

$$\tilde{\lambda}_k(t_2) \geq D^{D/(D+2)}(3M/4)^{2(D+1)/(D+2)}.$$

The upper bound on $c_s$ in Equation (58) proves Equation (59).

For the lower bound in Equation (60), we argue by contradiction. If $\beta_k^2(t_2) \leq \eta s_M$, then by Equation (97), we have

$$a_k^2(t_2) = \lambda_k + \beta_k^2(t_2) \leq (c + \eta)s_M, \qquad b_k^2(t_2) = b_0^2 + D\beta_k^2(t_2) \leq D(\bar{c}_b + \eta)s_M.$$

Consequently, using $\tilde{\lambda}_k(t_2) = a_k^2(t_2)b_k^{2D}(t_2)$, we have

$$\tilde{\lambda}_k(t_2) \leq D^D(c + \eta)(\bar{c}_b + \eta)^D s_M^{D+1} = D^{D/(D+2)}(c + \eta)(\bar{c}_b + \eta)^D M^{2(D+1)/(D+2)}.$$

Here the equality uses $s_M = D^{-D/(D+2)}M^{2/(D+2)}$, so

$$D^D s_M^{D+1} = D^{D/(D+2)}M^{2(D+1)/(D+2)}.$$

The last inequality in Equation (58) contradicts Equation (59). Thus $\beta_k^2(t_2) > \eta s_M$.

Finally, using $D^{-1}b_0^2 \leq \bar{c}_b s_M$ and Equation (60), we have

$$\frac{D^{-1}b_0^2}{\beta_k^2(t_2)} < \frac{\bar{c}_b}{\eta},$$

which proves Equation (61). $\qquad\square$

*Proof of Lemma G.6.* Assume the contrary: $\tilde{\lambda}_i(t_2) \geq \tilde{\lambda}_j(t_2)$. Since $j \in B_2$, we have $|\theta_j^*| \geq M$. By Equation (57), this implies $|\theta_j^*| \geq 8\varepsilon'$, and Equation (62) gives $t_2 \geq t(\varepsilon)$. Hence Proposition G.18 gives $|\theta_j(t_2) - \theta_j^*| \leq 2\varepsilon'$, so $|\theta_j(t_2)| \geq M/2$. Since $i \in A_1 \setminus T_{\text{tiny}}$, Lemmas G.4 and G.12 give $|\theta_i(t_2)| \leq |\theta_i^*| + \kappa_{\text{w}} \leq \tilde{\sigma} + \kappa_{\text{w}} \leq 2c'\varepsilon$. Since $M = C_M\varepsilon$, $\tilde{\sigma} = c'\varepsilon$, $\kappa_{\text{w}} \leq c'\varepsilon$, and Equation (62) gives $C_M > 4C_D c'$,

$$\frac{M}{2} = \frac{C_M\varepsilon}{2} > 2C_D c'\varepsilon \geq C_D(\tilde{\sigma} + \kappa_{\text{w}}).$$

Hence

$$|\theta_j(t_2)| > C_D|\theta_i(t_2)|.$$

Using $|\theta| = \tilde{\lambda}^{1/2}|\beta|$ and the contradiction assumption gives the beta-ratio bound. If $\beta_i(t_2) = 0$, it is immediate; otherwise,

$$\frac{|\beta_j(t_2)|}{|\beta_i(t_2)|} = \frac{|\theta_j(t_2)|}{|\theta_i(t_2)|}\sqrt{\frac{\tilde{\lambda}_i(t_2)}{\tilde{\lambda}_j(t_2)}} > C_D.$$

Thus

$$|\beta_j(t_2)| > C_D|\beta_i(t_2)|. \tag{129}$$

Because $i \in A_1$, Lemma G.20 with $q = M$ gives $\lambda_i < cs_M$. Because $j \in B_2$, Condition (4-i) of Theorem 5.2 and Lemma G.20 again give $\lambda_j < cs_M$. Together with $D^{-1}b_0^2 \leq \bar{c}_b s_M$ from Equation (62), these inequalities verify, for coordinate $j$, the hypotheses in Lemma G.5 that imply Equation (60). Therefore $\beta_j^2(t_2) > \eta s_M$.

Let $x = \beta_j^2(t_2)$. From Equation (129), $\beta_i^2(t_2) < C_D^{-2}x$. Therefore

$$\frac{D\beta_j^2(t_2) + b_0^2}{D\beta_i^2(t_2) + b_0^2} \geq \frac{1 + b_0^2/(Dx)}{C_D^{-2} + b_0^2/(Dx)}.$$

Since $x > \eta s_M$ and $D^{-1}b_0^2 \leq \bar{c}_b s_M$,

$$\frac{b_0^2}{Dx} \leq \frac{\bar{c}_b}{\eta}.$$

Since

$$\frac{1 + r}{C_D^{-2} + r} \longrightarrow C_D^2 \qquad \text{as } r \downarrow 0,$$

the two quotient conditions in Equation (62) are feasible and give

$$\frac{1 + \bar{c}_b/\eta}{C_D^{-2} + \bar{c}_b/\eta} \geq \rho_D.$$

The function $r \mapsto (1+r)/(C_D^{-2}+r)$ is decreasing for $r \geq 0$. Thus the quotient in the preceding display is at least the fixed $\rho_D \in (1, C_D^2)$. The inequality $\tilde{\lambda}_i(t_2) \geq \tilde{\lambda}_j(t_2)$, together with Equation (97), then implies

$$\beta_i^2(t_2) + \lambda_i \geq (\beta_j^2(t_2) + \lambda_j)\rho_D^D \geq \rho_D^D x.$$

Thus, from $\beta_i^2(t_2) < C_D^{-2}x$, we have

$$\lambda_i \geq x(\rho_D^D - C_D^{-2}) > c_1 s_M,$$

with $c_1 = \eta(\rho_D^D - C_D^{-2}) > 0$. The inequality $c < c_1$ in Equation (62) contradicts $\lambda_i < c s_M$. Hence $\tilde{\lambda}_j(t_2) > \tilde{\lambda}_i(t_2)$. $\quad\square$

*Proof of Lemma G.7.* The proof is divided into three parts. Part 1 proves an endpoint fitted-signal ratio, namely that the larger true signal still has larger fitted signal at time $t_2$ by a fixed factor. Part 2 uses this ratio when the initial spectra are already ordered as $\lambda_i \geq \lambda_j$. Part 3 treats the remaining case $\lambda_i < \lambda_j$, where the small learned-spectrum condition on $B_2$ makes the $b$-factor quotient dominate the possible loss in the $a$-factor quotient.

*Part 1: endpoint fitted-signal ratio.* Since $i, j \in A_2 \cup B_2 \subseteq S$, Condition (1) of Theorem 5.2 gives $|\theta_i^*| \geq M$ and $|\theta_j^*| \geq M$. The large-sample bounds and Equation (63) allow us to apply Proposition G.18 at time $t_2$, giving

$$|\theta_i(t_2) - \theta_i^*| \leq 2\varepsilon', \qquad |\theta_j(t_2) - \theta_j^*| \leq 2\varepsilon'.$$

Apply Condition (3) of Theorem 5.2 to the ordered pair $(i, j)$. With $\eta_{i,j} := |\theta_i^*| - |\theta_j^*|$, that condition says that at least one of the following three alternatives holds:

$$\eta_{i,j} \leq 0, \qquad \eta_{i,j} \geq C_\eta \varepsilon \text{ and } |\theta_i^*| \leq C_{\max}M, \qquad \frac{|\theta_i^*|}{|\theta_j^*|} > 1 + \frac{c_\eta}{D}.$$

The first alternative cannot occur because the lemma assumes $|\theta_i^*| > |\theta_j^*|$. Therefore either the second alternative gives an absolute gap together with the upper bound $|\theta_i^*| \leq C_{\max}M$, or the third alternative gives a lower bound for $\frac{|\theta_i^*|}{|\theta_j^*|}$. We verify that both remaining alternatives imply the ratio is bounded from below:

$$\frac{|\theta_i(t_2)|}{|\theta_j(t_2)|} \geq C_D = 1 + \gamma_D. \tag{130}$$

Indeed, in the case of an absolute gap, $|\theta_i^*| - |\theta_j^*| \geq C_\eta \varepsilon$ and $|\theta_j^*| \leq |\theta_i^*| \leq C_{\max}M = C_{\max}C_M\varepsilon$. Hence

$$\frac{|\theta_i(t_2)|}{|\theta_j(t_2)|} \geq \frac{|\theta_j^*| + C_\eta\varepsilon - 2\varepsilon'}{|\theta_j^*| + 2\varepsilon'} = 1 + \frac{C_\eta\varepsilon - 4\varepsilon'}{|\theta_j^*| + 2\varepsilon'}.$$

Since $|\theta_j^*| \leq C_{\max}C_M\varepsilon$, the first inequality in Equation (64) makes the endpoint ratio in the preceding display at least $1 + \gamma_D$. In the case where Condition (3) gives the ratio lower bound, $\frac{|\theta_i^*|}{|\theta_j^*|} > 1 + c_\eta/D$, and

$$\frac{|\theta_i(t_2)|}{|\theta_j(t_2)|} \geq \frac{(1 + c_\eta/D)|\theta_j^*| - 2\varepsilon'}{|\theta_j^*| + 2\varepsilon'}.$$

Since $|\theta_j^*| \geq M = C_M\varepsilon$ and the function

$$x \mapsto \frac{(1 + c_\eta/D)x - 2\varepsilon'}{x + 2\varepsilon'}$$

is increasing for $x > 0$, the second inequality in Equation (64) makes this ratio at least $C_D$. This proves Equation (130).

*Part 2: favorable initial-spectrum order $\lambda_i \geq \lambda_j$.* Consider the case where $\lambda_i \geq \lambda_j$.

If, toward contradiction, $\tilde{\lambda}_i(t_2) \leq \tilde{\lambda}_j(t_2)$, then Equation (130) gives

$$\frac{|\beta_i(t_2)|}{|\beta_j(t_2)|} = \frac{|\theta_i(t_2)|}{|\theta_j(t_2)|} \left(\frac{\tilde{\lambda}_j(t_2)}{\tilde{\lambda}_i(t_2)}\right)^{1/2} \geq C_D,$$

where $|\beta_j(t_2)| > 0$ because $|\theta_j(t_2)| \geq M - 2\varepsilon' > 0$. Thus $|\beta_i(t_2)| \geq C_D|\beta_j(t_2)|$. Using Equation (97),

$$\frac{\tilde{\lambda}_i(t_2)}{\tilde{\lambda}_j(t_2)} = \frac{\beta_i^2(t_2) + \lambda_i}{\beta_j^2(t_2) + \lambda_j} \left(\frac{D\beta_i^2(t_2) + b_0^2}{D\beta_j^2(t_2) + b_0^2}\right)^D > 1,$$

contradicting $\tilde{\lambda}_i(t_2) \leq \tilde{\lambda}_j(t_2)$. Thus $\tilde{\lambda}_i(t_2) > \tilde{\lambda}_j(t_2)$ in the present case.

*Part 3: reversed initial-spectrum order $\lambda_i < \lambda_j$.* It remains to consider $\lambda_i < \lambda_j$. By Equation (52), the set $B_2$ used here is the $B_2$ appearing in Condition (4) of Theorem 5.2. Hence Condition (4-i) applies to $j$ and gives

$$\tilde{\lambda}_j(t_1) < cb_0^{2D}s_M.$$

Applying Lemma G.20 with $q = M$, $\alpha = c$, and $t = t_1$ gives $\lambda_j < cs_M$, hence also $\lambda_i < cs_M$. The inequality $D^{-1}b_0^2 \leq \bar{c}_b s_M$ from Equation (63) verifies, for coordinate $i$, the hypotheses in Lemma G.5 that imply Equation (60). Thus $\beta_i^2(t_2) > \eta s_M$. If, toward contradiction, $\tilde{\lambda}_i(t_2) \leq \tilde{\lambda}_j(t_2)$, then Equation (130) and $|\theta_k(t_2)| = \tilde{\lambda}_k^{1/2}(t_2)|\beta_k(t_2)|$ give

$$\frac{|\beta_i(t_2)|}{|\beta_j(t_2)|} = \frac{|\theta_i(t_2)|}{|\theta_j(t_2)|} \left(\frac{\tilde{\lambda}_j(t_2)}{\tilde{\lambda}_i(t_2)}\right)^{1/2} \geq C_D.$$

Hence $\beta_i^2(t_2) \geq C_D^2\beta_j^2(t_2)$. Put $y = \beta_i^2(t_2)$. Then $y > \eta s_M$. By Equation (63), $c \leq \eta/L$ and $\bar{c}_b/\eta \leq r_0$. Moreover, $\lambda_j < cs_M$ and $D^{-1}b_0^2 \leq \bar{c}_b s_M$. Since $y > \eta s_M$, these inequalities give

$$\lambda_j < cs_M \leq \frac{\eta}{L}s_M < L^{-1}y, \qquad \frac{b_0^2}{Dy} \leq \frac{\bar{c}_b s_M}{y} < \frac{\bar{c}_b}{\eta} \leq r_0.$$

In particular,

$$\lambda_j < L^{-1}y, \qquad \frac{b_0^2}{Dy} \leq r_0.$$

Set $r = b_0^2/(Dy)$. Then $r \leq r_0$. Since $\beta_j^2(t_2) \leq C_D^{-2}y$,

$$\frac{D\beta_i^2(t_2) + b_0^2}{D\beta_j^2(t_2) + b_0^2} \geq \frac{1 + r}{C_D^{-2} + r}.$$

The function

$$f(r) := \frac{1 + r}{C_D^{-2} + r}$$

is decreasing on $r \geq 0$, because $C_D > 1$ and $f'(r) = \frac{C_D^{-2} - 1}{(C_D^{-2} + r)^2} < 0$. Since $r \leq r_0$, this gives

$$\frac{1 + r}{C_D^{-2} + r} \geq \frac{1 + r_0}{C_D^{-2} + r_0}.$$

The constant inequality in Equation (63) gives

$$\left(\frac{1 + r_0}{C_D^{-2} + r_0}\right)^D \geq C^*.$$

Combining the lower bound on the $b$-factor quotient, the bound $r \leq r_0$, and this constant inequality gives

$$\left(\frac{D\beta_i^2(t_2) + b_0^2}{D\beta_j^2(t_2) + b_0^2}\right)^D \geq C^*.$$

By the assumption $\tilde{\lambda}_i(t_2) \leq \tilde{\lambda}_j(t_2)$ and by Equation (97),

$$\frac{\tilde{\lambda}_i(t_2)}{\tilde{\lambda}_j(t_2)} = \frac{\beta_i^2(t_2) + \lambda_i}{\beta_j^2(t_2) + \lambda_j} \left( \frac{D\beta_i^2(t_2) + b_0^2}{D\beta_j^2(t_2) + b_0^2} \right)^D \leq 1.$$

Since the second factor is at least $C^*$, the first factor must satisfy

$$\frac{\beta_i^2(t_2) + \lambda_i}{\beta_j^2(t_2) + \lambda_j} \leq \frac{1}{C^*}.$$

But $\beta_j^2(t_2) \leq C_D^{-2}y$ and $\lambda_j \leq L^{-1}y$, so

$$\frac{\beta_i^2(t_2) + \lambda_i}{\beta_j^2(t_2) + \lambda_j} \geq \frac{y}{C_D^{-2}y + L^{-1}y} = \frac{1}{C_D^{-2} + L^{-1}} > \frac{1}{C^*},$$

where the final inequality follows from Equation (63). This contradiction proves the claim. $\qquad\square$

*Proof of Lemma G.8.* The proof separates the two sides of the comparison. First we obtain a time-$t_2$ lower bound for every coordinate in $A_3$. Then we obtain time-$t_2$ upper bounds for the two possible types of coordinates in $A_1$: non-tiny weak coordinates and tiny weak coordinates. The final paragraph compares these bounds with the strong-coordinate lower scale for $A_2 \cup B_2$.

Let $j \in A_3$. By definition, $j \in S^c$, $j \in T_1$, and $j \notin A_1$. Hence Condition (4-iii) of Theorem 5.2 gives $\tilde{\lambda}_j(t_1) \geq (1 + \delta_{\mathrm{w}})\tau_A$. The learned eigenvalue $\tilde{\lambda}_j(t)$ is nondecreasing in time: indeed, $\beta_j^2(t)$ is nondecreasing by Lemma G.12, and $\tilde{\lambda}_j(t) = (\lambda_j + \beta_j^2(t))(b_0^2 + D\beta_j^2(t))^D$ is increasing as a function of $\beta_j^2(t)$. Therefore

$$\tilde{\lambda}_j(t_2) \geq (1 + \delta_{\mathrm{w}})\tau_A.$$

We next upper-bound the learned eigenvalue of a coordinate in $A_1$. In the non-tiny case, Lemma G.20 converts $\tilde{\lambda}_i(t_1) < cb_0^{2D}s_M$ into $\lambda_i < cs_M$, and Lemma G.4 gives $|\xi_i| \leq \kappa_{\mathrm{w}}$ for $i \in S^c \setminus T_{\mathrm{tiny}}$.

Recall from Lemma G.4 that

$$T_{\mathrm{tiny}} = \{i \in S^c : \lambda_i < n^{-K_{\mathrm{tiny}}}\}.$$

Suppose $i \in A_1 \setminus T_{\mathrm{tiny}}$. Since $i \in A_1$, the definition of $A_1$ gives $\tilde{\lambda}_i(t_1) < cb_0^{2D}s_M$. Lemma G.20, applied with $q = M$, $\alpha = c$, and $t = t_1$, gives $\lambda_i < cs_M$. Moreover $A_1 \subseteq S^c$, so $i \in S^c \setminus T_{\mathrm{tiny}}$. Hence Lemma G.4 implies $|\xi_i| \leq \kappa_{\mathrm{w}}$. By the sign-invariant monotonicity in Lemma G.12, $|\theta_i(t_2)| \leq |z_i|$. Therefore, using $z_i = \theta_i^* + \xi_i$, Condition (2) of Theorem 5.2, and Equation (57), we have

$$|\theta_i(t_2)| \leq |z_i| \leq |\theta_i^*| + |\xi_i| \leq \tilde{\sigma} + \kappa_{\mathrm{w}} \leq 2c'\varepsilon.$$

Applying Equation (100) to the preceding bound on $|\theta_i(t_2)|$ gives

$$\beta_i^2(t_2) \leq D^{-D/(D+2)}|\theta_i(t_2)|^{2/(D+2)} \leq D^{-D/(D+2)}(2c'\varepsilon)^{2/(D+2)} \leq U_w,$$

where the last inequality follows from the definition of $U_w$ in Equation (54) and $C_w \geq 1$. Hence, by Equation (97),

$$\tilde{\lambda}_i(t_2) \leq (cs_M + U_w)(b_0^2 + DU_w)^D \leq (1 + \delta_{\mathrm{w}}/2)cs_Mb_0^{2D} = (1 + \delta_{\mathrm{w}}/2)\tau_A.$$

Here the second inequality is the first scale inequality in Equation (65), and the final equality is the definition of $\tau_A$ in Equation (54).

Suppose $i \in A_1 \cap T_{\mathrm{tiny}}$. Lemma G.4 gives $\tilde{\lambda}_i(t_2) < \tau_A/2$. Therefore, for every $i \in A_1$, we have

$$\tilde{\lambda}_i(t_2) \leq (1 + \delta_{\mathrm{w}}/2)\tau_A$$

because the tiny bound $\tau_A/2$ is smaller. Since $0 < \delta_{\mathrm{w}} < 1$, this upper bound is strictly smaller than $(1 + \delta_{\mathrm{w}})\tau_A$. Comparing with Equation (66) proves $A_3 \succ_{t_2} A_1$.

It remains to compare $A_1$ with the strong coordinates in $A_2 \cup B_2$. Lemma G.5 gives $\tilde{\lambda}_k(t_2) \geq \Lambda_{\mathrm{s}}$ for every $k \in A_2 \cup B_2$. By Equation (65), $\Lambda_{\mathrm{s}} > 2\tau_A$, while the preceding upper bound gives $\tilde{\lambda}_i(t_2) < 2\tau_A$ for every $i \in A_1$. Therefore $A_2 \cup B_2 \succ_{t_2} A_1$. $\qquad\square$

*Proof of Lemma G.9.* We first prove a common endpoint bound for every $j \in B_1 \setminus T_{\text{tiny}}$. We then treat the two cases $\tilde{\lambda}_j(t_1) \geq \tau_w$ and $\tilde{\lambda}_j(t_1) < \tau_w$ separately.

*Part 1: common endpoint bound* $\beta_j^2(t_2) \leq U_w$. Since $j \in B_1 \subseteq S^c$, Condition (2) of Theorem 5.2 gives $|\theta_j^*| \leq \tilde{\sigma}$, and Lemma G.4 gives $|\xi_j| \leq \kappa_w$. Thus $|z_j| \leq \tilde{\sigma} + \kappa_w$. Lemma G.12 gives $|\theta_j(t_2)| \leq |z_j|$. Using $\tilde{\sigma} = c'\varepsilon$ and Equation (57), we obtain $|\theta_j(t_2)| \leq 2c'\varepsilon$. Equation (100) gives

$$\beta_j^2(t_2) \leq D^{-D/(D+2)}|\theta_j(t_2)|^{2/(D+2)} \leq D^{-D/(D+2)}(2c'\varepsilon)^{2/(D+2)} \leq U_w, \tag{131}$$

where the last inequality follows from the definition of $U_w$ in Equation (54) and $C_w \geq 1$.

*Part 2: the case* $\tilde{\lambda}_j(t_1) \geq \tau_w$. We prove the first implication in the statement. The goal of the next paragraph is to verify Lemma G.21 with $t_a = t_1$, $t_b = t_2$, $\tau = \tau_w$, $U = U_w$, and $\delta_{\text{rel}} = \delta_w$. Its conclusion will be Equation (132). Assume $\tilde{\lambda}_j(t_1) \geq \tau_w$. Lemma G.12 gives the monotonicity of $x_j(t) = \beta_j^2(t)$. By the definition of $F_\lambda$ in Lemma G.21,

$$F_{\lambda_j}(x_j(t_1)) = \tilde{\lambda}_j(t_1) \geq \tau_w.$$

Also Equation (131) gives

$$x_j(t_2) = \beta_j^2(t_2) \leq U_w.$$

The first two inequalities in Equation (67) give

$$U_w \leq b_0^2/D, \qquad U_w\left(\frac{2^D b_0^{2D}}{\tau_w} + \frac{D^2}{b_0^2}\right) \leq \ln(1 + \delta_w/2),$$

which match the hypotheses in Lemma G.21 that $U \leq b_0^2/D$ and

$$U\left(\frac{2^D b_0^{2D}}{\tau} + \frac{D^2}{b_0^2}\right) \leq \ln(1 + \delta_{\text{rel}}/2).$$

Thus

$$\tilde{\lambda}_j(t_2) \leq (1 + \delta_w/2)\tilde{\lambda}_j(t_1). \tag{132}$$

The additional assumption in the first implication gives every $\ell \in T_1$ the lower bound $\tilde{\lambda}_\ell(t_1) \geq (1 + \delta_w)\tilde{\lambda}_j(t_1)$. Since learned eigenvalues are nondecreasing,

$$\tilde{\lambda}_\ell(t_2) \geq \tilde{\lambda}_\ell(t_1) \geq (1 + \delta_w)\tilde{\lambda}_j(t_1).$$

Combining this with Equation (132) gives

$$\tilde{\lambda}_j(t_2) \leq (1 + \delta_w/2)\tilde{\lambda}_j(t_1) < (1 + \delta_w)\tilde{\lambda}_j(t_1) \leq \tilde{\lambda}_\ell(t_2).$$

Thus every $\ell \in T_1$ remains above $j$ at time $t_2$. Since $|T_1| = m$, the coordinate $j$ cannot belong to the top-$m$ set $T_2$.

*Part 3: the case* $\tilde{\lambda}_j(t_1) < \tau_w$. We prove the second implication in the statement. If $\tilde{\lambda}_j(t_1) < \tau_w$, Lemma G.20, applied with $q = \varepsilon$, $\alpha = c$, and $t = t_1$, gives $\lambda_j < cD^{-D/(D+2)}\varepsilon^{2/(D+2)}$. Combining this with Equation (131) and Equation (97),

$$\tilde{\lambda}_j(t_2) \leq \left(cD^{-D/(D+2)}\varepsilon^{2/(D+2)} + U_w\right)(b_0^2 + DU_w)^D.$$

The last inequality in Equation (67) shows that this upper bound for $\tilde{\lambda}_j(t_2)$ is below $\min\{\Lambda_s/2, (1 + \delta_w/2)\tau_A\}$. For $k \in A_2 \cup B_2$, Lemma G.5 gives $\tilde{\lambda}_k(t_2) \geq \Lambda_s$. For $k \in A_3$, Condition (4-iii) of Theorem 5.2 and the monotonicity of $\tilde{\lambda}_k(t)$ give $\tilde{\lambda}_k(t_2) \geq (1 + \delta_w)\tau_A$. Thus $(A_2 \cup B_2 \cup A_3) \succ_{t_2} \{j\}$. $\qquad\square$

## H. An Application of Theorem 5.2

This appendix illustrates why the conditions in Theorem 5.2 are satisfiable and how they can be checked in a signal–spectrum misalignment example. After verifying the theorem hypotheses at the initial time $t_1 = 0$, we invoke the theorem to obtain $d^\dagger(t_2) \leq d^\dagger(0)$.

**Proposition H.1.** *Fix the depth parameter $D \geq 1$ and a finite ambient dimension $d \in \mathbb{N}_+$. Fix also a number $\delta \in (0, 1)$. Consider the sequence model in Equation (4) with noise level*

$$\sigma^2 = \frac{1}{n},$$

*initial spectrum*

$$\lambda_j = j^{-\gamma}, \qquad \gamma > 0, \qquad j \in [d],$$

*and target sequence*

$$\theta_j^* = \sum_{m=1}^{J} \alpha_m \, \mathbf{1}_{\{j = \ell(m)\}}, \qquad 1 \leq \ell(1) < \cdots < \ell(J) \leq d, \qquad \alpha_1 > \cdots > \alpha_J > 0,$$

*where $J \geq 2$, $\ell(1), \ldots, \ell(J)$, and $\alpha_1, \ldots, \alpha_J$ are fixed and do not depend on $n$.*

*Assume that the noise variables satisfy Assumption 5.1(1), namely each $\xi_j$ is sub-Gaussian with variance proxy bounded by $C_{\text{proxy}}\sigma^2$. Let*

$$\widetilde{d} := \sum_{j=1}^{d} \lambda_j, \qquad \varepsilon := 2C_{\text{proxy}}^{\frac{1}{2}} n^{-\frac{1}{2}} \sqrt{\ln(e + n\widetilde{d}) \cdot \ln n}$$

*as in Assumption 5.1. Take the constants $c$, $C$, $C_M$, $C_{\max}$, $C_\eta$, $c_\eta$, $c_B$, $\delta_{\text{w}}$, and $c'$ from Theorem 5.2 for the fixed values of $D$, $\delta$, $C_{\text{proxy}}$, and $\widetilde{d}$, and choose the initialization as in Theorem 5.2:*

$$b_0 = c_B D^{\frac{D+1}{D+2}} \varepsilon^{\frac{1}{D+2}}.$$

*Assume the following amplitude-ratio condition:*

$$\min_{1 \leq m < J} \frac{\alpha_m}{\alpha_{m+1}} > 1 + \frac{c_\eta}{D}. \tag{133}$$

*If $\ell(J) < d$, also assume the initial spectral cutoff margin*

$$\lambda_{\ell(J)} > (1 + \delta_{\text{w}})\lambda_{\ell(J)+1}, \tag{134}$$

*where $\delta_{\text{w}}$ is the constant from Theorem 5.2; if $\ell(J) = d$, no spectral cutoff margin is imposed.*

*Then, for all sufficiently large $n$, Assumption 5.1 and Conditions (1)–(4) of Theorem 5.2 hold at the initial time $t_1 = 0$. Consequently, with probability at least $1 - \frac{4}{n}$,*

$$d^\dagger(t_2) \leq d^\dagger(0) = \ell(J),$$

*where $t_2 = C \cdot D^{\frac{D}{D+2}} \varepsilon^{-\frac{2D+2}{D+2}} \ln n$ is the time appearing in Theorem 5.2.*

The role of this appendix is to translate the state conditions in Theorem 5.2, especially Condition (4), into explicit inequalities for this misalignment example. The verification is carried out at the special initial time $t_1 = 0$. More generally, Theorem 5.2 applies at any time $t_1$ satisfying its hypotheses; at that time, the sets $A_1$, $B_1$, and $B_2$ are evaluated from the learned ordering and learned eigenvalues at $t_1$. The spectral cutoff margin Equation (134) is included to verify the second alternative in Condition (4)(ii) of Theorem 5.2 at $t_1 = 0$.

*Remark* H.2. For the fixed $J \geq 2$ in Proposition H.1, a concrete case is obtained by taking

$$\ell(m) = \lfloor m^q \rfloor, \qquad \alpha_m = Am^{-\frac{p+1}{2}}, \qquad q > 1, \quad A > 0, \quad \lfloor J^q \rfloor \leq d.$$

For this concrete choice,

$$\frac{\alpha_m}{\alpha_{m+1}} = \left(\frac{m+1}{m}\right)^{\frac{p+1}{2}}, \qquad 1 \leq m < J,$$

so Equation (133) is implied by

$$\left(\frac{J}{J-1}\right)^{\frac{p+1}{2}} > 1 + \frac{c_\eta}{D}.$$

When Equation (133) holds and either $\ell(J) = d$ or Equation (134) holds, the proposition applies to this misalignment profile considered in Li & Lin (2024). The spectral cutoff margin Equation (134) is not automatic; for the spectrum $\lambda_j = j^{-\gamma}$, it is the finite-dimensional condition $\left(\frac{\ell(J)+1}{\ell(J)}\right)^\gamma > 1 + \delta_{\mathrm{w}}$ when $\ell(J) < d$.

*Proof of the proposition.* Because $d$ is fixed and finite, the quantity

$$\widetilde{d} = \sum_{j=1}^{d} j^{-\gamma}$$

is a finite constant independent of $n$. Therefore

$$\varepsilon = 2C_{\mathrm{proxy}}^{\frac{1}{2}} n^{-\frac{1}{2}} \sqrt{\ln(e + n\widetilde{d}) \cdot \ln n} \to 0 \qquad \text{as } n \to \infty.$$

This fact will be used repeatedly below.

**Step 1: Initial ordering.** At time $t = 0$, Equation (12) gives

$$a_j(0) = \lambda_j^{\frac{1}{2}}, \qquad b_j(0) = b_0, \qquad j \in [d].$$

Hence the learned eigenvalues at time $0$ are

$$\widetilde{\lambda}_j(0) = \left(a_j(0)b_j^D(0)\right)^2 = b_0^{2D}\lambda_j.$$

Since $b_0 > 0$ is common to all coordinates and $\lambda_j = j^{-\gamma}$ is strictly decreasing in $j$, the ordering of $\widetilde{\lambda}_j(0)$ is the natural ordering. Therefore

$$\pi_0^{-1}(i) = i, \qquad i \in [d].$$

**Step 2: Verification of Assumption 5.1.** Recall that

$$S := \{j \in [d] : |\theta_j^*| > \varepsilon\}.$$

Since the nonzero amplitudes are fixed and satisfy $\alpha_J > 0$, while $\varepsilon \to 0$, and since $J$ and $\lambda_{\ell(J)}$ are fixed, choose $N_0$ such that for all $n \geq N_0$,

$$\varepsilon < \alpha_J, \qquad J \leq n, \qquad \lambda_{\ell(J)} > n^{-\delta}.$$

For such $n$, every spike coordinate satisfies $|\theta_{\ell(m)}^*| = \alpha_m > \varepsilon$, while every non-spike coordinate satisfies $\theta_j^* = 0$. Therefore

$$S = \{\ell(1), \ldots, \ell(J)\}.$$

Moreover,

$$|S| = J \leq n,$$

and

$$\inf_{j \in S} \lambda_j = \lambda_{\ell(J)} = \ell(J)^{-\gamma} > n^{-\delta}.$$

The identity for $S$, the bound $|S| \leq n$, and the lower bound on $\inf_{j \in S} \lambda_j$, together with the assumed sub-Gaussian noise condition in Assumption 5.1(1), verify Assumption 5.1 for all $n \geq N_0$.

**Step 3: Verification of Condition** (1). In Theorem 5.2,

$$M := C_M \varepsilon.$$

Since $\varepsilon \to 0$ and $\alpha_J > 0$ is fixed, there exists $N_1$ such that for all $n \geq N_1$,

$$\alpha_J \geq C_M \varepsilon = M.$$

For any $j \in S$, there exists $m \in [J]$ such that $j = \ell(m)$, and hence

$$|\theta_j^*| = \alpha_m \geq \alpha_J \geq M.$$

Therefore Condition (1) holds for all $n \geq N_1$.

**Step 4: Verification of Condition** (2). In Theorem 5.2,

$$\widetilde{\sigma} := c' \varepsilon.$$

For $n \geq N_0$, if $j \in S^c$, then $\theta_j^* = 0$, so

$$|\theta_j^*| = 0 \leq \widetilde{\sigma}.$$

Hence Condition (2) holds for all $n \geq N_0$.

**Step 5: Verification of Condition** (3). Let $i, j \in S$. If $\eta_{i,j} \leq 0$, then alternative (a) in Condition (3) of Theorem 5.2 holds. It remains to consider the ordered pairs with $\eta_{i,j} > 0$, equivalently

$$|\theta_i^*| > |\theta_j^*|.$$

Then there exist indices $m, m' \in [J]$ with $m < m'$ such that

$$i = \ell(m), \qquad j = \ell(m'), \qquad |\theta_i^*| = \alpha_m, \qquad |\theta_j^*| = \alpha_{m'}.$$

Since each factor in $\prod_{r=m}^{m'-1} \alpha_r / \alpha_{r+1}$ exceeds one, Equation (133) gives

$$\frac{|\theta_i^*|}{|\theta_j^*|} = \frac{\alpha_m}{\alpha_{m'}} = \prod_{r=m}^{m'-1} \frac{\alpha_r}{\alpha_{r+1}} \geq \min_{1 \leq r < J} \frac{\alpha_r}{\alpha_{r+1}} > 1 + \frac{c_\eta}{D}.$$

Thus alternative (c) in Condition (3) of Theorem 5.2 holds for every pair with $|\theta_i^*| > |\theta_j^*|$. Hence Condition (3) is verified.

**Step 6: Computation of $d^\dagger(0)$.** We now show that

$$d^\dagger(0) = \ell(J)$$

for all sufficiently large $n$.

By Step 1, the ordering at time 0 is the natural ordering, so the sorted indices in Definition 3.1 are $1, \ldots, d$. Since $\sigma^2 = 1/n$, Definition 3.1 gives

$$d^\dagger(0) = \min \left\{ k \in [d] : \frac{1}{k} \sum_{i=k+1}^{d} (\theta_i^*)^2 \leq \frac{1}{n} \right\}.$$

We first show that $k = \ell(J)$ is feasible. Indeed, all nonzero coordinates of $\theta^*$ are located at the indices $\ell(1), \ldots, \ell(J)$, and all of these are at most $\ell(J)$. Hence

$$\sum_{i=\ell(J)+1}^{d} (\theta_i^*)^2 = 0,$$

so

$$\frac{1}{\ell(J)} \sum_{i=\ell(J)+1}^{d} (\theta_i^*)^2 = 0 \leq \frac{1}{n}.$$

Thus $k = \ell(J)$ is feasible.

Next, we show that no $k < \ell(J)$ is feasible for all sufficiently large $n$.

Since $J \geq 2$ and $1 \leq \ell(1) < \cdots < \ell(J)$, we have $\ell(J) \geq 2$. Let $1 \leq k < \ell(J)$. Then the last spike at index $\ell(J)$ lies in the tail $\{k+1, \ldots, d\}$, so

$$\sum_{i=k+1}^{d} (\theta_i^*)^2 \geq \alpha_J^2.$$

Therefore

$$\frac{1}{k} \sum_{i=k+1}^{d} (\theta_i^*)^2 \geq \frac{\alpha_J^2}{k} \geq \frac{\alpha_J^2}{\ell(J) - 1}.$$

Since $\alpha_J > 0$ and $\ell(J)$ are both fixed, while $\frac{1}{n} \to 0$, there exists $N_2$ such that for all $n \geq N_2$,

$$\frac{\alpha_J^2}{\ell(J) - 1} > \frac{1}{n}.$$

Hence every $k < \ell(J)$ is infeasible for all $n \geq N_2$. Combining this with the feasibility of $k = \ell(J)$ yields

$$d^\dagger(0) = \ell(J)$$

for all $n \geq N_2$.

**Step 7: Verification of Condition** (4). We now use the sets $A_1$, $B_1$, and $B_2$ from the *statement* of Theorem 5.2:

$$A_1 = \left\{ i \in S^c : \pi_0^{-1}(i) \leq d^\dagger(0), \ \widetilde{\lambda}_i(0) < cb_0^{2D} D^{-\frac{D}{D+2}} M^{\frac{2}{D+2}} \right\},$$

$$B_1 = \left\{ i \in S^c : \pi_0^{-1}(i) > d^\dagger(0) \right\}, \qquad B_2 = \left\{ i \in S : \pi_0^{-1}(i) > d^\dagger(0) \right\}.$$

At $t_1 = 0$, the three displayed sets $A_1$, $B_1$, and $B_2$ are the sets in Condition (4) of Theorem 5.2. Steps 3 and 4 verify the strong- and weak-signal bounds required in Conditions (1) and (2), respectively.

It remains to verify the four parts of Condition (4) of Theorem 5.2. We first identify the relevant sets $B_2$ and $A_1$, and then check the inequalities in Items (4)(ii) and (4)(iii).

**The set $B_2$ and Item** (4)(i). Since $\pi_0^{-1}(i) = i$ and $d^\dagger(0) = \ell(J)$, we have

$$B_2 = \{ i \in S : i > \ell(J) \}.$$

But every spike index is one of $\ell(1), \ldots, \ell(J)$, and thus is at most $\ell(J)$. Hence

$$B_2 = \emptyset.$$

Thus Item (4)(i) is vacuous.

**The set $A_1$.** We next prove that $A_1 = \emptyset$ for all sufficiently large $n$. Since $d^\dagger(0) = \ell(J)$ and $\pi_0^{-1}(i) = i$, every $i \in A_1$ must satisfy

$$1 \leq i \leq \ell(J), \qquad i \in S^c.$$

Therefore

$$A_1 \subset \{1, \ldots, \ell(J) - 1\},$$

which is a fixed finite set independent of $n$. Define

$$c_* := \min_{1 \leq i < \ell(J)} \lambda_i > 0.$$

Since $M = C_M \varepsilon$ and $\varepsilon \to 0$ as $n \to \infty$, we have $M \to 0$ as $n \to \infty$, and therefore

$$cD^{-\frac{D}{D+2}} M^{\frac{2}{D+2}} \to 0 \qquad \text{as } n \to \infty.$$

Because $\widetilde{\lambda}_i(0) = b_0^{2D}\lambda_i$, the learned-spectrum inequality in the definition of $A_1$ is equivalent at $t_1 = 0$ to $\lambda_i < cD^{-\frac{D}{D+2}}M^{\frac{2}{D+2}}$. After increasing the sample-size lower bound if necessary, choose $N_3 \geq \max\{N_0, N_1, N_2\}$ such that for all $n \geq N_3$,

$$(1 + \delta_{\mathrm{w}})cD^{-\frac{D}{D+2}}M^{\frac{2}{D+2}} < c_*. \tag{135}$$

But for every $1 \leq i < \ell(J)$,

$$\lambda_i \geq c_*.$$

Therefore no $i \in \{1, \ldots, \ell(J) - 1\}$ can satisfy

$$\lambda_i < cD^{-\frac{D}{D+2}}M^{\frac{2}{D+2}},$$

and hence

$$A_1 = \emptyset$$

for all $n \geq N_3$.

**Item (4)(iv).** Since $A_1 = \emptyset$ and $B_2 = \emptyset$, we have $C_{B_1} := \min\{(|A_1| - |B_2|)_+, |B_1|\} = 0$, so Item (4)(iv) holds.

**Item (4)(ii).** If $B_1 = \emptyset$ there is nothing to prove. Otherwise, let $j \in B_1$. The definition of $B_1$ gives $\pi_0^{-1}(j) > d^\dagger(0)$. Step 1 identifies $\pi_0^{-1}(j) = j$, and Step 6 gives $d^\dagger(0) = \ell(J)$; hence $j > \ell(J)$. In particular, $\ell(J) < d$, so Equation (134) applies, and

$$\widetilde{\lambda}_j(0) = b_0^{2D}\lambda_j \leq b_0^{2D}\lambda_{\ell(J)+1} < (1 + \delta_{\mathrm{w}})^{-1}b_0^{2D}\lambda_{\ell(J)} \leq (1 + \delta_{\mathrm{w}})^{-1}\min_{\ell:\pi_0^{-1}(\ell) \leq d^\dagger(0)}\widetilde{\lambda}_\ell(0),$$

where the strict inequality follows from Equation (134). The final inequality follows because Step 1 and Step 6 imply that the coordinates with $\pi_0^{-1}(\ell) \leq d^\dagger(0)$ are exactly $1, \ldots, \ell(J)$, whose minimum learned eigenvalue at time 0 is $b_0^{2D}\lambda_{\ell(J)}$. Hence every element of $B_1$ satisfies the second alternative in Item (4)(ii).

**Item (4)(iii).** Let $i \in S^c$ satisfy $\pi_0^{-1}(i) \leq d^\dagger(0)$ and $i \notin A_1$. By Step 1 and Step 6, $i \leq \ell(J)$. By Step 2, $\ell(J) \in S$, whereas $i \in S^c$; hence $i \leq \ell(J) - 1$ and $\lambda_i \geq c_*$. Since $n \geq N_3$, Equation (135) gives

$$\widetilde{\lambda}_i(0) = b_0^{2D}\lambda_i \geq b_0^{2D}c_* > (1 + \delta_{\mathrm{w}})cb_0^{2D}D^{-\frac{D}{D+2}}M^{\frac{2}{D+2}},$$

which is exactly the margin required in Item (4)(iii).

Combining these checks, Condition (4) holds for all $n \geq N_3$.

**Step 8: Application of Theorem 5.2.** Let $n_{\mathrm{Thm}}$ be the sample-size threshold supplied by Theorem 5.2, and set

$$N := \max\{n_{\mathrm{Thm}}, N_0, N_1, N_2, N_3\}.$$

For every $n \geq N$, Assumption 5.1 and Conditions (1)–(4) of Theorem 5.2 hold at $t_1 = 0$. Theorem 5.2 therefore implies that, with probability at least $1 - \frac{4}{n}$,

$$d^\dagger(t_2) \leq d^\dagger(0).$$

Using the identity established in Step 6, we obtain

$$d^\dagger(t_2) \leq d^\dagger(0) = \ell(J).$$

This completes the proof. $\square$

# I. Why Ridge Cannot Define an Intrinsic Complexity Measure

This section explains why a complexity measure defined by using the ridge estimator as a guideline is not suitable for characterizing intrinsic generalization difficulty. The key issue is the classical saturation effect of ridge: ridge cannot fully exploit the smoothness property of signals that are sufficiently smooth, so its optimal bias–variance trade-off becomes insensitive to the true source class.

## I.1. Ridge Estimator and Risk Decomposition

We work with the same Gaussian sequence model as in Equation (4). The ridge estimator can be written in the generic spectral form $\widehat{\theta}_j^{\mathrm{R},\nu} = \big(1 - \psi_\nu^{\mathrm{R}}(\lambda_j)\big)z_j$ with $\psi_\nu^{\mathrm{R}}(\lambda) = \nu/(\lambda + \nu)$, namely

$$\widehat{\theta}_j^{\mathrm{R},\nu} = \frac{\lambda_j}{\lambda_j + \nu}\, z_j.$$

Its mean squared error admits the standard bias–variance decomposition

$$\mathcal{R}_{\mathrm{R}}(\nu) := \mathbb{E}\Big[\|\widehat{\boldsymbol{\theta}}^{\mathrm{R},\nu} - \boldsymbol{\theta}^*\|_2^2\Big] = B_{\mathrm{R}}(\nu) + V_{\mathrm{R}}(\nu),$$

where the squared bias is given by

$$B_{\mathrm{R}}(\nu) = \sum_{j=1}^{d}\Big(\psi_\nu^{\mathrm{R}}(\lambda_j)\Big)^2(\theta_j^*)^2 = \sum_{j=1}^{d}\Big(\frac{\nu}{\lambda_j + \nu}\Big)^2(\theta_j^*)^2,$$

and the variance is

$$V_{\mathrm{R}}(\nu) = \sigma^2 \sum_{j=1}^{d}\Big(1 - \psi_\nu^{\mathrm{R}}(\lambda_j)\Big)^2 = \sigma^2 \sum_{j=1}^{d}\Big(\frac{\lambda_j}{\lambda_j + \nu}\Big)^2.$$

Before moving on, we introduce the ordering and a convenient reparametrization. Given the spectrum $\boldsymbol{\lambda} = \{\lambda_j\}_{j=1}^{d}$, let $\pi$ be the permutation that sorts the eigenvalues in decreasing order: $\lambda_{\pi_1} > \lambda_{\pi_2} > \cdots$ (assume no ties for simplicity), and write $\lambda_{(i)} := \lambda_{\pi_i}$. For any $\nu \in (0, \lambda_{(1)}]$, define the index

$$k_{\boldsymbol{\lambda}}(\nu) := \#\{j : \lambda_j \geq \nu\}.$$

In words, $k_{\boldsymbol{\lambda}}(\nu)$ is the number of eigenvalues no smaller than $\nu$. By definition, we have $\lambda_{(k_{\boldsymbol{\lambda}}(\nu))} \geq \nu > \lambda_{(k_{\boldsymbol{\lambda}}(\nu)+1)}$.

## I.2. Lower Bound on the Variance

We introduce a convenient lower bound for the variance term $V_{\mathrm{R}}(\nu)$.

For any $k \in [d]$, define

$$\widetilde{N}(k, d; \boldsymbol{\lambda}) := \frac{\sum_{i=k+1}^{d}\lambda_{(i)}^2}{k\,\lambda_{(k)}^2},$$

and define

$$\widetilde{V}(k) := \sigma^2\,k\big(1 + \widetilde{N}(k, d; \boldsymbol{\lambda})\big) = \sigma^2\left(k + \frac{\sum_{i=k+1}^{d}\lambda_{(i)}^2}{\lambda_{(k)}^2}\right).$$

**Proposition I.1.** *Fix $\nu \in (0, \lambda_{(1)}]$ and let $k := k_{\boldsymbol{\lambda}}(\nu)$. It holds that*

$$\frac{1}{4}\widetilde{V}(k) \leq V_{\mathrm{R}}(\nu).$$

*Proof.* We note that for the top $k$ terms, $\lambda_{(i)} \geq \nu$, so $\frac{\lambda_{(i)}}{\lambda_{(i)}+\nu} \geq \frac{1}{2}$ and hence

$$\sum_{i=1}^{k}\Big(\frac{\lambda_{(i)}}{\lambda_{(i)} + \nu}\Big)^2 \geq \frac{k}{4}.$$

For the tail terms, $\lambda_{(i)} < \nu$ and hence $\lambda_{(i)} + \nu \leq 2\nu$ for $i \geq k+1$, so $\frac{\lambda_{(i)}}{\lambda_{(i)}+\nu} \geq \frac{\lambda_{(i)}}{2\nu}$ and thus

$$\sum_{i=k+1}^{d}\Big(\frac{\lambda_{(i)}}{\lambda_{(i)} + \nu}\Big)^2 \geq \frac{1}{4\nu^2}\sum_{i=k+1}^{d}\lambda_{(i)}^2.$$

Combining the two cases, we have

$$V_{\mathrm{R}}(\nu) = \sigma^2 \sum_{i=1}^{d} \left( \frac{\lambda_{(i)}}{\lambda_{(i)} + \nu} \right)^2$$

$$\geq \sigma^2 \left( \frac{k}{4} + \frac{1}{4\nu^2} \sum_{i=k+1}^{d} \lambda_{(i)}^2 \right)$$

$$\geq \frac{1}{4} \sigma^2 \left( k + \sum_{i=k+1}^{d} \left( \frac{\lambda_{(i)}}{\lambda_{(k)}} \right)^2 \right).$$

$\square$

## I.3. A Lower Bound on the Ridge Bias at Small Regularization

The saturation phenomenon is driven by the fact that for small $\nu$ the ridge bias behaves as

$$\nu^2 \sum_{j} (\theta_j^* / \lambda_j)^2,$$

which involves the rescaled coefficients $\theta_j^* / \lambda_j$ and therefore can stop decreasing once the signal is smoother than ridge can exploit.

To be concrete, we place the following assumption.

**Assumption I.2.** There exist an index $i_0 \in [d]$ and a constant $c_r > 0$ such that

$$\sum_{i=1}^{i_0} \frac{(\theta_{\pi_i}^*)^2}{\lambda_{(i)}^2} \geq 4c_r.$$

We have the following lower bound on the squared bias term $B_{\mathrm{R}}(\nu)$.

**Proposition I.3.** *Under Assumption I.2, it holds for any $\nu \leq \lambda_{(i_0)}$ that*

$$B_{\mathrm{R}}(\nu) \geq c_r \nu^2. \tag{136}$$

*Proof.* For any $\nu \leq \lambda_{(i_0)}$, we have $\lambda_{(i)} + \nu \leq 2\lambda_{(i)}$ for all $i \leq i_0$. We then have

$$B_{\mathrm{R}}(\nu) \geq \sum_{i=1}^{i_0} \left( \frac{\nu}{\lambda_{(i)} + \nu} \right)^2 (\theta_{\pi_i}^*)^2 \geq \frac{\nu^2}{4} \sum_{i=1}^{i_0} \frac{(\theta_{\pi_i}^*)^2}{\lambda_{(i)}^2} \geq c_r \nu^2,$$

where the last inequality is due to Assumption I.2. $\square$

For any $k$, we define the proxy for the bias as

$$\widetilde{B}(k) := c_r \lambda_{(k)}^2.$$

Note that $\widetilde{B}(k)$ is comparable to the lower bound $c_r \nu^2$ if $k = k_{\boldsymbol{\lambda}}(\nu)$ and $k \geq i_0$.

## I.4. A Ridge-Guided Complexity

Suppose Assumption I.2 holds. Combining the bias proxy $\widetilde{B}(k)$ and the variance proxy $\widetilde{V}(k)$, we consider the proxy risk

$$\widetilde{R}(k) := \widetilde{B}(k) + \widetilde{V}(k) = c_r \lambda_{(k)}^2 + \sigma^2 k \big( 1 + \widetilde{N}(k, d; \boldsymbol{\lambda}) \big).$$

This leads to the ridge-guided analogue of the trade-off sequence used in the definition of ESD. Specifically, we define the *ridge trade-off sequence*

$$\bar{h}(k; \boldsymbol{\lambda}) := \frac{\lambda_{(k)}^2}{k \big( 1 + \widetilde{N}(k, d; \boldsymbol{\lambda}) \big)}.$$

**Definition I.4.** Define the *ridge saturating dimension* by

$$d^\Delta = d^\Delta(\sigma^2; \boldsymbol{\lambda}) := \max\Big(\{0\} \cup \Big\{k \in [d] : \; c_r \bar{h}(k; \boldsymbol{\lambda}) > \sigma^2\Big\}\Big).$$

The constant $c_r$ is fixed once Assumption I.2 is imposed, so we suppress it from the notation for $d^\Delta$.

By construction, the convention above gives the following boundary-adjusted balance:

$$\text{if } d^\Delta \geq 1, \quad c_r \lambda_{(d^\Delta)}^2 > \sigma^2 d^\Delta \big(1 + \widetilde{N}(d^\Delta, d; \boldsymbol{\lambda})\big), \qquad \text{if } d^\Delta < d, \quad c_r \lambda_{(d^\Delta+1)}^2 \leq \sigma^2 (d^\Delta + 1)\big(1 + \widetilde{N}(d^\Delta + 1, d; \boldsymbol{\lambda})\big).$$

It turns out that we can use $d^\Delta$ to lower bound the risk of the ridge estimator. The key point is that $d^\Delta$ is the quantity that ridge cannot beat (up to constants) when $\nu$ ranges over the natural regime $\nu \leq \lambda_{(i_0)}$.

**Proposition I.5.** *Suppose Assumption I.2 holds. For any integer $k \geq i_0$, it holds that*

$$\widetilde{R}(k) \geq \sigma^2 d^\Delta(\sigma^2; \boldsymbol{\lambda}).$$

The proof is straightforward. The case $d^\Delta = 0$ is trivial; otherwise,

- if $k \geq d^\Delta$, then $\widetilde{R}(k) \geq \widetilde{V}(k) \geq \sigma^2 k \geq \sigma^2 d^\Delta$;

- if $1 \leq k \leq d^\Delta$, then $\widetilde{R}(k) \geq \widetilde{B}(k) \geq \widetilde{B}(d^\Delta) > \sigma^2 d^\Delta$.

**Theorem I.6** (Lower bound for Ridge risk). *Suppose Assumption I.2 holds. For any $\nu \in (0, \lambda_{(i_0)}]$, let $k := k_{\boldsymbol{\lambda}}(\nu)$. Then*

$$\mathcal{R}_{\mathrm{R}}(\nu) \geq \max\Big\{c_r \nu^2, \; \frac{1}{4}\widetilde{V}(k)\Big\} \geq \frac{1}{4}\sigma^2 d^\Delta(\sigma^2; \boldsymbol{\lambda}).$$

*Consequently,*

$$\inf_{\nu \in (0, \lambda_{(i_0)}]} \mathcal{R}_{\mathrm{R}}(\nu) \geq \frac{1}{4}\sigma^2 d^\Delta(\sigma^2; \boldsymbol{\lambda}).$$

*Proof.* Fix $\nu \in (0, \lambda_{(i_0)}]$ and let $k := k_{\boldsymbol{\lambda}}(\nu)$. By definition of $k_{\boldsymbol{\lambda}}(\nu)$, we have $\lambda_{(k)} \geq \nu > \lambda_{(k+1)}$ and, since $\nu \leq \lambda_{(i_0)}$, we have $k \geq i_0$.

Under Assumption I.2, for any $\nu \leq \lambda_{(i_0)}$ we already showed in Equation (136) that $B_{\mathrm{R}}(\nu) \geq c_r \nu^2$. Furthermore, Proposition I.1 implies that $V_{\mathrm{R}}(\nu) \geq \frac{1}{4}\widetilde{V}(k)$. These two lower bounds together show that

$$\mathcal{R}_{\mathrm{R}}(\nu) = B_{\mathrm{R}}(\nu) + V_{\mathrm{R}}(\nu) \geq \max\Big\{B_{\mathrm{R}}(\nu), V_{\mathrm{R}}(\nu)\Big\} \geq \max\Big\{c_r \nu^2, \frac{1}{4}\widetilde{V}(k)\Big\}.$$

Recall

$$d^\Delta = \max\Big(\{0\} \cup \Big\{m \in [d] : \; c_r \lambda_{(m)}^2 > \sigma^2 m\big(1 + \widetilde{N}(m, d; \boldsymbol{\lambda})\big)\Big\}\Big) = \max\Big(\{0\} \cup \Big\{m \in [d] : \; c_r \lambda_{(m)}^2 > \widetilde{V}(m)\Big\}\Big).$$

If $d^\Delta = 0$, the desired lower bound is trivial. Hence we assume $d^\Delta \geq 1$ below. We consider two cases.

*Case 1: $k \geq d^\Delta$.* Then $\widetilde{V}(k) \geq \sigma^2 k \geq \sigma^2 d^\Delta$, so

$$\mathcal{R}_{\mathrm{R}}(\nu) \geq \frac{1}{4}\widetilde{V}(k) \geq \frac{1}{4}\sigma^2 d^\Delta.$$

*Case 2: $k \leq d^\Delta - 1$.* Since $\nu > \lambda_{(k+1)}$ and $k + 1 \leq d^\Delta$, monotonicity of the spectrum gives $\lambda_{(k+1)} \geq \lambda_{(d^\Delta)}$, hence $\nu \geq \lambda_{(d^\Delta)}$ and

$$\mathcal{R}_{\mathrm{R}}(\nu) \geq c_r \nu^2 \geq c_r \lambda_{(d^\Delta)}^2.$$

By definition of $d^\Delta$, we have $c_r \lambda_{(d^\Delta)}^2 > \widetilde{V}(d^\Delta) \geq \sigma^2 d^\Delta$, so

$$\mathcal{R}_{\mathrm{R}}(\nu) \geq c_r \lambda_{(d^\Delta)}^2 > \sigma^2 d^\Delta \geq \frac{1}{4}\sigma^2 d^\Delta.$$

In both cases, $\mathcal{R}_{\mathrm{R}}(\nu) \geq \frac{1}{4}\sigma^2 d^\Delta$. Taking the infimum over $\nu \in (0, \lambda_{(i_0)}]$ yields the second display. $\qquad \square$

## I.5. Why This Ridge-Guided Index Fails to Recover Source-Class Structure

As discussed in the main text (see Theorem 3.3), the minimax risk over the class

$$\mathcal{F}_{K,\boldsymbol{\lambda}}^{(\sigma^2)} = \left\{ \boldsymbol{\theta} \in \mathbb{R}^d : d^\dagger(\sigma^2; \boldsymbol{\theta}, \boldsymbol{\lambda}) \leq K \right\} \tag{137}$$

scales as $\sigma^2 K$ up to constants. Therefore, $d^\dagger$ captures the intrinsic difficulty of estimation.

By contrast, for $\nu$ in the small-regularization regime $(0, \lambda_{(i_0)}]$, the risk of the ridge estimator is lower bounded by $d^\Delta$, which depends on $\boldsymbol{\lambda}$ through $\lambda_{(k)}^2$ and $\widetilde{N}(k, d; \boldsymbol{\lambda})$, and it becomes insensitive to additional smoothness once the signal is beyond ridge's qualification. In particular, we can use the ratio

$$\frac{d^\dagger(\sigma^2; \boldsymbol{\theta}^*, \boldsymbol{\lambda})}{d^\Delta(\sigma^2; \boldsymbol{\lambda})}$$

to study whether the saturation effect appears for any given signal and spectrum. In particular, if this ratio is very small, Theorem C.8 and Theorem I.6 together suggest that, in this small-regularization regime, the risk of the ridge estimator is much larger than the risk of the PC estimator. We demonstrate this idea using the following example.

*Example* I.7. Suppose Assumption I.2 holds and consider the classical polynomial spectrum $\lambda_{(i)} = i^{-\beta}$ with degree $\beta > 1/2$. We work with $d = \infty$, or with finite $d = d(\sigma)$ sufficiently large, for instance $d(\sigma) \gg (\sigma^2)^{-1/(1+2\beta)}$, so that truncation at $d$ does not affect the rates displayed below. It can be shown that $\widetilde{N}(k, d; \boldsymbol{\lambda})$ is bounded by a constant for all $k$. Therefore, we have

$$\bar{h}(k; \boldsymbol{\lambda}) \asymp \frac{\lambda_{(k)}^2}{k} = \frac{k^{-2\beta}}{k} = k^{-(1+2\beta)}.$$

Given any $\sigma^2 > 0$, we have

$$d^\Delta(\sigma^2; \boldsymbol{\lambda}) \gtrsim (\sigma^2)^{-1/(1+2\beta)}.$$

Now consider a classical source condition: there exists $R > 0$ and $s > 0$ such that

$$\sum_{i=1}^d \lambda_i^{-s} (\theta_i^*)^2 \leq R.$$

This implies

$$h(k; \boldsymbol{\theta}^*, \boldsymbol{\lambda}) = \frac{1}{k} \sum_{i=k+1}^d (\theta_{\pi_i}^*)^2 \leq \frac{R}{k} \lambda_{(k)}^s = R\, k^{-(1+s\beta)},$$

and hence

$$d^\dagger(\sigma^2; \boldsymbol{\theta}^*, \boldsymbol{\lambda}) \lesssim (\sigma^2)^{-1/(1+s\beta)}.$$

If $s > 2$ (i.e., the signal is smoother than ridge's qualification), then

$$\frac{d^\dagger(\sigma^2; \boldsymbol{\theta}^*, \boldsymbol{\lambda})}{d^\Delta(\sigma^2; \boldsymbol{\lambda})} \lesssim (\sigma^2)^{\frac{1}{1+2\beta} - \frac{1}{1+s\beta}} \to 0 \qquad \text{as } \sigma^2 \to 0,$$

so the ridge-guided index $d^\Delta$ is asymptotically much larger than the intrinsic complexity $d^\dagger$. Equivalently, in the small-regularization regime covered by Theorem I.6, ridge has a lower bound on the order of $\sigma^2 d^\Delta$, even though the minimax rate over the same source class continues to improve with $s$ and is on the order of $\sigma^2 d^\dagger$. $\qquad\square$

The theory developed here and the above example demonstrate that within the small-regularization regime analyzed above, when the signal is sufficiently smooth, ridge cannot translate that additional smoothness into a smaller optimal bias term, and the resulting ridge-guided complexity index $d^\Delta$ fails to characterize the optimal risk. This is why it is unsuitable as a replacement for ESD in our rate statements.

