# OpenReview forum: "Alignment-Sensitive Minimax Rates for Spectral Algorithms with Learned Kernels"
_ICML.cc/2026/Conference — ICML 2026 spotlight_

### Official Review · Reviewer_BL1e · 2026-03-09

**Soundness:** 4
**Presentation:** 4
**Significance:** 3
**Originality:** 4
**Overall Recommendation:** 5
**Confidence:** 4

**Summary:**

The submission proposed a new effective dimension: effective span dimension (ESD) in regression task. Generally,  effective dimension is the basic assumption in analysis. However, the basic definition only relies on the kernel function, which is independent of data and noise. The new definition considers the data and noise, so that it can analyze the OP-GF and claim that the ESD varies during the optimization process. It is important for kernel methods and regression tasks. It may be possible to use the ESD to the integral operator framework to anaylze the learning rate of the regression task. Moreover, ESD is the canonical measure for alignment. The experiments also show the effectiveness of the definition.

**Compliance With Llm Reviewing Policy:**

Affirmed.

**Final Justification:**

The rebuttal addressed my main concerns and problems.

**Key Questions For Authors:**

1. Is it possible to specifically design  algorithms with adaptive effective dimension ?

2. Will there still be some equivalent forms of definitions?

3.The effective dimension is generally the measurement of the property of the kernel space, and the ESD considers the data and noise. Therefore, can ESD accurately measure the properties of the kernel space？

4.As the well-known analysis framework, integral operator framework, it always establishes the optimal rates based on effective dimension. Is it possible to use ESD in the framework?

**Limitations:**

yes

**Strengths And Weaknesses:**

Soundness: The whole model is soundness. It follows from the basic spectral algorithm and kernel methods. Spectral algorithm contains many well-known kernel methods, including kernel ridge regression.The ESD is well-defined for arbitrary kernels, therefore, the proposed method is useful.

Presentation: The presentation is clear and easy to follow. It first introduces kernel methods, then the ESD is proposed and compared with other effective dimensions. Finally the optimal convergence rate is established.

Significance: It is very important for kernel ridge regression or spectral algorithms, since it modifies a basic assumption: the effective dimension. Thus it may be useful to the analysis framework such as integral operator.

Originality: The new definition considers the data and noise,  which is totally new and is the canonical measure for alignment.

---

> ### Author Rebuttal · Authors · 2026-03-31
>
> Thank you for the encouraging review and for recognizing the value of our ESD framework.
> We respond to your four questions below.
>
>
> > Q1. Is it possible to specifically design algorithms with adaptive effective dimension?
>
>
> We believe this is an important future direction. Our current framework suggests that the benefit of adaptive learning is precisely the reduction of ESD, and our OP-GF analysis already provides a first tractable instance of this phenomenon. This naturally raises the question of whether one can design learning dynamics that reduce ESD more directly.
>
>
> At the same time, we would like to state the scope carefully. ESD is a **population quantity**, so it is not directly available during training. For this reason, the present paper does not propose a practical algorithm that explicitly optimizes ESD. Rather, our contribution is to identify ESD reduction as a principled mechanism behind improved generalization in adaptive spectral learning. Developing practical procedures or proxies that target ESD reduction is a natural next step, but it is beyond the scope of the current paper.
>
>
>
>
>
> > Q2. Will there still be some equivalent forms of definitions?
>
> Yes. We believe ESD can be reformulated in equivalent ways, and these formulations capture the same underlying object.
>
> First, the paper introduces not only the ESD itself, but also the span profile $D\_{\theta^\ast,\lambda}(\tau)$ and the trade-off function $H\_{\theta^\ast,\lambda}(k)$.
> These are two complementary views of the same bias-variance crossing structure. In particular, Proposition 3.6 shows that
> $$
> D_{\theta^\ast,\lambda}(\tau)=\min\\{k: H_{\theta^\ast,\lambda}(k)\le \tau\\}.
> $$
> Thus, the ESD, span profile, and trade-off function are different representations of the same alignment-sensitive quantity.
>
> Second, at the minimax level, Theorem 3.3 shows that for the ESD-bounded classes studied in the paper, any complexity measure that characterizes the same minimax rate must agree with ESD up to constants. In this sense, ESD is the canonical quantity for alignment-sensitive minimax difficulty in our setting.
>
> A broader equivalence theory for alternative formulations would be very interesting, but it is beyond the scope of the current paper.
>
>
>
> > Q3. Can ESD accurately measure the properties of the kernel space?
>
> Thank you for raising this important distinction. Our answer is that **ESD is not intended to measure the kernel space alone**, and in this sense it is fundamentally different from classical effective dimension.
>
>
> Classical effective dimension is a kernel-only or operator-only quantity: it describes the complexity of the kernel space independently of the target signal. By contrast, ESD is designed to measure the **alignment-sensitive statistical difficulty** with respect to the signal, the spectrum, and the noise level precisely because the estimation difficulty depends on all three.
>
> Therefore, we would not claim that ESD replaces classical kernel-space descriptors such as spectral decay or effective dimension. Rather, it **complements** them. If the question is "what is the intrinsic complexity of this kernel space?", then classical kernel-only quantities remain the right objects. If the question is "how favorable is this kernel for this signal at this noise level?", then ESD is the more relevant quantity.
>
>
>
>
>
> > Q4. Is it possible to use ESD in the integral operator framework?
>
> Yes. Appendix B already provides a first integration of ESD into the classical integral-operator viewpoint.
>
> We work in the Mercer eigenbasis of the kernel
> $$
> k(x,x')=\sum_j \lambda_j \phi_j(x)\phi_j(x'),
> $$
> expand the target as
> $$
> f^\ast=\sum_j \theta_j^\ast \phi_j,
> $$
> and study the transformed coordinates
> $$
> z_j=\frac{1}{n}\sum_i y_i\phi_j(x_i).
> $$
>
>
> Relative to the ideal sequence model, the new issue is the random-design fluctuation, which contributes the additional variance term
> $$
> \tau_j^2=\operatorname{Var} (f^\ast(x)\phi_j(x)).
> $$
>
> In Appendix B.2, we handle this by assuming $f^\ast\in L^\infty$, which yields the effective noise level
> $$
> \sigma^2=\frac{\sigma_0^2+\|f^\ast\|_\infty^2}{n}.
> $$
>
> This then leads to the RKHS ESD
> $$
> d^\dagger(\sigma^2;f^\ast,k)=\min\\{k:  H_{\theta^\ast,\lambda}(k)\le \sigma^2\\}.
> $$
>
>
> Appendix B then establishes minimax rates for ESD-bounded RKHS classes under a mild spectral regularity condition. Therefore, our RKHS analysis is already a first rigorous step toward using ESD within the integral-operator framework.
>
> Conceptually, what changes is **not** the operator framework. We still work with the Mercer expansion, spectral estimators, and operator-based risk analysis. What changes is the **complexity notion**: classical theory uses signal-agnostic effective dimension, whereas our framework uses the signal-aware ESD to capture alignment-sensitive difficulty. A fuller operator-level theory under more general assumptions would be a natural next step.

---

> > ### Author Rebuttal · Reviewer_BL1e · 2026-04-03
> >
> > My concerns have been adequately addressed. Thank you for your rebuttal.

---

> > > ### Author Response · Authors · 2026-04-05
> > >
> > > Thank you for reading our rebuttal and for your positive feedback. We are glad that your concerns have been adequately addressed.

---

### Official Review · Reviewer_QZUi · 2026-03-12

**Soundness:** 3
**Presentation:** 3
**Significance:** 2
**Originality:** 2
**Overall Recommendation:** 4
**Confidence:** 3

**Summary:**

This paper introduces the effective span dimension (ESD) and span profile to analyze the interplay between the signal
structure and the kernel spectrum. The framework moves beyond classical static assumptions relative to a fixed kernel
(e.g., source conditions and polynomial eigenvalue decay) and offers a dynamic, noise-dependent perspective on signal
complexity.

The key findings are:

- For models where the ESD is at most  $K$, the minimax excess risk scales as $\sigma^2 K$ (proportional to the noise times the ESD).

- For over-parameterized gradient flow, the paper proves that it can reduce the ESD.

- The paper demonstrates the generality of the ESD framework by extending it to linear models and RKHS regression.

The claims are supported by numerical experiments.

**Compliance With Llm Reviewing Policy:**

Affirmed.

**Final Justification:**

I raised my score.

**Key Questions For Authors:**

N/A

**Limitations:**

probably yes

**Strengths And Weaknesses:**

Strengths

- The introduced ESD framework as well as derived minimax optimal rates over classes defined by bounded ESD looks promising. This framework can encompass the classical minimax optimal rates over classical classes defined by source conditions, polynomial eigenvalue decay, and bounded variance noises.

Weakness
- While the paper presents the ESD as a new general framework, the core idea builds upon existing concepts in the literature on source conditions, eigenvalue decay, and variance-bias tradeoffs. The novelty may be viewed as a formalization and extension rather than a fundamentally new insight.

- Although the framework is mathematically general unifies several settings, its practical applicability beyond the presented adaptive kernel learning examples remains unclear.

- The authors expect to use the ESD framework to explain the superior generalization of neural networks by relating it to their learned representations, but this connection may be unclear.

---

> ### Author Rebuttal · Authors · 2026-03-30
>
> Thank you for the review and for recognizing that the ESD framework looks promising and can encompass classical minimax rates.
>
> We address your three concerns below.
>
> > Weakness 1: "the core idea builds upon existing concepts .... The novelty may be viewed as a formalization and extension..."
>
>
> Source conditions and eigenvalue decay conditions are not premises of our framework; they are the classical assumptions that our framework is designed to move beyond.
>
> We agree that the paper builds on the bias-variance viewpoint, but the novelty is elsewhere: **the ESD is a signal-aware population quantity that directly indexes minimax difficulty and remains meaningful for learned kernels, where the usual fixed-kernel assumptions are unavailable.**
>
>
> Classical minimax theory expresses rates through source conditions ($f^* \in [\mathcal{H}]^s$) and eigenvalue decay ($\lambda_j \asymp j^{-\gamma}$).
> When the kernel is learned from data, the corresponding $s$ and $\gamma$ for the learned kernel are generally unknown and may not even be well-defined. In that setting, the classical framework no longer provides a way to express the minimax rate.
>
>
> Theorem 3.3 addresses exactly this point by giving a minimax characterization directly over ESD-bounded classes: the minimax risk over $\{\mathrm{ESD} \le K\}$ is $\Theta(K\sigma^2)$.
> In Examples 3.4 and B.7, we show that the new framework recovers classical rates as special cases.
>
>
> In summary, our framework replaces the classical assumptions with a new complexity measure that still recovers the corresponding rates when they happen to hold, while also remaining meaningful for learned kernels, where those classical assumptions are often unavailable.
>
>
>
> > Weakness 2: "its practical applicability beyond the presented adaptive kernel learning examples remains unclear."
>
>
> We agree that the intended notion of applicability should be stated more clearly.
>
>
> We aim to develop a population-level explanatory framework that relates alignment, minimax risk, and adaptive kernel learning.
> Therefore, applicability here means **whether the framework yields informative population-level conclusions across multiple models and adaptive regimes**, rather than whether it provides a training procedure.
>
>
> First, our ESD framework covers sequence models, linear regression, and RKHS regression. In all of these settings, the ESD is used to define ESD-bounded classes and to characterize the minimax risk.
>
> Second, the intended role of the ESD is explanatory.
> By quantifying how signal-kernel alignment affects minimax difficulty, the ESD can be used to **compare kernels** through alignment and to **quantify whether adaptation moves a target into an easier class**.
> In particular, when adaptive learning reshapes the kernel so that the signal aligns better with the leading directions, the ESD decreases, and the same target function moves into a class with smaller minimax risk.
>
>
> Third, the paper already supports this viewpoint at two levels.
> In addition to the proven ESD reduction in Section 5, we introduce the pathwise ESD viewpoint in Appendix C.2 for more general learned kernels, where eigenfunctions evolve during training, and include a deep linear network experiment to illustrate this broader perspective.
>
>
> We will revise the paper to make this intended scope and role more explicit.
>
>
> > Weakness 3: "The authors expect to use the ESD framework to explain the superior generalization of neural networks ..., but this connection may be unclear."
>
>
> We agree that our current discussion of neural networks should be framed more carefully. Our intended claim has three levels.
>
>
> At the **theorem level**, our rigorous adaptive result is Theorem 5.2, which analyzes the first tractable regime of kernel adaptation (eigenvalue learning under a fixed eigenbasis).
>
>
> At the **formulation level**, the ESD itself is not restricted to fixed eigenfunctions.
> For a learned kernel $k_t$ with eigenvalues $\{\lambda\_j(t)\}$ and eigenfunctions $\{\phi\_j^{(t)}\}$, Appendix C.2 defines the ESD during training by $d^\dagger(t)=d^\dagger(\sigma^2;f^\ast,k\_t)$ using the time-dependent coefficients $\theta\_j^{\ast,(t)}=\langle f^\ast,\phi\_j^{(t)}\rangle$.
> Thus, the same ESD definition extends directly to learned kernels with time-varying eigenfunctions.
>
>
> At the **experiment level**, Appendix C.2 also includes a deep linear network experiment in this broader setting, illustrating that the pathwise ESD decreases during training along with improved generalization ability.
>
> Therefore, the paper proves the adaptive result in the first tractable case, and then shows how the same ESD viewpoint extends beyond that case at the level of formulation and experiment.
> In this sense, the limitation is the scope of the current adaptive theorem, not the scope of the ESD concept itself.
> We will revise the introduction and discussion to make this distinction clear.

---

> > ### Author Rebuttal · Reviewer_QZUi · 2026-04-04
> >
> > Thank you for your response. I will raise my score, though I may still have some concerns about the novelty and the potential impact of the notion in some other areas.

---

> > > ### Author Response · Authors · 2026-04-05
> > >
> > > We sincerely thank you for taking the time to reconsider the paper and for raising your score. We appreciate your continued consideration, and we will make the scope and positioning of the paper clearer in the revision.

---

### Official Review · Reviewer_gCEN · 2026-03-13

**Soundness:** 3
**Presentation:** 3
**Significance:** 3
**Originality:** 2
**Overall Recommendation:** 5
**Confidence:** 2

**Summary:**

This paper gives an approach for understanding the performance of spectral learning algorithms that applies when the kernel is learned from data, which isn't adequately handled by classical theory. It introduces the Effective Span Dimension (ESD), a complexity measure that doesn't depend on noise and that quantifies alignment between the target signal and the kernel spectrum. The authors show that, in sequence models, the minimax excess risk over classes of signals with bounded ESD scales linearly with the ESD. They further show that a properly tuned principal components estimator achieves this rate. The proposed approach doesn't rely on usual assumptions such as eigenvalue decay or source conditions and recovers classical rates as special cases. They extend the framework to linear and kernel regression by reduction to sequence models. They also analyze a stylized over-parameterized gradient flow model that adapts kernel eigenvalues, giving conditions under which training reduces the ESD. Finally, they give synthetic experiments illustrating improved signal–kernel alignment during training.

**Compliance With Llm Reviewing Policy:**

Affirmed.

**Final Justification:**

The rebuttal addressed my main concerns, hence my revision of the score.

**Key Questions For Authors:**

1. The main minimax results are proved for sequence models. The conditions needed for extensions to kernel regression are not discussed enough to be made clear to me. A further discussion of the precise conditions and their implications would help make clear how generally applicable the framework is.

2. The analysis for the adaptive case is based on fixed eigenfunctions; what happens when the eigenfunctions themselves evolve? This would be expected in many important ML settings e.g. neural networks.

**Limitations:**

Yes

**Strengths And Weaknesses:**

Soundness: As far as I can tell, the paper is sound. The results are most powerful in the case of sequence models, and the use of reductions  to analyze linear regression and kernel regression shows how the theory can be applied more generally. The role and significance of the ESD is well-developed. However, it is not obvious how tight the minimax results remain under finite-sample effects, empirical kernel estimation errors, or evolving eigenfunctions. The implications of the conditions used in the analysis for adaptivity are discussed enough, making it unclear how applicable the results are.
Presentation: The key idea of the ESD is clearly defined and well-motivated. The motivation is very clear. The overall narrative of the paper is clear. The paper sometimes switches between rigorous proofs and heuristic-backed claims without making it very clear which is which, though this could likely be easily improved. The limitations of the approach, in particular how restrictive the assumptions are, could be made clearer.
Significance: The problem of understanding the performance of spectral algorithms based on adaptively-learned kernels is important. The notion of ESD offers an approach that doesn't rely on the usual assumptions made by other approaches. The work is primarily theoretical and uses the ESD to give an explanation for why adaptive kernels can generalize well. Again, the insufficient discussion about the limitations of the approach make it difficult to evaluate how significant the approach is.
Originality: To the best of my knowledge the idea of ESD is novel. The analysis done for ESD-bounded classes is similarly novel. This work shows that the ESD may be a useful tool for moving beyond standard fixed kernel analysis. I am open to updating my score if these concerns are adequately addressed.

---

> ### Author Rebuttal · Authors · 2026-03-30
>
> Thank you for the careful reading and for being open to updating your assessment. We respond to your two questions below.
>
> ## Response to Q1 and weakness in soundness
>
> We agree that this part should be discussed more explicitly.
>
> We first clarify the role of Appendix B. Since the ESD is a population complexity measure, Appendix B extends the ESD framework to RKHS regression at the population level; it is not intended as a separate analysis of empirical kernel eigensystem estimation.
>
>
> ---
>
>
> We next explain the role of **Assumption B.1**. As shown in Appendix B.1, after writing the transformed coordinates as
> $$z\_j=\theta\_j^\ast+\xi\_j+\Delta\_j,$$
> the only additional difficulty relative to the sequence model is the fluctuation term $\Delta\_j$ introduced by random design. This is the only relevant finite-sample issue in Appendix B, and it is precisely what is absorbed into the effective noise level.
>
> Assumption B.1 is imposed to control this term. It assumes that $f^\ast$ is essentially bounded, which is mild and satisfied by, for example, any continuous function on a compact domain. As shown in Equation (23) of Appendix B.2, the random covariates inflate the per-coordinate variance from $\sigma\_0^2/n$ to $\left(\sigma\_0^2+\|f^\ast\|\_\infty^2\right)/n$. Assumption B.1 controls this inflated variance term, so that the oracle KPCPE risk is governed by $d^\dagger/n$ as in the sequence model (Proposition B.3). In this sense, Assumption B.1 is the key condition in the RKHS extension.
>
>
> ---
>
>
> To obtain the $K/n$ minimax scaling, we further impose **Assumption B.4**. This is a regularity condition on the kernel spectrum and on the scale parameter $K$. It ensures that the leading $O(K)$ directions relevant for estimation are not so weak relative to sample size $n$ that the $K/n$ scaling becomes impossible. Thus, Assumption B.4 is not an assumption on $f^\ast$; it is the spectral condition needed for the ESD-bounded class to have the expected minimax behavior in RKHS regression.
>
> As discussed in **Example B.7**, Assumption B.4 is satisfied in the classical setting with polynomial eigenvalue decay and source conditions, so the classical minimax rate is recovered as a special case.
>
> ---
>
> In summary, Assumption B.1 controls random-design fluctuation, while Assumption B.4 ensures the $K/n$ minimax scaling remains valid in the RKHS setting.
>
>
> We will revise the paper to state the meaning of Assumptions B.1 and B.4, and the scope of Appendix B more explicitly.
>
> ## Response to Q2 and weakness in significance
>
> This is an important point. When the eigenfunctions themselves evolve, the ESD is still well defined. As stated in Appendix C.2, for a learned kernel $k\_t$ with eigenvalues $\{\lambda\_j(t)\}$ and eigenfunctions $\{\phi\_j^{(t)}\}$, we define the ESD at time $t$ by $d^{\dagger}(t)=d^{\dagger}(\sigma^2 ; f^\ast, k\_t)$ using the time-dependent coefficients $\theta\_j^{\ast,(t)}=\langle f^\ast, \phi\_j^{(t)}\rangle$.
> Accordingly, improved alignment means that the signal coefficients become more concentrated on the leading indices of the learned basis. Equivalently, the tail functional $H\_t(k)=\frac{1}{k} \sum\_{i>k}(\theta\_i^{\ast,(t)})^2$ decreases, which in turn implies a decrease in the pathwise ESD.
>
> Therefore, the limitation is not the scope of the ESD itself, but the scope of Theorem 5.2, which analyzes a tractable adaptive setting where one can rigorously track ESD reduction with a fixed eigenbasis. For the more general case with evolving eigenfunctions, Appendix C.2 already provides the corresponding formulation, and the deep linear network experiment gives supporting evidence for this broader viewpoint.
>
>
>
> ## Response to weakness in presentation
>
> We agree that the current draft does not mark this distinction clearly enough.
>
> The rigorous results are the sequence-model minimax theory, the linear model extension in Appendix A, the RKHS extension in Appendix B, and Theorem 5.2 for eigenvalue learning under a fixed eigenbasis.
>
> By contrast, the RKHS reduction as presented in Section 2 is motivational before the rigorous Appendix B treatment, and Appendix C.2 gives the broader evolving-eigenfunction viewpoint at the level of formulation and supporting experiment rather than as a full theorem. We will revise the paper to flag these transitions explicitly.
>
> We thank the reviewer again for these constructive suggestions.

---

> > ### Author Rebuttal · Reviewer_gCEN · 2026-04-04
> >
> > My concerns were fully addressed. I thank the authors for their enlightening responses.

---

> > > ### Author Response · Authors · 2026-04-05
> > >
> > > Thank you for your thoughtful follow-up and for your encouraging feedback. We are very glad that our responses were helpful and that your concerns were fully addressed.

---

### Official Review · Reviewer_bHVq · 2026-03-13

**Soundness:** 3
**Presentation:** 3
**Significance:** 3
**Originality:** 3
**Overall Recommendation:** 5
**Confidence:** 3

**Summary:**

The paper proposes the Effective Span Dimension, a signal-aware complexity measure for spectral algorithms that depends jointly on the signal, kernel spectrum, and noise level. In contrast to classical signal-agnostic measures, ESD captures signal-kernel alignment and yields sharp minimax rates without requiring source conditions or polynomial eigenvalue decay. The authors further show that over-parameterized gradient flow provably reduces ESD, connecting adaptive feature learning to generalization improvement.

**Compliance With Llm Reviewing Policy:**

Affirmed.

**Final Justification:**

The authors have provided a solid response that addressed my initial concerns and the previously identified weaknesses have been either resolved or mitigated. In particular, the authors have extensively clarified the discussion around Theorem 5.2, the limitations of their setting, and the practical contributions. Since the authors have committed to incorporating these clarifications in the final version, I am increasing my score to Accept.

**Key Questions For Authors:**

1. See Weakness 1. Could the author clarify these conditions, extending the discussion and providing concrete examples of models satisfying them?
2. Are there identifiable conditions under which adaptive training increases the ESD, and can the framework predict such failure in advance?
3. The ESD is admittedly introduced for theoretical purposes. However, do the author have some intuitions on possible agnostic and consistent procedures to estimate it (or a surrogate quantity) in practical scenarios?

**Limitations:**

Yes

**Strengths And Weaknesses:**

### Strengths
The paper makes a genuine and well-motivated contribution filling a gap in the theory of spectral methods, and the authors make a convincing case that the ESD is the canonical quantity for alignment-sensitive minimax rates. The framework is technically sound and the exposition is clear and well-organized throughout. Notably, the work is broad and its ideas extend coherently across sequence models, linear regression, and RKHS regression.

### Weaknesses
1. The conditions under which Theorem 5.2 holds are opaque, hard to parse and not sufficiently commented upon.
2. The contribution is weakened by the lack of analysis for eigenfunctions adaptation, leaving a gap with respect to more realistic scenarios.
3. As acknowledged, the empirical scope is limited, and, by definition, experiments can only be performed in synthetic settings. See also Question 3.
4. Minor weakness: the figures' labels should be enlarged.

---

> ### Author Rebuttal · Authors · 2026-03-31
>
> Thank you for the careful and constructive review. We are glad that you find the paper technically sound and well motivated. We address your questions and concerns below.
>
> ## Response to Q1 and Weakness 1
>
> We agree that the discussion around Theorem 5.2 is too compressed. Informally, the theorem considers a regime where:
>
> 1. there is a set $S$ of coordinates with strong signals;
> 2. coordinates outside $S$ have weak signals;
> 3. within $S$, signal magnitudes are sufficiently separated, so that problematic near-ties are excluded; and
> 4. at time $t_1$, the learned-eigenvalue ordering is not so unfavorable that too many strong coordinates lie behind weak ones.
>
> Under these conditions, OP-GF amplifies coordinates carrying strong signal faster than weak ones, improves the ordering of the learned eigenvalues, and therefore reduces the ESD.
>
> A concrete class covered by Theorem 5.2 is the misalignment model from Section 6.
> Take $t_1=0$ and $\sigma^2=1/n$, and let
> $$
> \lambda\_j=j^{-\gamma}, \quad \theta\_j^*=\sum_{m=1}^J a\_m \mathbf{1}\_{\{j=\ell(m)\}},
> $$
> where $J, \ell(1), \ldots, \ell(J)$ are fixed, and the amplitudes satisfy $a\_1>\cdots>a\_J>0$ and
> $$
> \min\_{1\le m<J}\frac{a\_m}{a\_{m+1}}>1+\frac{c\_\eta}{D}.
> $$
>
> For large $n$, since $\varepsilon \to 0$ and $a_J>0$ is fixed, we have $S=\{\ell(1),\ldots,\ell(J)\}$. Hence Conditions 1 and 2 hold, and Condition 3 is implied by the assumption on $a_m$.
> At time $t_1=0$, one can also verify that the ESD cutoff occurs at the last spike location, so $B_2=A_1=\emptyset$ and Condition 4 holds.
>
> In particular, the choice $a_m=m^{-(p+1)/2}$ and $\ell(m)=\lfloor m^q\rfloor$ with $q>1$ recovers the setting used in Section 6.
>
>
> We will revise the paper to add this interpretation in the main text and place the full derivation in the appendix.
>
> ## Response to Q2
>
> We appreciate this interesting question.
> A theory of when adaptive training increases the ESD is beyond the scope of the current paper, but Theorem 5.2 already gives a useful first step.
>
>
> Theorem 5.2 identifies conditions under which OP-GF improves alignment and reduces the ESD.
> At the same time, it isolates a structural bottleneck for successful adaptation.
> In particular, Condition 4 excludes cases where too many strong coordinates are buried too far behind weak ones near the ESD cutoff.
> If this happens, OP-GF may not reorder the learned eigenvalues enough to improve alignment, and the ESD reduction guarantee can fail.
>
>
> Thus, while we do not yet characterize all cases where adaptive training increases the ESD, our theory already helps identify configurations where ESD reduction is less likely. We will revise the paper to make this interpretation more explicit and to highlight it as a natural direction for future work.
>
>
> ## Response to Q3 and Weakness 3
>
> We appreciate this important question.
>
> Practical surrogates are plausible.
> In the sequence model, because the ESD corresponds to the PC truncation level where bias and variance balance, one can estimate a surrogate by selecting the truncation point through data-driven risk estimation, for example via penalized projection methods in the spirit of [1].
>
> We will discuss this point in the revision.
>
> ## Response to Weakness 2
>
> This is an important point, and we agree that the current theorem-level analysis for adaptive dynamics is limited to eigenvalue learning under a fixed eigenbasis.
> This is the first tractable adaptive regime in which our current tools allow a rigorous analysis of ESD reduction.
>
>
> However, this limitation is in the adaptive analysis, not in the ESD definition itself. As noted in Appendix C.2, for a learned kernel $k\_t$ with eigenvalues $\{\lambda\_j(t)\}$ and eigenfunctions $\{\phi\_j^{(t)}\}$, we define the pathwise ESD by
> $$
> d^\dagger(t)=d^\dagger(\sigma^2;f^\ast,k\_t),
> $$
> using the time-dependent coefficients $\theta\_j^{\ast,(t)}=\langle f^\ast,\phi\_j^{(t)}\rangle$.
> Thus the same ESD concept extends directly to evolving eigenfunctions. Appendix C.2 includes deep linear network experiments to illustrate this broader framework-level viewpoint.
>
> What is currently missing is a general theorem, analogous to Theorem 5.2, for this broader adaptive regime. We view this as a natural next step opened by the ESD framework.
>
> We will revise the paper to make this distinction more explicit.
>
>
> ## Response to Weakness 4
>
> We agree and will enlarge the labels in the revision.
>
>
>
>
> **Reference**
>
> [1] Birgé, L. and Massart, P., 2001. Gaussian model selection. Journal of the European Mathematical Society, 3(3), pp.203-268.

---

> > ### Author Rebuttal · Reviewer_bHVq · 2026-04-01
> >
> > I thank the authors for their rebuttal. My understanding of Theorem 5.2 is now much clearer, and the rebuttal has resolved my previous doubts. I believe that integrating these clarifications into the revised version will strengthen the paper's presentation and clarity.

---

> > > ### Author Response · Authors · 2026-04-05
> > >
> > > Thank you for your careful reading of our rebuttal. We are glad that our clarification has addressed your previous concerns, and we will incorporate it into the revised version.

---

### Decision · Program_Chairs · 2026-04-30

**Decision:**

Accept (spotlight)

**Comment:**

This paper studies the problem of learning the reproducing kernel through a newly introduced tool, effective span dimension (ESD). Solid theoretical investigation is carried out supplemented by numerical experiments. The idea is to align kernel with signal from data. Main issues identified by reviewers include writing problem (by Reviewers bHVq, QZUi, and gCEN; e.g., Theorem 5.2) which I think can be quickly fixed; the coverage of the cases with evolving eigenfunctions (by Reviewers bHVq and gCEN) which is a true limitation but this is reasonable to be deferred as future works considering the current manuscript is already substantial in scope and length; novelty issue mentioned by Reviewer QZUi but not shared across the discussion.